# Interplay and cooperation between SREBF1 and master transcription factors regulate lipid metabolism and tumor-promoting pathways in squamous cancer

Squamous cell carcinomas (SCCs) comprise one of the most common histologic types of human cancer. Transcriptional dysregulation of SCC cells is orchestrated by tumor protein p63 (TP63), a master transcription factor (TF) and a well-researched SCC-specific oncogene. In the present study, both Gene Set Enrichment Analysis (GSEA) of SCC patient samples and in vitro loss-of-function assays establish fatty-acid metabolism as a key pathway downstream of TP63. Further studies identify sterol regulatory element binding transcription factor 1 (SREBF1) as a central mediator linking TP63 with fatty-acid metabolism, which regulates the biosynthesis of fatty-acids, sphingolipids (SL), and glycerophospholipids (GPL), as revealed by liquid chromatography tandem mass spectrometry (LC-MS/MS)-based lipidomics. Moreover, a feedback co-regulatory loop consisting of SREBF1/TP63/Kruppel like factor 5 (KLF5) is identified, which promotes overexpression of all three TFs in SCCs. Downstream of SREBF1, a non-canonical, SCC-specific function is elucidated: SREBF1 cooperates with TP63/KLF5 to regulate hundreds of cis-regulatory elements across the SCC epigenome, which converge on activating cancer-promoting pathways. Indeed, SREBF1 is essential for SCC viability and migration, and its overexpression is associated with poor survival in SCC patients. Taken together, these data shed light on mechanisms of transcriptional dysregulation in cancer, identify specific epigenetic regulators of lipid metabolism, and uncover SREBF1 as a potential therapeutic target and prognostic marker in SCC.

Cancer is characterized by dysregulated metabolic activity leading to enhanced cell growth and proliferation[1,2]. The most well-understood metabolic perturbations in cancer cells include the Warburg effect[3] (glucose metabolism) and glutamine metabolism[4]. Alterations in lipid metabolism in cancer are also becoming increasingly recognized. In fact, several decades ago, tumors were found to synthesize fatty acids de novo[5] and to exhibit a shift toward fatty-acid synthesis[6]; in contrast, most normal human cells prefer exogenous sources of fatty acids. Indeed, tumors show heightened de novo fatty-acid synthesis to sustain growth and proliferation, because lipids are not only components of biological membranes, but also have important roles in signal transduction[7]. Consistently, reducing fatty-acid availability by blocking fatty-acid synthesis pathways (e.g., through inhibition of the enzymes FASN, ACLY, ACC, and SCD) has demonstrated antitumor activity in different cancer types[8–10]. At the regulatory level, hyperactive fatty-acid synthesis and lipogenesis pathways in cancer are often associated with dysregulation of sterol regulatory element-binding proteins (SREBPs)[11], which control the expression of factors involved in the uptake and synthesis of cholesterol, fatty acids, and phospholipids[12–14]. Of the two mammalian SREBP genes, SREBF1 mainly regulates the expression of factors required for fatty-acid synthesis, while SREBF2 is responsible for cholesterol homeostasis[12,15]. SREBF1 has strong protumor functions in several cancer types, including prostate, breast, and liver cancers[11]. SREBF inhibitors, such as Fatostatin and betulin, markedly suppress tumor cell growth and viability[16–19]. The levels and activities of SREBF1 are tightly controlled by endogenous sterol levels via negative feedback regulation. In cancer cells, SREBF1 protein is also stabilized and activated by the PI3K/Akt/mTOR signaling pathway[20,21]. However, cancer-specific epigenomic regulation of SREBF1 remains unclear.

Squamous cell carcinomas (SCCs) are malignancies derived from stratified squamous epithelium of multiple organs, including esophagus (ESCC), head and neck (HNSC), lung (LUSC), cervix (CESC), and skin (SSCC). Several common forms of SCC, such as ESCC and LUSC, are especially lethal and have very few, if any, effective therapeutic targets. Despite arising from various anatomical locations, SCCs share many genomic characteristics specific to the squamous cell lineage. For example, one of the most notable abnormalities specific to SCC is genetic lesions targeting squamous cell differentiation factors (TP63, SOX2, ZNF750, and NOTCH family members)[22–28]. As one of the most well-established master transcription factors (TFs), TP63 plays an essential role in both normal squamous differentiation and SCC development. Indeed, we and others have shown that in normal development, TP63 controls squamous cell identity[29], while amplification and overexpression of TP63 promotes SCC tumorigenesis and progression[30–33]. Mechanistically, TP63 occupies and controls hundreds to thousands of cis-regulatory elements (particularly enhancers and super-enhancers), and orchestrates gene regulatory networks for cell-type-specific functions[30–33]. However, the vast majority of previous mechanistic studies have been performed using cell line models in vitro. To identify putative in vivo TP63-target genes and signaling pathways in pan-SCC patients, in the current study we initially performed Gene Set Enrichment Analysis (GSEA) using SCC patient samples. This unbiased analysis identified fatty-acid metabolism as one of the most significantly enriched pathways downstream of TP63 across different types of SCCs. In addition, SREBF1 was pinpointed as the central mediator between TP63 and fatty-acid metabolism.

These results are of great potential interest, considering that dysregulated fatty-acid metabolism is a metabolic hallmark of cancer[7]. However, how fatty-acid metabolism is dysregulated in the context of SCC is not well understood. Moreover, whether and how TP63 regulates SCC cell biology by controlling fatty-acid metabolism remains unexplored. In addition, the regulatory relationship between TP63 and SREBF1 is unknown, and how SREBF1 is epigenetically activated in cancer in general remains unclear.

In this work, we perform epigenomic, metabolomic and transcriptomic analyses in SCC and elucidate that SREBF1 acts as a central mediator linking TP63 with fatty-acid metabolism. Moreover, we find that SREBF1 cooperates with TP63/KLF5 to regulate hundreds of cis-regulatory elements across the SCC-specific epigenome, which converges on activating cancer-promoting pathways, uncovering SREBF1 as a potential therapeutic target and prognostic marker in SCC.

## Results

**TP63 regulates fatty-acid metabolism pathway through SREBF1.** To explore putative TP63-regulated signaling pathways in vivo, we initially performed GSEA using SCC patient samples. In an unbiased fashion, we identified signaling pathways enriched in SCC samples with high TP63 expression by analyzing RNA-Seq data from TCGA cohorts, including ESCC ($n = 81$), HNSC ($n = 436$) and LUSC ($n = 501$) samples. Cervical SCC and human papillomavirus (HPV)[+] HNSC samples were excluded, because HPV[+] tumors have different gene expression programs than HPV[-] SCCs[25,34]. Tumor samples from each SCC cohort were stratified as either TP63-high (top 30%) or TP63-low (bottom 30%) based on expression of this TF. Genes that were differentially expressed between TP63-high and TP63-low groups were first determined. Notably, genes exhibiting either positive or negative correlation with TP63 expression were highly overlapping between these three types of SCCs (Fig. 1a and Supplementary Data 1). For example, 72.8% (182/250) of positively correlated genes identified in ESCC were shared with either LUSC ($P < 1E-06$) or HNSC ($P < 1E-06$) cohorts. This high degree of overlap supports the notion that TP63 shares biological functions among different types of SCC by directly regulating gene transcription[30,31,33,35,36]. These differentially expressed genes were next used to perform GSEA with the Hallmark geneset. Again, signaling pathways significantly enriched in TP63-high samples were largely overlapping between the three major types of SCCs (Fig. 1b), suggesting a uniform role of TP63 in regulating gene expression programs across different types of SCC.

Among the nine shared signaling pathways, cell-cycle-related (E2F targets, G2M checkpoint, mitotic spindle)[37], MYC[38,39], and mTOR[40] signaling pathways were expectedly enriched, consistent with findings from previous studies of TP63. In addition, enrichment of the estrogen response late pathway was also observed. Interestingly, two lipid metabolism-related pathways (fatty-acid metabolism and cholesterol homeostasis) were identified (Fig. 1b, right panel). We next performed similar GSEA across all cancer types, which showed that fatty-acid metabolism was positively correlated with TP63 only significantly in SCC samples, with the exception of thyroid cancer (Fig. 1c); cholesterol homeostasis did not show such SCC specificity (Supplementary Fig. 1a). As discussed earlier, dysregulated fatty-acid metabolism is a metabolic hallmark of cancer. However, the biological significance of fatty-acid metabolism in SCC, as well as fatty-acid metabolism is regulated by TP63, have not been explored. Therefore, we next focused on characterizing the regulatory function of TP63 on fatty-acid metabolism in SCCs.

To validate these TCGA-based results, additional GSEA was performed using other independent, large-scale SCC transcriptomic datasets, including ESCC (GSE53624) and LUSC

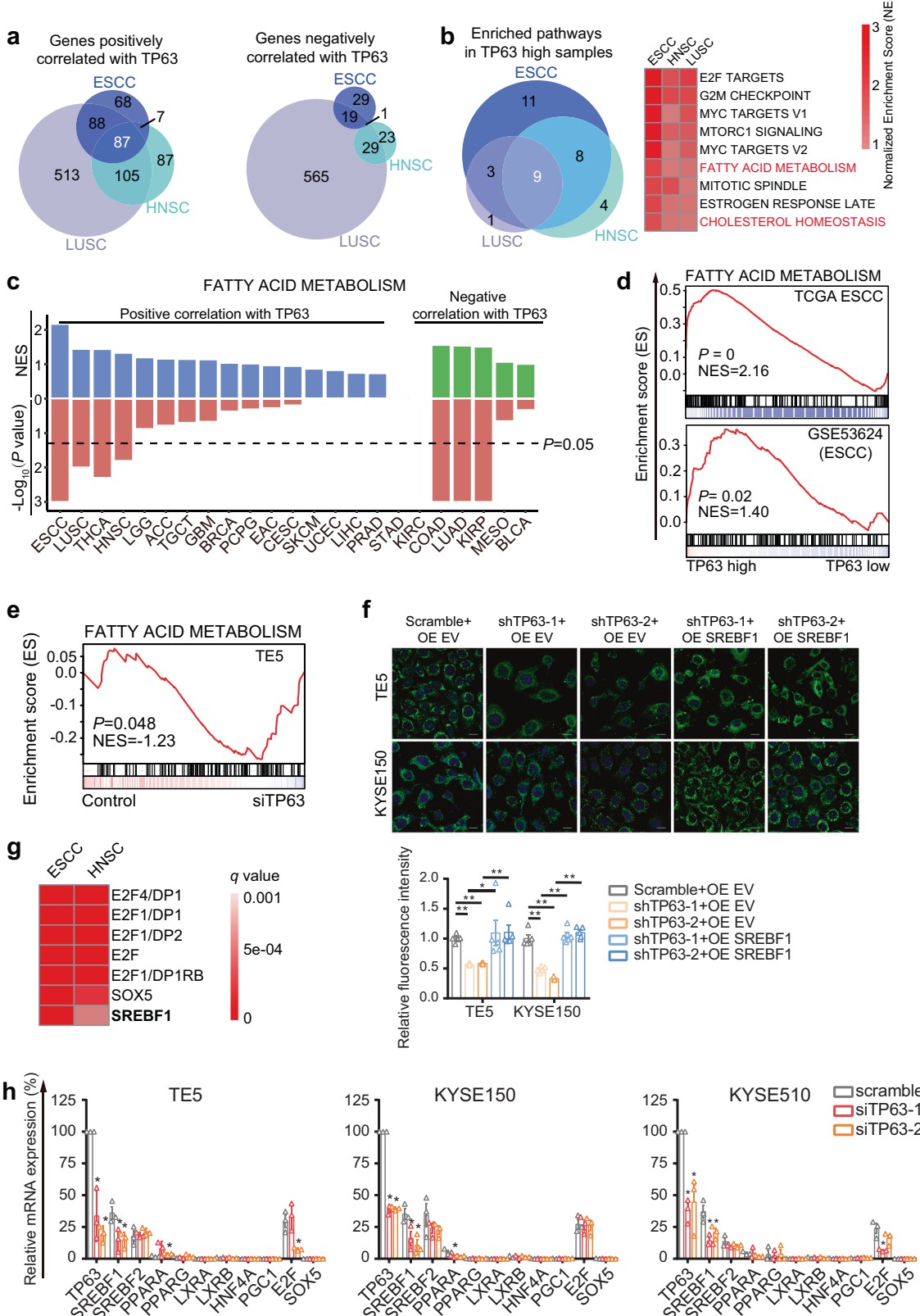

(GSE4573); no such independent HNSC cohort was available for this reanalysis. Fatty-acid metabolism was again enriched in TP63-high samples from either ESCC (Fig. 1d) or LUSC (Supplementary Fig. 1b) cohorts. Among the three types of SCC, ESCC showed the highest enrichment score and lowest P-value (Fig. 1c); we therefore selected this cancer type for further characterization. Notably, GSEA of RNA-Seq data from in vitro perturbation assays showed that fatty-acid metabolism was significantly downregulated in TP63-knockdown ESCC cells (Fig. 1e and Supplementary Fig. 1c). Moreover, depletion of TP63 consistently reduced total lipid droplet levels, suggesting decreased lipid storage (Fig. 1f).

**Fig. 1 TP63 regulates fatty-acid metabolism pathway through SREBF1 in SCCs. a** Venn diagrams of genes exhibiting either positive or negative correlation with TP63 across three types of SCCs ($|Log_2FC| > 2$, $q$-value < 0.05). **b** Left panel, Venn diagram of significantly enriched hallmark pathways in TP63-high SCC samples. Right panel, heatmap of GSEA results of the nine overlapped enriched pathways. **c** Bar plots showing the NES (normalized enrichment score, upper) and $P$-value (lower) from GSEA results of fatty-acid metabolism pathway in TP63-high samples from 23 types of cancers from TCGA. **d** Individual GSEA plots of fatty-acid metabolism pathway in two independent cohorts of ESCC samples (TCGA and GSE53624). **e** GSEA plot of fatty-acid metabolism pathway in RNA-Seq data upon silencing of TP63 in TE5 cells. NES normalized enrichment score. $P$-values in panel **c–e** were adjusted for multiple comparisons. **f** Confocal images of staining of lipid droplets after either TP63 knockdown alone or combined with full-length SREBF1 overexpression in TE5 and KYSE150 cells. Scale bar, 50 μm. Bottom panel, quantitative analysis of lipid droplet staining based on the confocal images; Mean ± SEM are shown, $n = 5$ except for the group of KYSE150-shTP63-1+OE SREBF1 ($n = 6$), as the number of microscopic vision. \*$P$ < 0.05; \*\*$P$ < 0.01; $P$-values were determined by a two-sided $t$-test. **g** Heatmap showing the enriched motifs ($q$ value < 0.001) in promoter regions of genes positively correlated with TP63. **h** Relative mRNA levels of indicated regulators of fatty-acid metabolism following siRNA knockdown of TP63 in TE5, KYSE150, and KYSE510 cells. Mean ± SEM are shown, $n = 3$ (biological replicates). \*$P$ < 0.05; $P$-values were determined by a two-sided $t$-test.

To understand how TP63 regulates fatty-acid metabolism, TF-binding motif enrichment analysis was performed using promoters of genes exhibiting positive correlations with TP63 (Fig. 1g). Consistent with pathway enrichment results, TFs of the E2F family (E2F1 and E2F4) were top-ranked. SOX5 was also highly ranked, likely because it recognizes an identical sequence motif as SOX2, a well-known TP63 partner in SCC[30,33,41]. Notably, SREBF1 was among the highest-ranking TFs, indicating that SREBF1 is involved in the regulation of a significant fraction of genes positively correlated with TP63. Since SREBF1 is one of the most important master regulators of fatty-acid metabolism, we postulated that SREBF1 might act as the chief mediator between TP63 and the fatty-acid metabolism pathway. Supporting this hypothesis, silencing of TP63 downregulated SREBF1 expression in ESCC cell lines (Fig. 1h). Moreover, ectopic expression of SREBF1 reversed the reduction of lipid droplet content caused by TP63-depletion (Fig. 1f) and conversely, silencing of SREBF1 abolished the effect of the overexpression of TP63 (Supplementary Fig. 2a), suggesting that SREBF1 functionally mediates the effect of TP63 on fatty-acid metabolism. Furthermore, knockdown of TP63 inhibited the mRNA expression of key rate-limiting enzymes for fatty-acid synthesis. Importantly, overexpression of SREBF1 rescued the decreased expression of these genes (Supplementary Fig. 2b). On the other hand, knockdown of SREBF1 largely and consistently reversed the effect of TP63 overexpression on the mRNA levels of the enzymes (Supplementary Fig. 2c). To explore whether additional regulators of lipid metabolism were involved, we screened eight known regulators of lipid metabolism (SREBF1, SREBF2, LXRA, LXRB, PPARA, PPARG, PGC1, and HNF4A)[11,42–46]. Most of these TFs were barely expressed in ESCC cells, while only SREBF1 was consistently regulated by TP63 (Fig. 1h and Supplementary Fig. 2d). Moreover, only SREBF1 was moderately upregulated in tumors vs. their corresponding normal tissues ($Log_2FC > 0.5$) across all three types of SCC (Supplementary Fig. 2d).

**SREBF1, TP63, and KLF5 co-regulate the transcription of each other.** We next investigated the regulatory mechanism of TP63 on SREBF1 transcription in SCC, focusing on ESCC model because compared with HNSC and LUSC, ESCC not only showed the highest expression of SREBF1 (Supplementary Fig. 2e), but also had the highest tumor/normal ratio (that is, the greatest upregulation in tumors; Supplementary Fig. 2d). Interestingly, a top-ranking super-enhancer (No. 2 among 1157 super-enhancers) was found to flank the *SREBF1* locus in TE5 cells (Fig. 2a). In fact, SREBF1 had super-enhancers in 8/8 ESCC cell lines (TE5 and KYSE150 cells are shown as examples in Fig. 2b). To determine whether this super-enhancer region indeed regulates SREBF1 transcription, circularized chromosome conformation capture (4C) assays were performed in TE5 cells using the

*SREBF1* promoter as bait (Fig. 2c). Importantly, 4C assays identified complex, extensive interactions between the *SREBF1* promoter and the super-enhancer region (Fig. 2c, Supplementary Data 2). Moreover, these DNA–DNA contacts were strictly confined within the super-enhancer region, highlighting the specificity of chromatin interactions. Notably, TP63 ChIP-Seq data from the same TE5 cells revealed multiple binding peaks within both the *SREBF1* super-enhancer and the promoter (Fig. 2d), suggesting direct transcriptional regulation by TP63. Because we recently demonstrated that in SCC cells, TP63 often cooperates with SOX2 and KLF5 to co-regulate hundreds of enhancers and super-enhancers[33], we hypothesized that TP63 might also require either SOX2 or KLF5 to co-regulate transcription of SREBF1. KLF5- (but not SOX2)-binding peaks were identified within both the *SREBF1* super-enhancer and the promoter in TE5 cells (a positive control peak of SOX2 ChIP-Seq was provided in Supplementary Fig. 3a). Consistently, knockdown of KLF5 (but not SOX2) inhibited expression of SREBF1 across ESCC cell lines at both transcriptional and protein levels (Fig. 2e and Supplementary Fig. 3b). Additionally, verifying our recent findings on co-regulation between KLF5 and TP63, silencing either one of these two TFs decreased the expression of the other (Fig. 2e, f).

To further characterize the activity of the SREBF1 super-enhancer, we intersected 4C, H3K27Ac, and TP63/KLF5 ChIP-Seq data[30,33]. Three putative enhancer constituents were identified to have 4C contacts as well as the peaks of H3K27Ac, TP63, and KLF5. Subsequently, these three enhancer elements and *SREBF1* promoter region were individually cloned into luciferase reporter vectors: robust activities of all three constituent enhancers were confirmed in TE5 cells (Fig. 3a, b). As anticipated, the SREBF1 promoter also harbored strong reporter activity (Fig. 3a, b). Furthermore, silencing of either TP63 or KLF5 decreased the reporter activity of the three enhancers and the promoter (Fig. 3a, b). To measure the direct regulatory effect of TP63 on its targeting enhancers, we selected enhancer E3 and performed site-directed mutagenesis of TP63 canonical binding motif. Indeed, overexpression of TP63 increased significantly the reporter activity of the wild-type enhancer, but produced no detectable effect on the mutant enhancer (Supplementary Fig. 4a–c). To further verify the direct transcriptional regulation of this super-enhancer on SREBF1, a CRISPR interference (CRISPRi) system was utilized, wherein a single guide RNA (sgRNA) directs the complex of dCas9/KRAB (fusion of nuclease-inactive dCas9 to the Krüppel-associated box repressor) to suppress targeted cis-regulatory elements[47]. sgRNAs against either E1, E2, E3, or the promoter were designed and individually transfected into TE5 cells together with dCas9/KRAB (Fig. 3c). To exclude off-target effects, two independent sgRNAs were designed for each element. Importantly, sgRNAs targeting either E1, E2, E3 or the promoter each potently and consistently reduced SREBF1

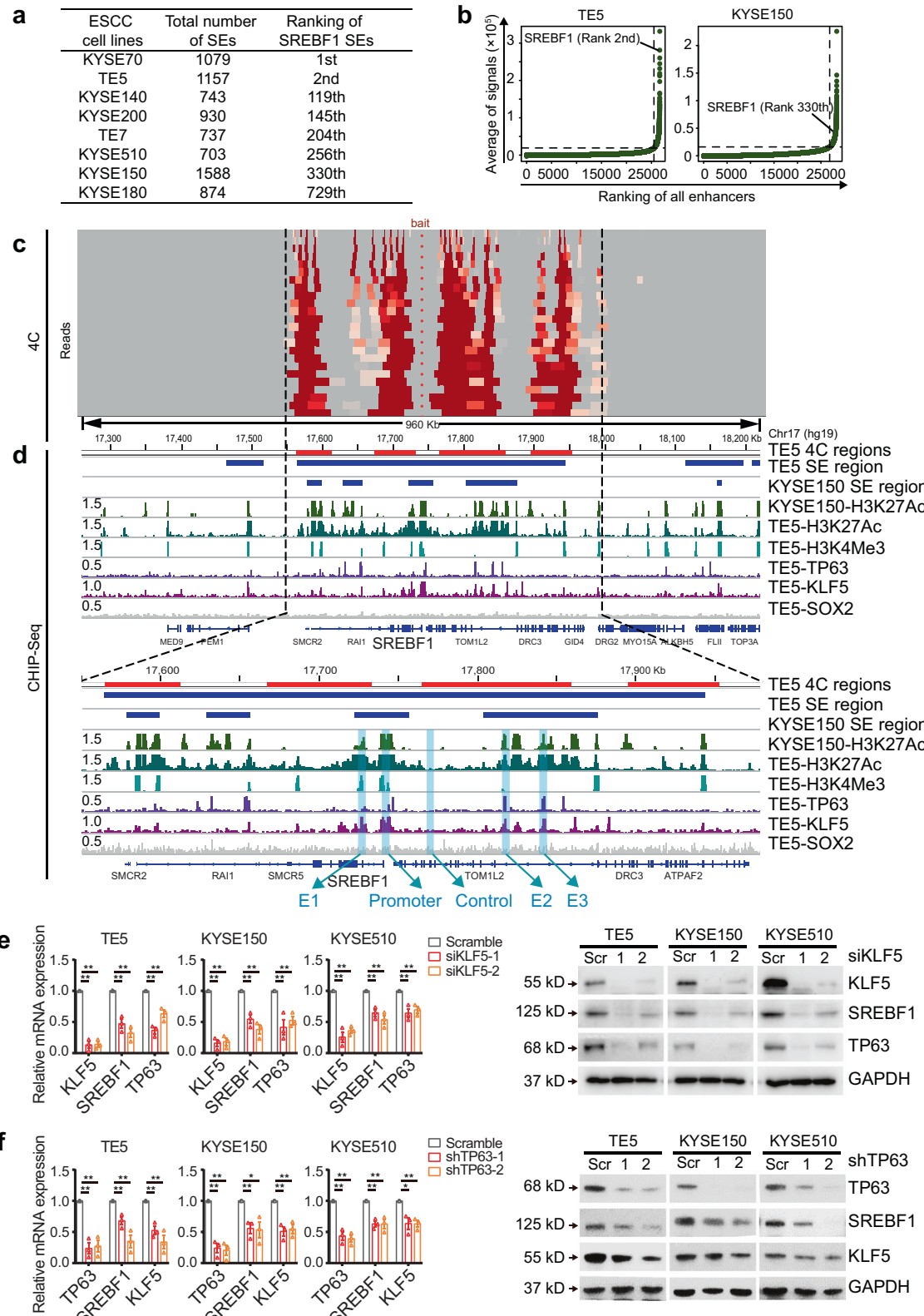

**Fig. 2 TP63 and KLF5 co-regulate SREBF1 in ESCC cells. a** Ranking of SREBF1 super-enhancers (SEs) in eight ESCC cell lines. **b** Inflection plots ranking all typical- and super-enhancers in TE5 and KYSE150 cells. **c** 4C assay showing the long-range interactions anchored on SREBF1 promoter in TE5 cells. Deeper red color indicates higher interaction frequency. **d** IGV plots showing ChIP-Seq profiles of indicated factors in SREBF1 gene locus. Blue shadows highlighting selected constitutive enhancers (E1, E2, and E3), promoter and one negative control (Control) region. 4C-postive regions are depicted as red bars and super-enhancer (SE) regions are depicted as blue bars on top of the IGV plots. RPM (Reads per million mapped reads) values of peaks are on the left of the tracks. **e, f** qRT-PCR and western blotting assays upon knockdown of either **e** KLF5 or **f** TP63 in ESCC cells. Mean ± SEM are shown, $n = 3$ (biological replicates). *$P < 0.05$; **$P < 0.01$; $P$-values were determined by a two-sided $t$-test. Full-length SREBF1 is shown.

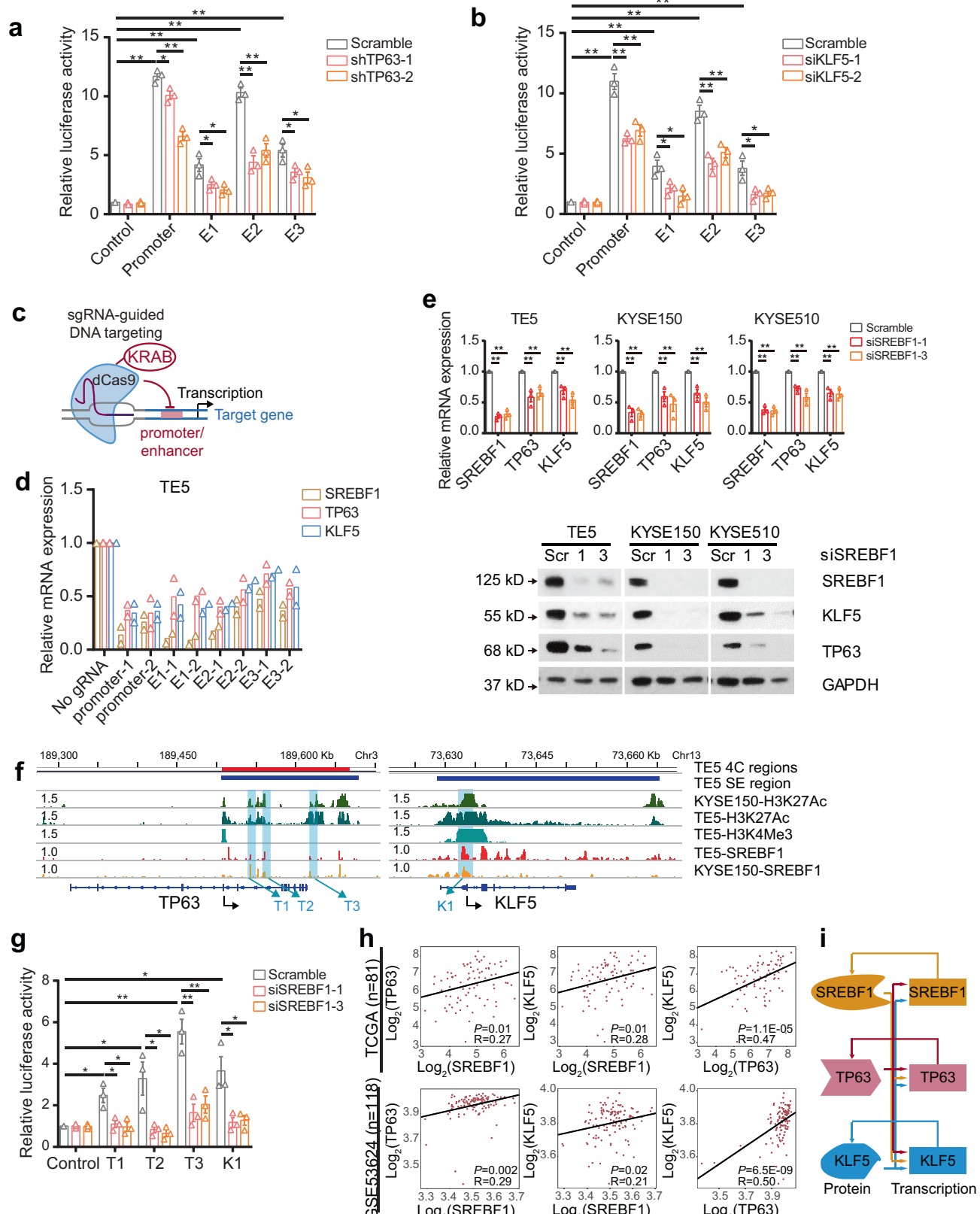

expression (Fig. 3d), highlighting the prominent regulatory activity of these constitutive enhancers.

Quite surprisingly, sgRNAs targeting either E1, E2, E3 or the promoter also significantly and consistently reduced expression levels of both TP63 and KLF5, which are upstream regulators of SREBF1 (Fig. 3d). This intriguing effect suggests that SREBF1

may also regulate the transcription of TP63 and KLF5, forming a feedback co-regulatory loop. Indeed, knockdown of SREBF1 significantly reduced the expression of TP63 and KLF5 at both mRNA and protein levels across different ESCC cell lines (Fig. 3e). We next explored how SREBF1 controls the transcription of both TP63 and KLF5, and performed SREBF1 ChIP-Seq in

**Fig. 3 SREBF1 regulates TP63 and KLF5 in ESCC cells. a, b** Enhancer activity measured by luciferase reporter assays after either **a** TP63 or **b** KLF5 knockdown in TE5 cells. Mean ± SEM are shown, $n = 3$ (biological replicates). *$P < 0.05$; **$P < 0.01$; $P$-values were determined by a two-sided $t$-test. **c** Schematic showing inhibition of constitutive enhancers or promoter of SREBF1 using a catalytically-dead dCas9 fused to a transcriptional repressor domain (KRAB). **d** qRT-PCR measuring mRNA levels of SREBF1, TP63, and KLF5 after transfection of either dCas9/Krab vector alone (No sgRNA) or together with siRNAs for E1, E2, E3, promoter in TE5 cells. Mean values are shown, $n = 2$ (biological replicates). **e** qRT-PCR and western blotting analyses upon knockdown of SREBF1 in ESCC cells. Mean ± SEM are shown, $n = 3$ (biological replicates). **$P < 0.01$; $P$-values were determined by a two-sided $t$-test. **f** IGV plots of ChIP-Seq profiles of indicated factors at either TP63 or KLF5 gene loci. Blue shadows highlighting selected constitutive enhancers of TP63 (T1, T2, and T3) and promoter of KLF5 (K1) occupied by SREBF1. 4C-positive regions (using TP63 promoter as the bait) are depicted as red bars and super-enhancer (SE) regions are depicted as blue bars. RPM (Reads per million mapped reads) values of peaks are on the left of the tracks. **g** Luciferase reporter assays after SREBF1 knockdown in TE5 cells. Mean ± SEM are shown, $n = 3$ (biological replicates). *$P < 0.05$; **$P < 0.01$; $P$-values were determined by a two-sided $t$-test. **h** Pearson correlation coefficient between SREBF1, TP63, and KLF5 in TCGA ESCC cohort ($n = 81$) and GSE53624 cohort ($n = 118$). **i** Schematic graph of the regulatory relationship between SREBF1, TP63, and KLF5.

both TE5 and KYSE150 cell lines (Fig. 3f). As expected, SREBF1-binding peaks strongly overlapped in these two ESCC cell lines ($P = 10^{-8}$, Supplementary Fig. 4d). Sequence motif analysis confirmed that SREBF1 motif was the most strongly enriched (Supplementary Fig. 4e), and canonical SREBF1 target genes (Supplementary Fig. 4f) all harbored SREBF1 peaks at their promoters. These data validate the quality of SREBF1 ChIP-Seq results. Importantly, in the genome locus of TP63, we identified multiple SREBF1-binding peaks in a super-enhancer for TP63 (Fig. 3f). Notably, our recent 4C results identified direct contacts of this SREBF1-occupying super-enhancer with the TP63 promoter[33]. In the case of KLF5 locus, we also noted several SREBF1 peaks at both KLF5 promoter and enhancers. We next selected several candidate enhancer elements (T1, T2, and T3) of TP63 and a promoter element (K1) of KLF5 for luciferase reporter assays. Robust reporter activities of T3 and K1 were detected, and they were reduced upon silencing of SREBF1 (Fig. 3g). We further performed site-directed mutagenesis to mutate SREBF1-binding motif in T2 and T3 elements. We found that overexpression of SREBF1 consistently increased the reporter activity of both wild-type T2/T3 elements, but failed to affect mutant T2/T3 (Supplementary Fig. 5a–d), confirming the direct regulation of SREBF1 on these enhancers. Moreover, RNA-Seq data from different ESCC patient cohorts showed that the expression levels of the three TFs were modestly correlated with each other (Fig. 3h and Supplementary Fig. 5e). Together, these results demonstrate that SREBF1, TP63, and KLF5 co-activate the transcription of each other, forming a co-regulatory feedback loop (Fig. 3i).

**SREBF1 promotes biosynthesis of fatty acids, sphingolipids, and glycerophospholipid in SCCs.** Having established the molecular basis of the super-enhancer activation of SREBF1, as well as the co-regulatory feedback loop of SREBF1/TP63/KLF5 in SCC cells, we next investigated the functional impact of SREBF1 on fatty-acid metabolism identified earlier (Fig. 1f). SREBF1 is known to activate rate-limiting enzymes for de novo fatty-acid biosynthesis (ACLY, FASN, ACSS2 and SCD) in different cell types[12,15,48–50]. Here in SCC cells, we verified that silencing of SREBF1 by siRNAs (Fig. 4a, b, and Supplementary Fig. 6a, b) markedly decreased both the mRNA and protein expression of these enzymes. Inhibition of SREBF1 activity by its antagonist, Fatostatin[16,51] (which also targets SREBF2), also produced the same effect (Fig. 4c and Supplementary Fig. 6c). Moreover, lipid droplet content reduced after silencing of SREBF1 (Supplementary Fig. 6d, e). We next comprehensively profiled the landscape of lipid species in SCC cells considering that the lipidome has enormous structure complexity. Specifically, LC-MS/MS-based lipidomics was performed in the presence and absence of SREBF1-inhibition (by Fatostatin) in KYSE510 cells. To ensure reproducibility, triplicates of both control and experimental

samples were profiled, which exhibited high correlation within each group (Pearson correlation coefficient: 0.82–0.95, Supplementary Fig. 6f). As a result, a total of 1561 lipid ions were identified, which belonged to 35 classes of lipids, suggesting high coverage of lipidome by this systematic approach (Supplementary Data 3). SREBF1-inhibition resulted in the downregulation of 69 and upregulation of 65 lipid ions ($|Log_2FC| > 2$, $q$-value $< 0.1$, Fig. 4d, e). The downregulated lipid ions were notably enriched in two lipid classes: (i) Sphingolipid family (SL), which included Ceramide (Cer), Sphingosine (So), Glucosylceramide (CerG1), Diglucosylceramide (CerG2), and Triglucosylceramide (CerG3) and (ii) Glycerophospholipid family (GPL), which included Cardiolipin (CL), Lipopolysaccharide (LPS), Phosphatidylserine (PS), Phosphatidic acid (PA), Phosphatidylglycerol (PG), and Lyso-phosphatidylethanolamine (LPE). In contrast, no discernable pattern was found in the upregulated lipid ions (Fig. 4d, e). These results suggest that SREBF1 regulates lipid metabolism in addition to fatty-acid synthesis. To further understand the alterations in the lipidome upon SREBF1-inhibition, we integrated RNA-Seq following SREBF1-inhibition (Supplementary Table 1) and its ChIP-Seq, and identified direct SREBF1-targeted enzymes for lipid metabolism process. Indeed, consistent with lipidomic data, SL anabolic enzymes, such as SPTLC1, SPTLC2, ELOV4, ELOV6, ELOV7, and CERS6, were direct downstream targets of SREBF1 (Fig. 4f). LIPIN1, an enzyme for GPL synthesis, was also under direct control of SREBF1 (Fig. 4f). Either knockdown or inhibition of SREBF1 downregulated the expression of all of these factors. Together, these findings demonstrate that SREBF1 promotes the biosynthesis of fatty acid, SL, and GPL by directly activating the transcription of many chief enzymes in SCC cells.

**SREBF1 is essential for cell growth and migration of SCC.** We next sought to determine the biological significance of SREBF1 in SCC, again using ESCC as a primary disease model. As anticipated, immunohistochemistry (IHC) staining showed that SREBF1 protein was significantly more abundant in the nucleus than cytoplasm (Fig. 5a, b). Overexpression of nuclear SREBF1 protein in ESCC was confirmed in 179 cancerous esophageal samples and 57 matched adjacent nonmalignant esophageal tissues (Fig. 5a, right panel). Importantly, high-expression of SREBF1 was significantly associated with poor overall survival of ESCC patients (Fig. 5c). Regression analysis identified SREBF1 as an independent survival predictor in a multivariable model comprising the clinical tumor-node-metastasis (TNM) parameters (HR = 2.24, 95% CI = 1.36–3.71, $P = 0.002$) (Supplementary Table 2). Given the notable co-regulatory feedback loop of SREBF1/TP63/KLF5 identified earlier, proteins of TP63 and KLF5 were also stained using the same cohort of ESCC samples. Consistently, significant overexpression of TP63 and KLF5 were verified (Fig. 5a, b).

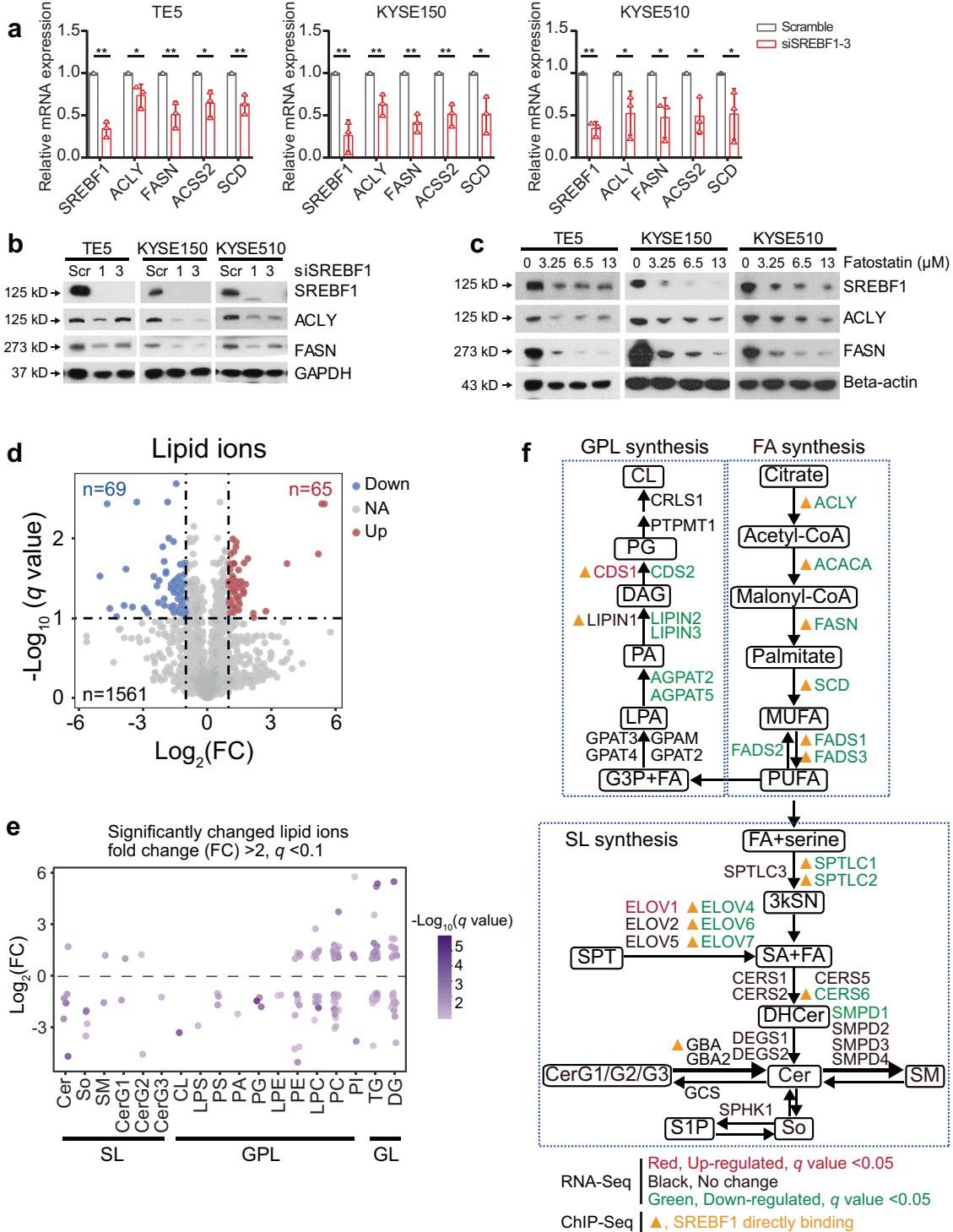

**Fig. 4 SREBF1 promotes biosynthesis of fatty acids, SL, and GPL in SCCs. a** qRT-PCR measuring mRNA levels of central enzymes for fatty-acid synthesis upon knockdown of SREBF1 in ESCC cells. Mean ± SEM are shown, $n = 3$ (biological replicates). $*P < 0.05$; $**P < 0.01$; $P$-values were determined by a two-sided $t$-test. **b, c** Western blotting analyses showing protein levels of central enzymes for fatty-acid synthesis upon either **b** knockdown of SREBF1 or **c** treatment with Fatostatin (0, 3.25 μM, 6.5 μM, 13 μM) in ESCC cells. The western blotting experiments were performed in three biologically independent replicates, and the representative result were shown here. **d** Volcano plot of LC-MS/MS-based lipidomics after treatment with Fatostatin in KYSE510 cells. Each dot is one lipid ion. **e** Scatter plot of significantly changed lipid ions, which were grouped by lipid classes. Each dot is one lipid ion. SL sphingolipids, GPL glycerophospholipids, GL glycerolipids. **f** Schematic diagram showing the regulation of lipid synthesis pathways by SREBF1 via integration of RNA-Seq (SREBF1 knockdown vs scramble) and ChIP-Seq in TE5 cells. FA fatty acid, SL sphingolipid, GPL glycerophospholipids, MUFA monounsaturated FAs, PUFA polyunsaturated FAs, 3kSN 3-keto-sphinganine, SA sphinganine, SPT serine palmitoyltransferase, DHCer dihydroceramide, CER ceramide, CerG1 glucosylceramide, CerG2 giglucosylceramide, CerG3 triglucosylceramide, SM sphingomyelin, SO Sphingosine, S1P sphingosine 1 phosphate, G3P glycerol 3-phosphate, LPA lysophosphatidic acid, PA phosphatidic acid, DAG diacylglyceride, PG phosphatidylglycerol, CL cardiolipin.

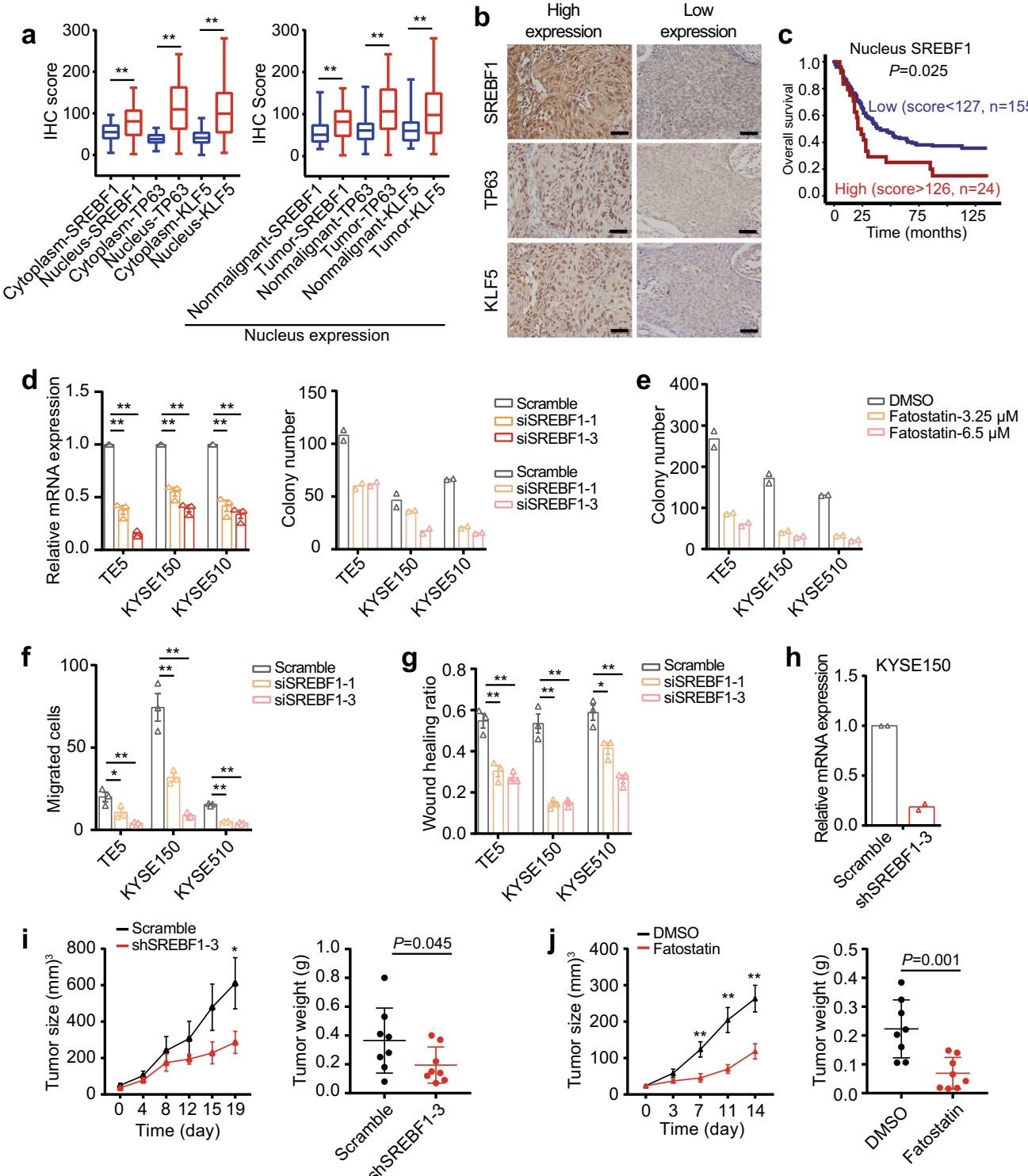

**Fig. 5 SREBF1 is essential for SCC cell growth and migration. a** Boxplots of IHC scores for nucleus and cytoplasmic staining of SREBF1, TP63, and KLF5 proteins in ESCC tumor and adjacent nonmalignant esophagus samples. Boxplots indicate the median (middle line), 25th, 75th percentile (box) and 5th and 95th percentile (whiskers). The numbers of samples for IHC staining are 57 (the nonmalignant) and 179 (ESCC tumors). **P < 0.01; P-values were determined by a two-sided t-test between the IHC scores of two groups. **b** Representative IHC images of SREBF1, TP63, and KLF5 proteins in ESCC tumor samples. Original magnification is ×400; scale bar is 50 μm. **c** Kaplan–Meier analyses of ESCC patient survival stratified by the protein expression of SREBF1. A Log-rank test was used for Kaplan–Meier curve, and P-value was two-tailed and significance level was 0.05. **d** Knockdown of SREBF1 by individual siRNAs and **e** treatment of Fatostatin inhibited cell proliferation in colony formation assay. Mean values are shown, three biological replicates for qRT-PCR assays and two biological replicates for colony assays. **f** Migration assay and **g** wound healing assay after knockdown of SREBF1 in ESCC cells. Mean ± SEM are shown, n = 3 (biological replicates). *P < 0.05; **P < 0.01; P-values were determined by a two-sided t-test. **h** SREBF1 was stably silenced by shRNA in KYSE150 cell line. **i** Mouse xenograft assay upon knockdown of SREBF1 by shRNA or **j** treatment of Fatostatin (30 mg/kg/day) by intraperitoneal injection. Mean ± SEM are shown, n = 8 (the numbers of tumor in one group). *P < 0.05; **P < 0.01; P-values were determined by a one-sided t-test.

The biological effects of SREBF1 on the proliferation and migration of ESCC cells were next investigated. Silencing of endogenous SREBF1 expression by independent siRNAs markedly reduced colony growth (Fig. 5d) across different ESCC cells. Similarly, inhibition of SREBF1 activity by Fatostatin also significantly suppressed colony formation (Fig. 5e). In addition, knockdown of SREBF1 markedly inhibited short-term migration capacity of ESCC cells in both wound healing and trans-well migration assays (Fig. 5f–g). In a xenograft model, targeting SREBF1 by either shRNA or its inhibitor Fatostatin potently suppressed xenograft growth in vivo (Fig. 5h–j, Supplementary Fig. 7a, b left panels) and targeting of SREBF1 using either shRNA or Fatostatin potently reduced the expression of KI67 (Supplementary Fig. 7c, d). In addition, we validated (i) the co-regulatory feedback loop between SREBF1/TP63/KLF5, and (ii) canonical target genes of SREBF1 in fatty-acid metabolism (e.g., ACLY, FASN, and SCD) in xenograft tumor samples (Supplementary Fig. 7a, b right panels). These data together characterize SREBF1 as a strong SCC-promoting factor.

**SREBF1, TP63, and KLF5 cooperatively regulate SCC cell transcriptome**. Our above findings of SREBF1 as a strong SCC-promoting factor, as well as its co-regulatory feedback loop with TP63/KLF5 strongly suggest cell-type-specific functions of SREBF1 in SCC, in addition to its canonical role in lipid metabolism pathway. To explore this hypothesis, we analyzed the genome-wide occupancy of SREBF1 in three different cell types, namely SCC (TE5 and KYSE150 cell lines), liver cancer (HepG2 cell line) and breast cancer (MCF7 cell line). SREBF1 ChIP-Seq data of HepG2 and MCF7 cells were retrieved from the ENCODE project, and were reprocessed together with our internal data from ESCC cells using the same computational pipeline. Notably, cell-type-specific SREBF1-binding peaks outnumbered either shared (defined as peaks identified in two cell types) or ubiquitous peaks (defined as peaks identified in all three cell types) (Fig. 6a, Supplementary Fig. 8). Representative genes with either ubiquitous or cell-type-specific peaks are shown in Fig. 6c. To identify putative binding TFs, sequence motif enrichment analysis was performed for each group of peaks (Fig. 6b). Considering the distinct DNA sequence contents between promoter and distal regions (e.g., promoters are often CG rich, while distal enhancer regions are depleted of CpG islands[52]), motif analyses were performed separately in these two genomic contexts, and top 15 most significantly enriched motifs were focused (Supplementary Data 4). As anticipated, regardless of either ubiquitous or specific peak sets, SREBF or NFY (a known SREBF1 co-factor) motifs were almost always top ranked. Notably, known cell-type-specific TF motifs were enriched in cell-type-specific peak sets. In particular, TP63 and HNF4A motifs were significantly and uniquely enriched in ESCC- and HepG2-specific peaks, respectively (Fig. 6b), congruent with their well-defined cell-type-specific functions. KLF5 was also enriched in both ESCC- and HepG2-specific peaks, consistent with its important role in both squamous and GI cancers[53–55]. In contrast, we did not identify such cell-type-specific TF motifs in the ubiquitous peak set.

Focusing on the 473 ESCC-specific SREBF1-binding peaks, because TP63 and KLF5 motifs were strongly enriched, we next analyzed ChIP-Seq data of SREBF1, TP63, and KLF5 generated in the same TE5 cell line. Indeed, validating motif enrichment results, trio-occupancy of SREBF1/TP63/KLF5 was observed in 57.8% (274/473) peaks, dual-occupancy of either SREBF1/TP63 or SREBF1/KLF5 occurred in 14.0% (66/473) or 14.8% (70/473) peaks, and a minor fraction of peaks (63/473, 13.4%) were solo-occupied by SREBF1 alone (Fig. 6d, e). The majority of these peaks were flanked by strong H3K27Ac signals, indicative of

transcription-promoting activity. Meta-gene analysis showed that the binding peaks of these three TFs strongly aligned (Fig. 6e). To understand the transcriptional impact of the co-binding of SREBF1/TP63/KLF5 in these ESCC-specific peaks, RNA-Seq data upon knockdown of each TF in TE5 cells were interrogated. Importantly, GSEA showed that the corresponding transcripts of these 274 trio-occupied ESCC-specific peaks were significantly enriched in the downregulated genes following silencing of either of the three TFs (Fig. 6f). Collectively, these findings demonstrate the prominent co-binding pattern of SREBF1/TP63/KLF5 in hundreds of ESCC-specific peaks, which translates to co-operative regulation of gene expression by the three TFs.

**SREBF1, TP63, and KLF5 cooperate to activate transcriptionally ErbB/mTOR signaling pathways specifically in ESCC cells**. To explore the biological functions of both ubiquitous and cell-type-specific SREBF1-binding peaks, KEGG pathway analysis was performed. In genes associated with the ubiquitous peak set ($n = 120$), seven signaling pathways were significantly enriched. As expected, four pathways belonged to lipid metabolism process, including fatty-acid metabolism, steroid biosynthesis, fatty-acid biosynthesis and biosynthesis of unsaturated fatty acids (Fig. 7a). These data confirm the common function of SREBF1 in regulating lipid and fatty-acid synthesis, regardless of cell types. In stark contrast, of the 13 signaling pathways significantly enriched in genes associated with ESCC-specific trio-occupied peaks, none were related to lipid metabolism. Instead, the majority of these 13 pathways were notably cancer-related, including ErbB signaling pathway, mTOR signaling pathway, glioma, HIF-1 signaling pathway, non-small cell lung cancer, breast cancer, choline metabolism in cancer as well as EGFR tyrosine kinase inhibitor resistance (Fig. 7b). Because ErbB and mTOR signaling pathways were highly ranked, these two oncogenic cascades were selected for further investigation. Consistent with the trio-occupancy of SREBF1/TP63/KLF5 on these ESCC-specific peaks (Fig. 6d), all 13 genes enriched in these two pathways were trio-occupied by SREBF1/TP63/KLF5 (Fig. 7c). A trio-binding peak at the promoter of WNT9A gene is shown as an example, which is expectedly absent in both HepG2 and MCF7 cells (Fig. 7d). Importantly, knockdown of SREBF1, TP63, and KLF5 co-decreased five components of mTOR signaling pathway across different ESCC cell lines (Fig. 7c, Supplementary Fig. 9). Similarly, in ErbB signaling pathway, silencing of SREBF1, TP63, and KLF5 led to the reduction of seven genes (Fig. 7c, Supplementary Fig. 9). Furthermore, the levels of phospho-mTOR, phospho-p70S6K, and phospho-MEK1/2 markedly decreased after knockdown of SREBF1 in SCC cell lines, validating the regulation of ErbB/mTOR signaling pathways by SREBF1 (Fig. 7e). Taken together, these results highlight that SREBF1/TP63/KLF5 cooperate to activate transcriptionally ErbB/mTOR signaling pathways in SCC cells (Fig. 7f).

**Discussion**

In this work, focusing on the most well-established SCC oncogene, TP63, we performed GSEA across three different types of SCC patient samples and unbiasedly identified fatty-acid metabolism as one of the key pathways regulated by this master TF (Fig. 1b). This strong enrichment was not only validated by in vitro perturbation assays (Fig. 1e), but also was highly SCC-specific in a pan-cancer analysis (Fig. 1c). At the molecular level, sequence motif enrichment analysis identified SREBF1 as the key mediator bridging TP63 and fatty-acid metabolism in SCC (Fig. 1g).

As a master regulator of both lipid and fatty-acid metabolism, the activity of SREBF1 is known to be under precise and intricate

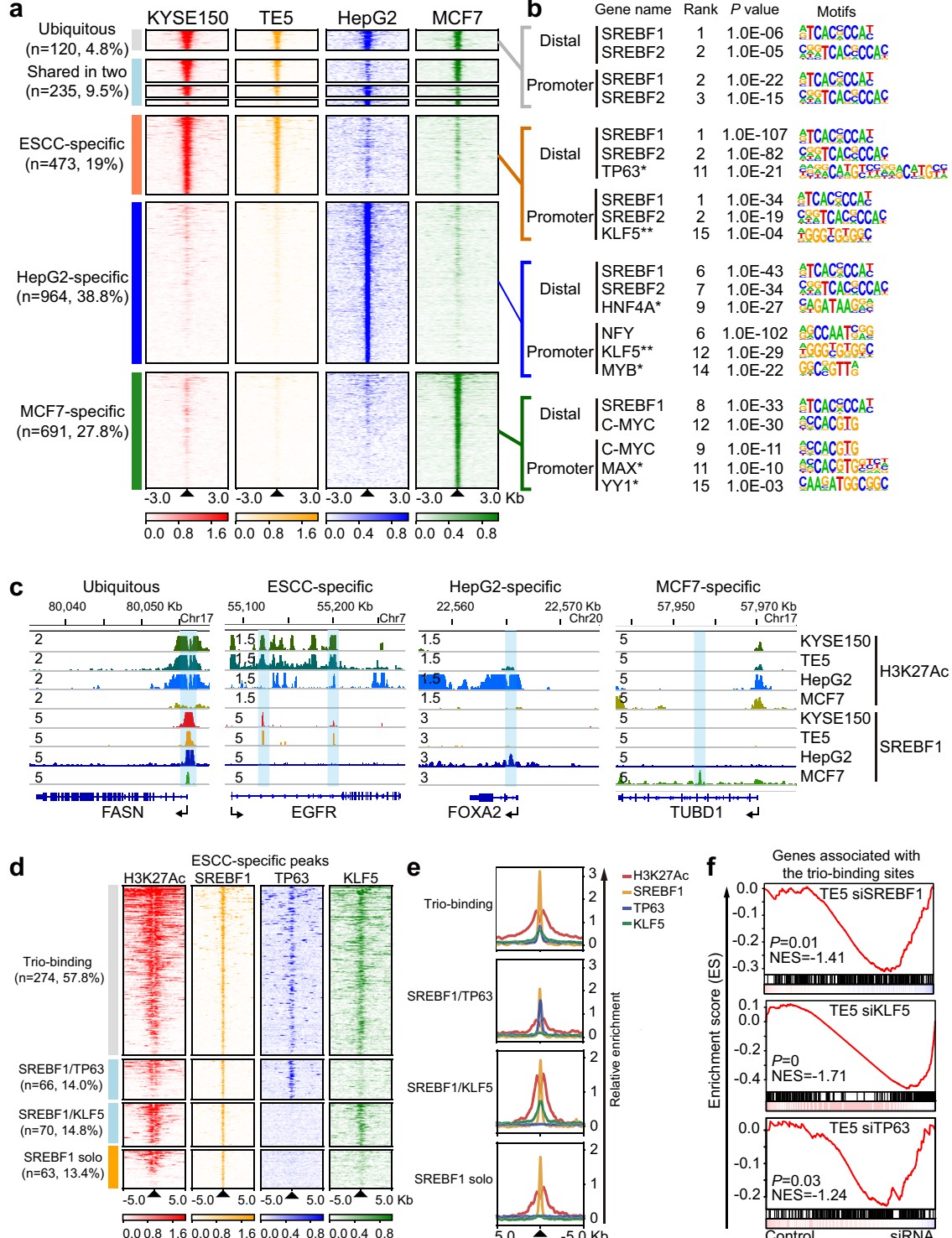

**Fig. 6 SREBF1, TP63, and KLF5 cooperatively regulate SCC cell transcriptome. a** Heatmaps of ChIP-Seq signals at SREBF1 peak regions (±3 Kb of peak center), grouped as ubiquitous, shared or cell-type-specific peak sets, and rank ordered by intensity of SREBF1 peaks based on reads per million mapped reads (RPM). Lines, peaks; color scale of peak intensity is shown at the bottom. **b** Representative top shared or cell-type-specific TF motifs (denoted by *) in each peak sets. Note that KLF5 motif was found in both ESCC and HepG2 promoter regions (denoted by **). *P*-values were adjusted for multiple comparisons. **c** IGV tracks of H3K27Ac and SREBF1 ChIP-Seq profiles at the loci of representative genes from each peak set. **d** Heatmaps of ChIP-Seq signals at ESCC-specific SREBF1 peaks grouped as either trio-, dual-, or solo-occupied by SREBF1/TP63/KLF5 (±5 Kb of peak center), rank ordered by intensity of SREBF1 peaks. **e** Similar as **d**, line plots showing the distribution of indicated ChIP-Seq signals at ESCC-specific SREBF1 peak regions from indicated groups. **f** GSEA plots of the changes of the corresponding transcripts assigned to the 274 ESCC-specific peaks from RNA-Seq upon silencing of either SREBF1, TP63, or KLF5 in TE5 cells. NES normalized enrichment score. *P*-values were adjusted for multiple comparisons.

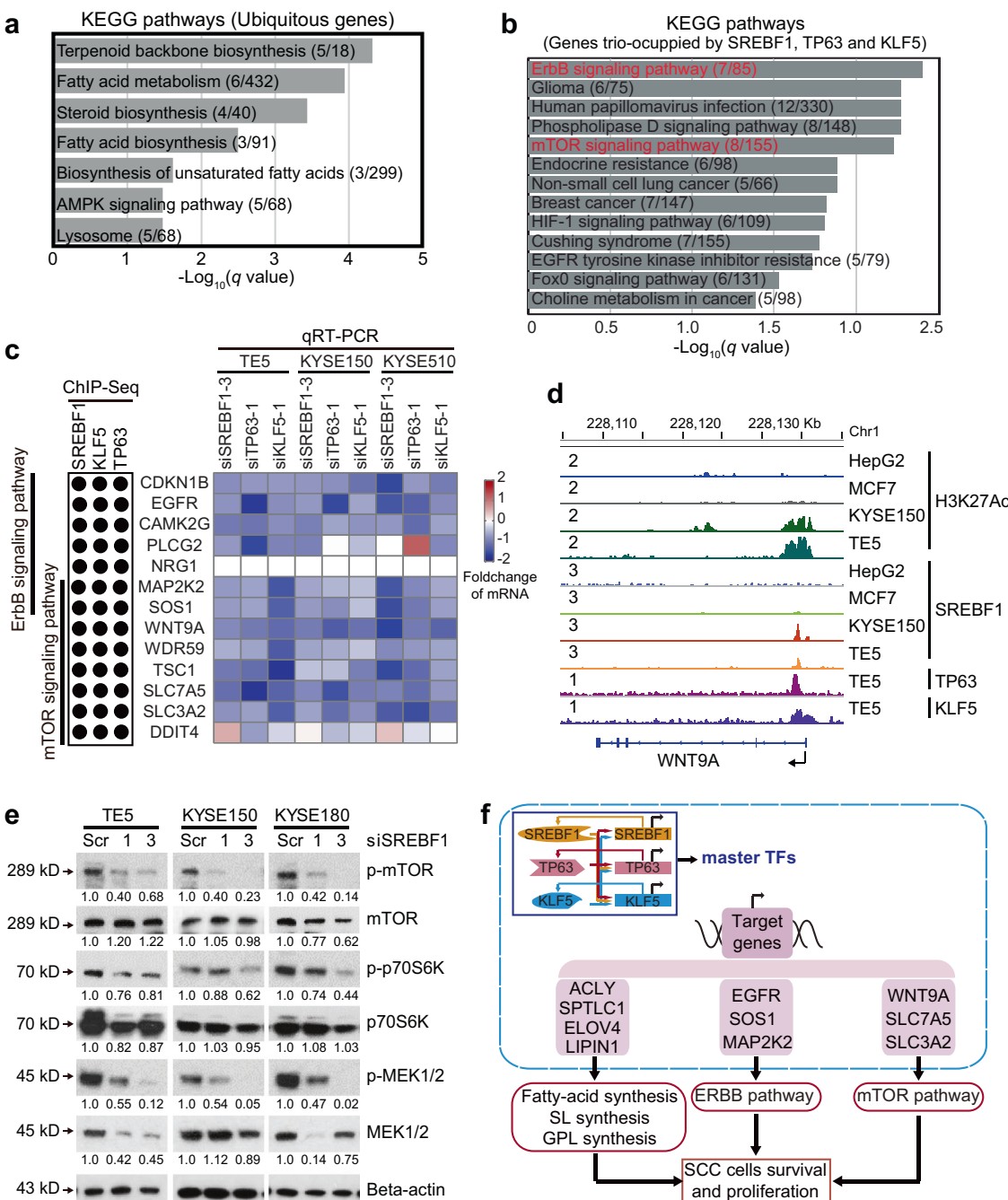

**Fig. 7 SREBF1, TP63, and KLF5 cooperate to transcriptionally activate ErbB/mTOR signaling pathways in ESCC cells. a** Significantly enriched KEGG pathways of the corresponding transcripts assigned to the ubiquitous peaks or **b** the 274 ESCC-specific peaks trio-occupied by SREBF1, TP63, and KLF5. The number of genes enriched/the total number of genes in each pathway are shown in the brackets. **c** Left panel showing genes enriched in ErbB/mTOR pathways, which were occupied by SREBF1/TP63/KLF5 based on ChIP-Seq data. Black, occupied. Right panel: the fold changes (knockdown v.s. scramble control) of mRNA levels of indicated genes upon the silencing of either SREBF1, TP63, or KLF5 in three cell lines. In each group, 30 nM siRNA was transfected; after 48 h, total RNA was extracted for the measurement the mRNA expression by qRT-PCR. **d** IGV tracks of ChIP-Seq profiles of indicated factors at WNT9A gene locus in different cell lines. **e** Western blotting analyses following siRNA knockdown of SREBF1 in ESCC cells. Quantification of the blots is shown below. **f** A model diagram of the regulations and functions of SREBF1 in SCC cells.

regulation. In addition to the feedback regulation by sterols, SREBF1 has been shown to be stabilized and activated by the PI3K/AKT/mTOR signaling cascade. Specifically, AKT stabilizes the SREBF1 nuclear form through downregulation of FBXW7, an E3 ubiquitin ligase, which mediates SREBF1 N-terminus degradation[56–58]. Alternatively, mTORC1 regulates LIPIN1, a phosphatidic acid phosphatase, to control SREBF1 nuclear localization and its transcriptional activity[59,60]. Together, previous

reports on SREBF1 regulation were largely focused on either its protein or post-translational levels, and the epigenomic regulation on SREBF1 has not been extensively investigated. Here, we first noted a prominent (high ranking) super-enhancer region for SREBF1 in SCC cells (Fig. 2a, b). 4C and ChIP-Seq further determined that TP63, together with KLF5, regulated the transcription of SREBF1 by binding to its promoter and super-enhancer in SCC. Moreover, CRISPRi and luciferase reporter

assays identified three enhancer constituents with strong activity to promote the transcription of SREBF1. These findings add another layer to the complex regulatory mechanisms on SREBF1. The epigenomic activation of SREBF1 in SCC cells indeed leads to its overexpression, as confirmed by our IHC staining of SCC patient samples. Moreover, we identified SREBF1 as an independent prognostic marker in SCC, consistent with prior reports in prostate, breast, and hepatocellular cancers[11].

Functionally, our work first confirmed the canonical role of SREBF1 in the regulation of both lipid and fatty-acid metabolism. Specifically, LC-MS/MS-based lipidomics, RNA-Seq, and SREBF1 ChIP-Seq together established that SREBF1 directly targeted a number of rate-limiting enzymes for fatty-acid synthesis. This integrative approach further revealed that in addition to fatty-acid metabolism, SREBF1 also controlled the biosynthesis of SL and GPL in SCCs by regulating key enzymes in these processes. Of note, some of these SREBF1-regulated lipids, such as ceramides, have been shown to have prosurvival or antiapoptotic functions in certain cancer cells[61,62]. Importantly, in addition to its canonical function in lipid metabolic process, our study further established two more intriguing SCC-specific roles of SREBF1: (i) SREBF1, TP63, and KLF5 co-activate the transcription of each other, forming a feedback co-regulatory loop; (ii) SREBF1, TP63, and KLF5 cooperatively regulate the transcription of hundreds of genes specifically in SCC cells, activating cancer-associated signaling pathways such as ErbB and mTOR signaling cascades.

The feedback co-regulatory loop of SREBF1/TP63/KLF5 was initially discovered by CRISPRi experiments, which showed that inhibition of the super-enhancer for SREBF1 quite unexpectedly led to the reduction of expression of both TP63 and KLF5 (Fig. 3d). Further in-depth examination revealed multiple SREBF1-binding peaks on both TP63 and KLF5 loci (Fig. 3f), and luciferase reporter assays validated their regulation on these two TFs. Between SREBF1 and TP63/KLF5, we also observed a strong correlation at the mRNA level, but weak at the nuclear protein level. This weak correlation of the subcellular proteins is not entirely surprising, considering the complex activation and regulation of SREBF1 protein which involves three different subcellular compartments: endoplasmic reticulum, Golgi, and nucleus. Importantly, all of the three proteins were overexpressed in SCC primary samples and were associated with poor survival of SCC patients. These results resemble the "Core transcription regulatory circuitry (CRC)" paradigm, which have been found in many different cell types[63–65], including both normal and cancerous cells. Interestingly, we recently demonstrated that SOX2/TP63/KLF5 constitute such CRC machinery in ESCC cells[33]. However, here we found that SOX2 and SREBF1 did not show co-regulation, indicating that different CRC models may be present in the same cell type, which has been increasingly recognized[65,66].

The other SCC-specific function of SREBF1 uncovered in this work is its binding of ~500 genome loci uniquely in SCC cells. Indeed, genome-wide occupancy analyses showed that SREBF1-binding peaks were notably cell-type-specific (Fig. 6a). Sequence motif analysis identified TP63/KLF5 as putative co-binding partners with SREBF1 on these ~500 ESCC-specific peaks, which was validated by ChIP-Seq data. RNA-Seq results confirmed the co-operative regulation of SREBF1/TP63/KLF5 on the transcription of the genes associated with these ESCC-specific peaks. More importantly, this cell-type-specific occupancy was associated with different biological functions compared with ubiquitous binding. For example, genes assigned to ubiquitous peaks were responsible for the canonical role of SREBF1 in lipid metabolism, whereas transcripts assigned to ESCC-specific peaks were enriched in many cancer-associated signaling pathways, including ErbB and mTOR pathways. Indeed, in-depth

characterization identified 13 genes of these signaling pathways which were trio-occupied by SREBF1/TP63/KLF5 (Fig. 7c). These genes included well-known cancer-promoting factors, such as EGFR, WNT9A, and MAP2K2.

In summary, this study identifies an important feedback co-regulatory loop of SREBF1/TP63/KLF5 in SCC cells. As a master regulator of lipid metabolism, SREBF1 controls the biosynthesis of fatty acids, SL, and GPL in SCCs. Moreover, SREBF1 shows a noncanonical, SCC-specific function by cooperating with TP63/KLF5 to regulate hundreds of cis-regulatory elements across the SCC epigenome, which converge on activating cancer-promoting pathways. Both of the canonical and noncanonical functions of SREBF1 constitute the basis of its prominent biological significance in SCC. Indeed, SREBF1 is essential for SCC viability and migration. These results provide important mechanistic insights into the transcriptional dysregulation in cancer, and discover SREBF1 as a potential therapeutic target and prognosis marker for SCC.

## Methods

**GSEA and KEGG pathway analysis**. RNA-Seq (level-3) data of ESCC ($n = 81$), HNSC (HPV- samples, $n = 436$), and LUSC ($n = 501$) were downloaded from TCGA (released on March 26, 2019; GDC V16.0). Microarray expression data of ESCC (GSE53624 and SRP064894) and LUSC (GSE4573) were downloaded from Gene Expression Omnibus (GEO) database[67–69]. We first ranked tumor samples based on the expression of TP63, and classified the samples into two groups (top and bottom 30% samples). Secondly, differentially expressed genes were determined using limma R package[70]. GSEA Preranked method[71] was performed to identify the Hallmark pathways. ClusterProfiler R package was used to perform KEGG pathway analysis[72].

**Motif enrichment analysis**. HOMER *findMotifsGenome.pl* function was used to identify enriched motifs in selected regions. The parameters in HOMER is hg19 -size 200 -len 8,10,12[73].

**RNA-seq data analysis**. The RNA-seq data of siSREBF1, siTP63, and siKLF5 were generated in TE5 and KYSE150 cell lines. Trim galore was used to remove the adapters. 49 bp single-end and 150 bp paried-end reads were aligned to human reference genome (HG19) using STAR (–alignIntronMin 20–alignIntronMax 1000000 –alignSJoverhangMin 8 –quantMode GeneCounts) method[74]. DESeq2 was used to identify the differentially expressed genes based on the read counts[75]. Data have been deposited to Gene Expression Omnibus (GSE143803).

**Human cancer cell lines**. ESCC cell lines, including TE5 (kindly provided by Dr Koji Kono from Cancer Science Institute of Singapore), KYSE150, KYSE180, and KYSE510 (kindly provided by Dr Y Shimada from Kyoto University) and HNSCC cell line UMSCC1 (kindly provided by Dr Timothy Chan from Memorial Sloan Kettering Cancer Center)[76] were cultured in RPMI-1640, supplemented with 10% fetal bovine serum (FBS) (Omega Scientific, Tarzna, CA), 100 U/ml penicillin, and 100 mg/ml streptomycin. Cells were cultured at 37 °C in a humidified atmosphere containing 5% $CO_2$. All cell lines were verified by short tandem repeat analysis recently.

**RNA extraction, cDNA synthesis, and quantitative real-time PCR**. Total RNA was extracted with RNeasy Mini kit (QIAGEN, 70106) and cDNA was obtained from the total RNA using Maxima™ H Minus cDNA Synthesis Master Mix with dsDNase (Thermo Scientific, M1682). Quantitative real-time PCR (qRT-PCR) was conducted with PowerUp™ SYBR™ Green Master Mix (Thermo Scientific, A25918). Actin was used for normalization. Primers used in the study were listed in Supplementary Table 3.

**Antibodies and reagents**. The following antibodies and reagents were used: Anti-SREBF1 (Proteintech, 14088-1-AP, 1:1000 for western blotting and 4 μg for ChIP), anti-KLF5 (Santa Cruz Biotechnology, sc-398409X, 1:1000 for western blotting, and 4 μg for ChIP), anti-TP63 (R&D Systems, AF1916-SP, 1:1000), anti-Actin (Santa Cruz Biotechnology, sc-8432, 1:2000), anti-ACLY (Cell Signaling Technology, 4332, 1:1000), Anti-FASN (Cell Signaling Technology, 3180, 1:1000), anti-GAPDH (Cell Signaling Technology, 2118, 1:2000), anti-mTOR (Cell Signaling Technology, 2972S, 1:1000), anti-Phospho-mTOR (S2448) (Cell Signaling Technology, 5536T, 1:1000), anti-Phospho-MEK1 (Ser298) (Cell Signaling Technology, 9128S, 1:1000), anti-MEK1/2(D1A5) Rabbit (Cell Signaling Technology, 9124S, 1:1000), anti-p70 S6 Kinase (Cell Signaling Technology, 9202S, 1:1000), anti-Phospho-p70 S6 Kinase (Cell Signaling Technology, 9205S, 1:1000), anti-mouse IgG-HRP (Jackson ImmunoResearch Laboratories, Inc., 115-035-003, 1:10000),

anti-rabbit IgG-HRP (Jackson ImmunoResearch Laboratories, Inc., 111-035-144, 1:10000), anti-goat IgG-HRP (Jackson ImmunoResearch Laboratories, Inc., 705-035-003, 1:10000), HCS LipidTOX™ Green Neutral Lipid Stain (Thermo Scientific, H34475, 1:100), Lipofectamine RNAiMAX (Thermo Scientific, 13778150), and Fatostatin (Cayman Chemical, 12562).

**Immunofluorescence**. Cells grown on coverslips, were fixed with 4% paraformaldehyde in PBS for 15 min, and permeabilized with 0.1% Triton X-100 in PBS for 10 min. Then cells were stained with HCS LipidTOX™ Green Neutral Lipid Stain (Thermo Scientific, H34475) for 1 h at room temperature. After washing, cells were analyzed using the Zeiss LSM800 confocal microscope.

**Western blotting**. Total cell lysates were prepared in Laemmli sample buffer (Bio-Rad, 161-0737). Protein concentrations were determined with Bio-Rad Protein Assay Kit (Bio-Rad) according to the instruction. Western blotting was performed for 20 μg protein using SDS-PAGE followed by transfer to nitrocellulose membrane (Bio-Rad). Primary antibody was incubated overnight in cold room. Secondary antibody was incubated for 1 h at room temperature.

**Chromatin immunoprecipitation (ChIP)**. ChIP assay was performed in TE5 and KYSE150 cells. Using 15 ml tubes, we first harvested $1 \times 10^7$ cells TE5 or KYSE150 cells cultured in 15 cm dish and washed cells with cold PBS twice. Then the cells were fixed in 4 ml of 1% paraformaldehyde for 10 min at room temperature, which were quenched with 4 ml of 250 mM glycine for 5 min, followed by two washes with cold PBS. These cells were lysed twice with 1 ml lysis/wash buffer (formula: 150 mM NaCl, 0.5 M EDTA pH 7.5, 1 M Tris pH 7.5, 0.5% NP-40) containing protease inhibitors. Cells were lysed by mechanically pipetting up and down several times in a microcentrifuge tube. We then centrifuged these samples at $12,000 \times g$ for 5 minutes at 4 °C and discarded the supernatant. Cell pellets were next resuspended in 1 ml shearing buffer (formula: 1% SDS, 10 mM EDTA pH 8.0, 50 nM Tris pH 8.0) and were sonicated using a Covaris sonicator. Subsequently, we cleared the samples by centrifugation at $12,000 \times g$ for 10 min at 4 °C and retained the supernatants. The supernatants were then diluted five times using the dilution buffer (formula: 0.01% SDS, 1% Triton X-100, 1.2 mM EDTA pH 8.0, 150 nM NaCl). We then added primary antibodies (5 μg SREBF1 or KLF5) and incubated with the samples at 4 °C overnight on a rotating platform. Dynabeads Protein G beads (Thermo Scientific, 10004D) were then added the next morning, and the samples were incubated at 4 °C for an additional 4 h on a rotating platform. These Dynabeads were collected by centrifugation and were washed five times with cold wash buffer and once with cold TE buffer. DNA samples were generated by reverse crosslinking using 5 M NaCl at 65 °C overnight and purification. The final products were subject to DNA library preparation and deep sequencing using Illumina HiSeq platform.

**ChIP-Seq data analysis**. ChIP-Seq data of SREBF1 were generated in both TE5 and KYSE150 cell lines, and ChIP-Seq data of KLF5 was generated in TE5 cell line. ChIP-Seq data of SREBF1 in MCF7 and HepG2 cell lines were from ENCODE consortium. In addition, we reprocessed H3K27Ac ChIP-Seq data in eight ESCC cell lines (TE5, TE7, KYSE70, KYSE150, KYSE140, KYSE180, KYSE200, and KYSE510) from our previous work[30,31,33]. Briefly, 150 bp pair-end and 50 bp single-end reads were aligned to human reference genome (HG19) using Bowtie2 (v2.2.6) (k = 2)[77]. Then we used Picard MarkDuplicates tool to mark PCR duplicates. ENCODE blacklisted regions were removed (https://sites.google.com/site/anshulkundaje/projects/blacklists). Macs2 was utilized to identify the peaks with the parameters –bdg –SPMR –nomodel –extsize 200 -q 0.01. Bigwig files were generated by bamCompare in DeepTools (v3.1.3) using parameters –binSize 10 –numberOfProcessors 5 –scaleFactorsMethod None –normalizeUsing CPM –ignoreDuplicates –extendReads 200 from Ramirez et al., 2014[78]. In addition, bigwig files of H3K27Ac, H3K4Me3, TP63, and SOX2 ChIP-Seq in TE5 cell line were generated by us previously[30,33]. The bigwig files were visualized in Integrative Genomics Viewer (IGV)[79]. Data have been deposited to GEO database (GSE143803).

**Circular chromosome conformation capture sequencing (4C-seq) assay**. 4C-seq experiments[33,80–82] were performed in TE5 cells. We first collected 40 million TE5 cells and performed crosslinking using 1% formaldehyde for 10 min by rotating. The reaction was quenched by incubation with 2.66 M glycine for 5 min. We then isolated the nuclei pellets using lysis buffer containing protease inhibitors. Nuclei pellets were next resuspended in 1× CutSmart buffer (NEB) containing 0.3% SDS and incubated at 37 °C with shaking in thermomixer for 1 h. We then added 10% Triton X-100 to a final concentration of 2% followed by incubating with shaking at 37 °C for 1 h. Nuclei were next digested by 4000 U HindIII (NEB) at 37 °C, 900RPM overnight. HindIII digestion efficiency was assessed by gel electrophoresis using the extracted DNA from QC samples. Upon the verification of the digestion efficiency of QC samples, HindIII digestion was inactivated by 10% SDS (to a final concentration of 1.6%) and incubated at 65 °C for 25 min. 100 μg digested DNA was quantified by Qubit BR kit and used for ligation with 1% Triton X-100 and 990 U T4 DNA ligase (Thermo-Fisher Scientific) at 16 °C overnight. We then reverse cross-linked the ligation reaction by proteinase K (Thermo-Fisher

Scientific) at 65 °C for 4 h and incubated at 37 °C overnight. The "3 C libraries" were generated by ligation products after purification by phenol-chloroform and ethanol precipitation. This "3 C libraries" were then digested by a 500 μl DpnII reaction overnight at 37 °C using this formula: 50 μg 3 C library, DpnII buffer (NEB) and 50 U DpnII (NEB). This reaction was again followed by phenol-chloroform purification and ethanol precipitation and Qubit BR quantification, which produced the "4C libraries". 3.2ug 4C libraries for each viewpoint were then amplified by two rounds of PCR reactions. The scaled-up product was loaded into 4–20% TBE gel followed by gel crush, clean-up (by AMpure beads) and Bioanalyzer quantification to generate the final libraries. DNA Sequencing was performed using MiSeq platform with Illumina 150 bp pair-end kit. Two replicates were generated and sequencing results were analyzed by r3Cseq package[83].

The viewpoint of SREBF1 promoter region was chr17:17738740-17740085 (hg19) with 4C primers: NEST-Forward-GCAACCAGCTGGGCTCAT, OUTER-Forward-GAAGCAACGGGCCTCCTAAT, NEST-Reverse-CTGCTGACCGACATCGAAGG and OUTER-Reverse-TTGCGAGGTTACTCACGGTC. 4C-seq Data have been deposited to the Gene Expression Omnibus (GSE178923).

**Computational annotation of typical-enhancer and super-enhancer**. The method of ROSE (Rank Order of Super Enhancers)[84,85] was first used to identify enhancers defined as H3K27Ac peaks 2 Kb away from any transcription start site (TSS). Following stitching enhancer elements clustered within a distance of 12.5 Kb, typical-enhancers and super-enhancers were then classified using a cutoff at the inflection point (tangent slope = 1) based on the ranking order. The scatter-plots (Fig. 2b) contained all typical-enhancers and super-enhancers from each sample.

**Luciferase reporter assays**. Enhancer and promoter elements were amplified by PCR and cloned into pGL3-based luciferase reporter vectors (Promega, E1761). Primers were provided in Supplementary Table 4. Constructs were verified by Sanger sequencing. Vectors were transfected into TE5 cell line using BioT (Bioland, B01-01). A Renilla luciferase vector was co-transfected as a control for normalization. After 48 h of transfection, the luciferase activity was measured by the Dual-Luciferase Reporter Assay System (Promega, E1960).

**Construction of expression vectors**. The shRNA expression vector was designed based on siRNA sequences and cloned into pLKO-puro vector by Guangzhou IGE Biotechnol OGY LTD. The ORF of SREBF1 was cloned into pCDH-CMV-MCS-EF1-copGFP-T2A-Puro by Guangzhou IGE Biotechnology Ltd. The sgRNA sequences were cloned into pLV hU6-sgRNA hUbC-dCas9-KRAB-T2a-puro vector, which was from Addgene (plasmid catolog#: 71236). To produce viral particles, the recombinant viral vectors and packaging vectors were co-transfected into 293FT cells. Supernatants were harvested and filtered through a 0.45 μM filter 48 h after transfection. ESCC cells were then infected with the virus in the presence of 10 mg/ml Polybrene. The siRNA, shRNA and sgRNA sequences are provided in Supplementary Table 5.

**Liquid chromatography tandem mass spectrometry (LC-MS/MS)-based lipidomics**. LC-MS/MS-based lipidomics[86] was performed in KYSE510 cells. We first extracted lipid species from $1 \times 10^6$ cells by methyl tert-butyl ether (MTBE) (Sigma Aldrich) and concentrated the samples in a SpeedVac concentrator (Thermo Scientific). Dried lipid samples were resuspended using a loading buffer (formula: 50% isopropanol, 50% methanol) and analyzed by LC-MS/MS, using an Ultimate 3000 XRS LC system connected to an Orbitrap Fusion Lumos mass spectrometer (Thermo Scientific). We loaded 20 μl of lipid solution onto a 15-cm Accucore Vanquish $C_{18}$ column (1.5 μm particle size, 2.1 mm diameter). Lipid molecules were separated using a 28-min LC gradient (35–60% mobile phase B for 4 min, 60–70% B for 16 min, 70–100% B for 1 min, 100% B for 3 min, 100–35% B for 0.1 min, and 35% B for 3.9 min) at a flow rate of 0.3 ml/min. Here, the mobile phase A consisted of 60% acetonitrile, 10 mM ammonium formate and 0.1% formic acid, while the mobile phase B consisted of 90% isopropanol, 10% acetonitrile, 10 mM ammonium formate and 0.1% formic acid. Mass spectra were next acquired by an Orbitrap Fusion Lumos operated in both positive and negative ion modes. Parameter settings for FTMS1 were as follows: orbitrap resolution of 120,000, scan range of $m/z$ 250–1,200, AGC of $2 \times 10^5$, maximum injection time of 50 ms, RF lens of 50%, data type of profile, dynamic exclusion for 8 s with a mass tolerance of 25 ppm, and cycle time of 2 s. Parameters for FTMS2 were as follows: orbitrap resolution of 30,000, isolation window of 1.2 $m/z$, activation type of HCD, collision energy of 30 ± 3%, maximum injection time of 70 ms, AGC of $5 \times 10^4$, and data type of profile.

We analyzed the acquired raw files using the software LipidSearch (v1.4) (Thermo Scientific) for sample peak alignment, MS/MS identification as well as calculation of MS1 peak area. Statistical analyses were performed using the Perseus (v1.6.6.0) software[87], and the $P$ values were determined by two-tailed Student's $t$-test and followed by multiple hypothesis correction using the Benjamini–Hochberg method.

**Immunohistochemistry**. A total of 179 formalin-fixed, paraffin embedded (FFPE) tissues were collected from surgical samples of ESCC patients treated at the Shantou Central Hospital (November 2007 to January 2011). All tumor and nonmalignant samples were confirmed by pathologists of the Shantou Central Hospital. This study was approved by the ethical committees of the Shantou Central Hospital as well as the Medical College of Shantou University.

We first constructed tissue microarrays using FFPE specimens described above, which were cut into 4-μm sections. Immunohistochemistry (IHC) was performed by a two-step protocol (PV-9000 Polymer Detection System, ZSGB-BIO, Beijing, China) according to the manufacturer's instructions. We cleared the paraffin of the FFPE sections three times with xylene for ten minutes each, followed by rehydration using gradient alcohol (100%, 95%, 75%, 50% alcohol). Then we rinsed these glass slides twice by deionized water for five minutes each. The tissue sections were blocked with 5% BSA for 30 min and then incubated with primary antibodies (Anti-SREBF1, Proteintech, 14088-1-AP, 1:300; anti-TP63, Proteintech, 60332-1, 1:500; anti-KLF5, Santa Cruz Biotechnology, sc-398409X, 1:500) at 4 °C overnight. The tissue sections were rinsed with PBS, followed by the incubation of Reagent 1 and then Reagent 2 at 37 °C for 20 min each. After three washes by PBS, glass slides were stained with diaminobenzidine and counterstained by hematoxylin. The staining of SREBF1, TP63, and KLF5 was evaluated by IHC scores[88,89]. Briefly, the images of IHC on tissue microarrays were obtained using PerkinElmer Vectra on brightfield microscope. The level of nucleus or cytoplasm of SREBF1 protein in individual sample was automatically evaluated and digitally scored based upon the intensity of staining, as well as the proportion of cells with positive staining by the Nuance™ system and InForm™ software (Cambridge Research & Instrumentation). The staining score (ranging from 0 to 300) was used for statistical analysis. The overall survival was defined as the time from the date of primary surgery to the date of death due to esophageal cancer and data on survivors were recorded at the last follow-up.

**Colony formation assay**. Five hundred cells in 2 ml of complete medium were seeded into six-well plates, which were kept at 37 °C in an incubator with 5% $CO_2$ for 1 week. Colonies were fixed with methanol and stained with 0.05% of crystal violet, and counted using ImageJ software.

**Trans-well migration assay**. Trans-well migration assay was performed in 6.5 mm insert with 8.0 μm polycarbonate membranes in 24-well plates (COSTAR, 3422). $5 \times 10^4$ cells were seeded on top of the chamber with 100 μl of serum-free medium. The bottom chambers were supplied with regular medium with 10% FBS. After 48 h, top chamber surfaces were wiped by a cotton stick, and cells that migrated to the other side of chamber membrane were fixed by methanol, stained with 0.5% crystal violet and quantified by counting 10 random fields under a light microscope (×200). Data obtained from three separate chambers were shown as mean values.

**Xenograft assays in nude mice**. Five-week-old male nude mice were purchased for the xenograft assays from Vital River Laboratories (Beijing, China)[88]. All animal studies were conducted in accordance with protocols approved by the Animal Research Committee of the Shantou Administration Center of Shantou University. Ten mice were randomly separated in two groups and subcutaneously injected with $1 \times 10^6$ cells expressing inducible vectors containing either nontargeting scrambled shRNA or shRNA against SREBF1. The animal housing room was maintained at 22.2 ± 1 °C (72 °F), 30–40% humidity with at least 12 fresh-air changes hourly and a controlled 14:10-h light:dark cycle.

For experiments using Fatostatin[16,51], at day 7 post tumor cell injection, drug administration was initiated. Briefly, a 100 mg/ml Fatostatin stock solution was prepared in 100% dimethyl sulfoxide. On the day of injection, Fatostatin was diluted in PBS to a final concentration of 30 mg/kg in 200 μL solution, which was administered to mice. Fatostatin and its diluent control was intraperitoneally administered daily for 14 days. Xenograft size was measured by (length × width² × 0.5) every 4 days for 21 days. Mice were euthanized at the end of experiment, and xenograft tumors were extracted for analysis.

**Statistical analysis**. The analyses for IHC were performed with SPSS for Windows ver.18.0 software (SPSS Institute, Chicago, IL, USA). Kaplan–Meier curve was constructed for overall survival analysis using a Log-rank test. Each $P$-value is two-tailed and significance level is 0.05. For comparisons of continuous variables between groups, two-tailed Student $t$-test was used. The values at $P < 0.05$ (*) and $P < 0.01$ (**) were considered statistically significant. Diagrams were created by GraphPad Prism software and data were shown as the mean ± SEM. The exact $P$-values were provided in Supplementary Data 5.

**Reporting summary**. Further information on research design is available in the Nature Research Reporting Summary linked to this article.

## Data availability
The mRNA expression (RNA-Seq level-3 data) data of 23 types of cancers were retrieved from the datasets produced by The Cancer Genome Atlas Network (TCGA, GDC v16.0) using TCGAbiolinks (V2.14.1) R package. Microarray RNA expression data of ESCC and LUSC were retrieved from GEO database (GSE53624, SRP064894, and GSE4573). ChIP-Seq data of SREBF1 in MCF7 and HepG2 cell lines were from ENCODE database (https://www.encodeproject.org/). H3K27Ac ChIP-Seq data in eight ESCC cell lines (TE5, TE7, KYSE70, KYSE150, KYSE140, KYSE180, KYSE200, and KYSE510) were collected from our previous studies (GSE106563, GSE131493, and GSE106434)[30,31,33]. H3K4Me3, TP63, and SOX2 ChIP-Seq in TE5 cell line were collected from our previous studies (GSE106563 and GSE131493)[30,33]. The blacklisted regions were downloaded from ENCODE database (https://sites.google.com/site/anshulkundaje/projects/blacklists). Molecular Signatures Database v7.4 (https://www.gsea-msigdb.org/gsea/msigdb/index.jsp) were used to do the GSEA (v3.0) (https://www.gsea-msigdb.org/gsea/index.jsp). The ChIP-Seq, RNA-Seq, and 4C-seq datasets generated in this study have been deposited in the Gene Expression Omnibus (GEO) repository under the accession code GSE143803 and GSE178923, respectively. The remaining data are available in the article or Supplementary Information files, or available from the authors upon request. The full scans of Western blotting and the data presented in a plot, chart or other visual representation format were provided in the Source Data file. Source data are provided with this paper.

## Code availability
No unpublished code was used in this manuscript.

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

## Acknowledgements

This work was supported by Cedars-Sinai Cancer at Cedars-Sinai Medical Center through the Translational Oncology Program Developmental Fund, Cedars-Sinai Cancer Project Acceleration Fund, and Cancer Prevention and Control Program Discovery Fund (to D.-C.L.); This research was also partly supported by NIH under award R37CA237022 (to D.-C.L.).

## Author contributions

D.-C.L. conceived and devised the study. L.-Y.L., Q.Y., Y.-Y.J., W.Y., Y.J., and D.-C.L. designed experiments and analysis. L.-Y.L., Y.-Y.J., Y.J., B.Z., X.L., G.-W.H., X.-E.X., and Y.Z. performed the experiments. L.-W.D. and Q.Y. performed bioinformatics and statistical analysis. M.H., S.G., A.H., Z.Z., S.J.M., L.-Y.X., and E.-M.L. contributed reagents and materials. L.-Y.L., Q.Y., Y.-Y.J., Y.J., B.Z., M.-R.W., M.J.F., S.J.F., W.Y., H.P.K., and D.-C.L., analyzed the data. L.-Y.X., E.-M.L., H.P.K., and D.-C.L. supervised the research and wrote the manuscript.

## Competing interests

The authors declare no competing interests.

## Additional information

Li-Yan Li[1,2,11✉], Qian Yang[2,11], Yan-Yi Jiang[2,11], Wei Yang[3,11], Yuan Jiang[2,11], Xiang Li[1], Masaharu Hazawa[4], Bo Zhou[3], Guo-Wei Huang[2], Xiu-E Xu[1], Sigal Gery[2], Ying Zhang[5], Ling-Wen Ding[5], Allen S. Ho[6], Zachary S. Zumsteg[7], Ming-Rong Wang[8], Melissa J. Fullwood[5], Stephen J. Freedland[9], Stephen J. Meltzer[10], Li-Yan Xu[1✉], En-Min Li[1✉], H. Phillip Koeffler[2] & De-Chen Lin[2✉]

[1]The Key Laboratory of Molecular Biology for High Cancer Incidence Coastal Chaoshan Area, Shantou University Medical College, Shantou, China. [2]Department of Medicine, Samuel Oschin Cancer Center, Cedars-Sinai Medical Center, Los Angeles, CA, USA. [3]Departments of Surgery and Biomedical Sciences, Cedars-Sinai Medical Center, Los Angeles, CA, USA. [4]Cell-Bionomics Research Unit, Innovative Integrated Bio-Research Core, Institute for Frontier Science Initiative, Kanazawa University, Kanazawa, Japan. [5]Cancer Science Institute of Singapore, National University of Singapore, Singapore, Singapore. [6]Division of Otolaryngology-Head and Neck Surgery, Department of Surgery, Samuel Oschin Cancer Center, Cedars-Sinai Medical Center, Los Angeles, CA, USA. [7]Department of Radiation Oncology, Cedars-Sinai Medical Center, Los Angeles, CA, USA. [8]State Key Laboratory of Molecular Oncology, National Cancer Center/Cancer Hospital, Chinese Academy of Medical Sciences and Peking Union Medical College, Beijing, China. [9]Division of Urology, Department of Surgery, Cedars-Sinai Medical Center, Los Angeles, USA and the Durham VA Medical Center, Durham, NC, USA. [10]Departments of Medicine and Oncology, Johns Hopkins University School of Medicine and Sidney Kimmel Comprehensive Cancer Center, Baltimore, MD, USA. [11]These authors contributed equally: Li-Yan Li, Qian Yang, Yan-Yi Jiang, Wei Yang, Yuan Jiang. ✉email: lly170@126.com; lyxu@stu.edu.cn; nmli@stu.edu.cn; dchlin11@gmail.com

