## [Peer Review File · Nature Communications]

Reviewers' comments:

Reviewer #1 (Remarks to the Author); expert on SCC:

TP63 is often deregulated in SCCs. By gene set enrichment analysis of public available SCC data sets the authors identify fatty-acid metabolism pathway as a key effector downstream of p63. The mechanism involves the positive regulation by p63 of the ER-associated SREBP1 transcription factor, a known and well characterised regulator of such pathway. p63 and SREBP1 act as a complex with KLF5, previously shown by the authors to associate at enhancer/super-enhancer chromatin regulatory regions in SCCs.

All three TF are over-expressed in SCCs and their dys-regulation targets colony formation, viability, migration of ESCC cells and might be used as a prognostic marker being associated with poor SCC patient survival.

The work addresses a potentially important mechanism of genetic/epigenetic deregulation in SCCs involving the modulation of fatty acid metabolism, a relevant pathway often deregulated in tumors. However, none of the several mechanisms that are explored is convincingly demonstrated.

The relevance of SREBP1 deregulation in vivo is marginal (only tumor weight is assessed) and further evidence should be obtained. Is fatty acid metabolism really involved? The relative contribution of the canonical fatty acid versus AKT-TOR pathways, both regulated by the p63/SREBP1/KLF5 axis, is not clear.

The manuscript would benefit substantially from a better focus on selected aspects, to be more clearly and definitely investigated. Several data obtained from public sets should be relegated to the supplementary section. As it is, the multiple observations provided by the manuscript are limited general significance.

Specific points:

- The list of genes in Fig1a should be provided (as supplementary table)
- The GSEA in Fig1e, does not provide sufficient information on the regulation of fatty acid genes; this association should be verified in multiple SCC cell lines after p63 silencing by multiple siRNAs with adequate controls for off target effects. Also, a GSEA with $p=0.048$ is of questionable statistical significance.
- Fig1f is too small and it is not clear how the quantification was made; the data, like in other figures, have no adequate statistical analysis and error bars.
- in Fig1h, relative RNA expression changes (measured as $2^{\Delta\Delta Ct}$) are unclear. Values for scrambled siRNA control should be equal to 1 (not 0.10) and it is difficult to believe that the observed changes in gene expression are statistically significant (for a difference between 0.02 and 0.01). 0 should be 0 not 0.00 and 0.000. The data for several genes are not there. Importantly, the experiment should be repeated with several SCC cell lines. This is also a major concern for the rest of the paper, which is based on analysis of very few SCC lines, often with p63 silencing by a single si/shRNA.
- Fig1i should be moved to supplementary. The data for the transcription factors "screening" are from TCGA. Most of these TFs were barely expressed in ESCC cells, bringing into

question the biological significance. Some statements are not supported by any evidence.

- Fig 3a : silencing of KLF5 in KYSE510 does not change levels of SREBF1 or p63, is it not countering one of the main conclusions ?
- Fig 3e " knockdown of SREBF1 significantly reduced the expression of TP63 and KLF5 at both mRNA and protein levels across different ESCC cell lines". This is not evident in TE5 cells at mRNA (for both KLF5 and p63) or protein (klf5) levels.
- The in vivo work should provide analysis of proliferation/differentiation markers and be expanded to multiple cell lines.
- Fig 3f is too small and cannot be deciphered. What are 1.0 and 1.5 in the panel?
- In the text is indicated that, for figure 4b, a shSREBF1 was used while in the figure it is indicated siRNA.
- Some more information should be provided regarding the doses of Fatostatin used for these experiments (dose/response experiment – literature reference).
- Typing error in lines 252-253 and in line 208 probably wrong reference to fig2.
- For in vitro proliferation studies (one week assays), how persistent are the gene silencing effects by siRNAs ?

Reviewer #2 (Remarks to the Author); expert on lipid metabolism:

Review Li et al Nat Comms

The manuscript by Li et al. reports a study investigating the role of the lipogenic transcription factor SREBP1 in regulating gene expression in squamous cancers (SCC). This is based on their initial finding that TP63, a transcription factor (TF) highly expressed in SCC, regulates many genes involved in fatty acid and cholesterol metabolism. The authors then perform a series of experiments ranging from RNAseq, ChIPs and targeted analyses to show cooperation between TP63, KLF5 and SREBP1 in regulating genes involved in fatty acid metabolism in SCC. Most experiments are performed in at least two cell lines derived from esophageal squamous cell carcinoma (ESCC). The authors use 4C ChIP to demonstrate that TP63 and KLF5 cooperate to regulate the expression of the SREBF1 gene. They also provide evidence that SREBP1, TP63 and KLF5 occupy the same promoters in ESCC cells. Moreover, gene silencing experiments show that these factors regulate each other by binding to their respective promoter elements.

Using lipidomics and RNAseq data, the authors also show that SREBP1 controls the expression of genes involved in fatty acid, glycerophospholipid and sphingolipid metabolism. Moreover, they show that SREBP1 expression is enhanced in human ESCC compared to normal tissue and that silencing of SREBP1 using shRNA or inhibition of SREBP activation using the SCAP inhibitor fatostatin blocks xenograft tumour growth of KYSE150 cells. Moreover, they use ChIPseq analysis to identify genes that are occupied by SREBP1 selectively in ESCC cells and those that are occupied by all three TFs (SREBP1, TP63 and

KLF5). They then show that these genes map to ErbB2 and mTOR signalling and determine the regulation of mTORC1 and MEK1/2 signalling by SREBP1 in ESCC cells. From these experiments, the authors conclude that TP63, KLF5 and SREBP1 cooperate to regulate gene expression to drive lipid metabolism and signalling in ESCC.

Overall, the manuscript provides a substantial amount of data from well-executed experiments. These data mostly support the conclusions. However, there are some limitations in the stringency applied in data interpretation, specifically those based on data derived from public sources. This is particularly problematic in Figure 1i and 3h, where only a weak regulation and correlation can be derived. Those weaker data distract from the substantial findings of the manuscript and should be revised. Moreover, the text and figure legends contain a number of inaccuracies that make it difficult to follow the line of argument and assess the strength of the data. In addition, some experiments are not very well described and some conclusions not fully supported by the data.

The weakest part of the manuscript is the analysis of mTORC1 and MEK regulation downstream of SREBP1 in ESCC cells. Regulation of mTORC1 signalling by fatty acid metabolism has already been demonstrated (for example see Menon et al. JBC 2017 or Madak-Erdogan et al. Cancer Research 2019) and the authors do not provide substantial insight beyond the known findings. The manuscript falls short of identifying the exact mechanism by which SREBP1, TP63 and KLF5 control mTORC1 and MEK1/2. The manuscript also does not formally prove that the regulation of lipid metabolism and/or mTORC1/ErbB signalling downstream of SREBP1 is important to maintain ESCC survival and proliferation, as stated in the diagram shown in Figure 7f.

Nevertheless, the manuscript provides important insight into the cooperation of TP63 and KLF5 in ESCC and will be of substantial interests to the field. Below is a list of points that need to be carefully addressed before the manuscript can be considered for publication.

General comments:

Size markers should be provided for all western blots.

Figure numbers should be matched to the order in which they are mentioned in the text.

Care should be taken to ensure that the figure legends correctly describe the experiments shown in the figures.

Methods are quite superficial and should be checked for accuracy. All siRNA and shRNA sequences need to be provided.

Specific comments:

Fig 1f: Efficiency of TP63 k/d and overexpression of SREBP1 (full length or mature?) should be demonstrated and indicators of variance across independent biological replicates and significance of the results should be displayed. If full length SREBP1 is shown, it is unclear how this relates to its transcriptional activity.

Fig. 1h: This display is somewhat confusing. It would be better to display levels relative to control. According to methods, the results are based on semi-quantitative PCR and an absolute comparison of mRNA levels between different genes is not possible using this method. A second RNAi sequence should be shown.

Fig. 1i: This figure does not show any indicators of significance. This is essential to support the conclusions stated in the text. The upregulation of SREBP1 expression in tumour tissue is rather mild (log₂FC between 0.5 and 1.5). Is this sufficient to conclude that SREBP1 is overexpressed in squamous tumours?

Fig. 2d: A positive control for SOX2 ChIP needs to be shown in order to confirm specificity of

the antibody.

Fig. 2f and 3a: A second sequence targeting KLF5 is needed to substantiate the conclusions drawn from this result.

Fig. 3a, b and e (and others): It should be stated which form of SREBP1 (full length or mature) is displayed in all western blot experiments.

Fig. 3d: Statistical indicators are needed. What is the negative control used here?

Fig. 3e: The evidence for reduced expression of TP63 and KLF5 following SREBP1 k/d is not particularly strong in TE5 cells. A second siRNA sequence needs to be shown to validate this result, as it forms the basis of the overall conclusions of the work as shown in Figure 3i.

Fig. 3g: The control is strongly affected by SREBP1 k/d making the results inconclusive. This needs to be addressed. The figure legend also contains errors.

Fig. 3h: The correlations between SREBF1 and TP63 and KLF5 shown here are rather weak (R values around 0.3). This can only be considered as a “weak correlation” and it is not clear how this finding supports the overall conclusions.

Fig. 4b: Was siRNA or shRNA used in this experiment? There is a discrepancy between text and figure.

Fig 4e: Could a different colour code be used rather than shades of grey?

Fig. 4f: The RNAseq data displayed in this figure require further explanation in the text.

Which comparison was made here (control vs fatostatin or SREBP1 silencing vs control)?

This is important as fatostatin also blocks SREBP2. Abbreviations should be explained in the legend.

Fig. 5b: The increased expression of SREBF1 in ESCC looks interesting and convincing. However, in contrast to the staining for TP63 and KLF5, SREBF1 seems to be present in both cytoplasm and nucleus. It could be interesting to score nuclear and cytoplasmic staining separately, which could make the difference between normal and tumour tissue more stringent. Scale bars need to be labelled.

Fig. 6a: Given the substantial cell line specificity of the SREBP1 binding peaks identified in HepG2 and MCF7 cells, peaks that are specific to KYSE150 and TE5 cells should also be displayed. This would allow an estimation whether ESCC cell lines show a higher overlap in occupancy compared to cell lines derived from other tissues. Ideally, statistical methods should be employed to calculate the similarity between the two ESCC cell lines.

Fig. 6f: It is surprising that the GSEA shown in here only shows moderate correlation between the 473 ESCC specific genes and genes regulated by silencing of SREBP1 in TE5 cells (NES -1.24, $p=0.04$). The same applies to the silencing of TP63 (NES -1.14, $p=0.06$). Would it make more sense to conduct this analysis on the 274 triple-binding genes? Care should be taken to not overstate the conclusions from this result.

Fig 7a and b: It would be helpful to indicate the size of the pathway (i.e. number of genes in the pathway) rather than just the number of occupied genes.

Fig. 7b: It is unclear whether the analysis was performed using the 473 ESCC specific genes, as indicated in the text and legend, or the 274 genes co-occupied by SREBP1, TP63 and KLF5, as indicated in the figure. The second analysis would be far more meaningful.

Fig. 7c: It needs to be explained what exactly is displayed in the heatmap. Fold change over control siRNA? How much siRNA was used? How long was the silencing? There is very little information provided in the methods. Moreover, based on the results of this figure, the authors conclude that genes of the mTOR and ErbB signalling pathways are downregulated by silencing of SREBP1, TP63 or KLF5. What is the cut-off used to identify those downregulated genes? This is difficult to discern from the heatmap. It would be better to apply statistical methods to analyse if triple occupied genes are more likely to be downregulated by silencing of one or all of these TFs. This is crucial as some genes are only moderately or not at all downregulated upon silencing. However, the final conclusion of the manuscript, also outlined in the diagram in Figure 7f, strongly depends on the regulation of

these genes by all three TFs. In its current form, this only represents an incidental observation.

Fig. 7e: The results on MEK are difficult to interpret as there are effects on total MEK1/2 levels. The WB results should be quantified and normalised to GAPDH or total proteins for phospho-analysis.

Fig. S2: Statistical indicators should be shown.

Comments to text:

Line 76: Please provide reference for this statement.

Line 196: Where are the data on SOX2?

Line 250: Fatostatin is not a specific SREBP1 inhibitor. It prevents processing of both SREBP1 and SREBP2.

Line 257: Typo in cell line name.

Reviewer #3 (Remarks to the Author); expert on transcriptional regulation:

Lin and colleagues provide strong data to support a regulatory and potential function relationship between SREBF1, TP63 and KLF5 in ESCC. They carry out a thorough characterization of the transcriptional landscape of these factors and propose how they may cooperate to regulate both lipid metabolism selective pathways including ErbB/mTOR signaling in this context. Some clarifications and a more direct demonstration of the role of p63 in lipid metabolism would further strengthen the manuscript.

Specific Comments:

Fig. 1f. It would be helpful to have a direct comparison of p63 knockdown to SREBF knockdown.

1h/i It seems surprising that the other factors shown are suppressed/not expressed in tumors. Can the authors comment?

Fig. 2e. These data are interesting but do not prove a direct effect of p63. Can the authors demonstrate effects on an enhancer in which the canonical p63 binding site is mutated?

Fig. 3g. Similar to 2e, can the authors demonstrate effects on an enhancer in which the SREBF binding site is mutated? That would provide evidence of a direct effect.

Fig. 4d/e These findings are interesting and consistent with the established function of SREBF. However, a major claim of the paper is that p63 itself through its regulation of SREBF controls lipid metabolism. It would add substantially to the paper if lipidomics could be performed in the setting of p63 knockdown and compared to the results shown.

Fig. 5a It is not too surprising that the levels of these proteins are higher in tumor than in normal. What we would actually like to know is whether the levels are correlated in individual tumors. Specifically, the correlation coefficients for IHC scores should be depicted in the same way that the RNA expression correlations are depicted in 3h.

Fig. 7b Are the results the same if one looks at genes that are both associated with ESCC-specific peaks and also significantly regulated by these factors (i.e. as defined by p63 and KLF5 knockdown studies)?

Reviewer #4 (Remarks to the Author); expert on superenhancers and 4Cseq:

NCOMMS-20-05458 from Li et al. is a well-written manuscript that identifies an important finding for an integrated set of key transcription factors that affect 2 functional pathways in squamous cell carcinoma (SCC). The major claims of the paper for SCC relevance are the 1) functional significance of the fatty acid metabolic pathway that is regulated by the well-established SCC driver TP63 and mediated by SREBF1 TF-driven epigenetic regulation and 2) complex co-regulatory feedback loop for SREBF1, TP63, and KLF5 to further drive SCC formation and for additional identification of the mTOR/ERBB pathway. The discoveries were rigorously determined using GSEA analyses of comprehensive TCGA in vivo datasets, bioinformatics analyses to identify enrichment of SREBF1 motifs in fatty acid pathway genes, multiple functional experiments using gene knockdown (siRNA), epigenetic ChIP, 4C-seq, gene expression, epigenetic repressed (CRISPRi), cellular assays for lipid formation and migration and evident in replicates, knockdown xenograft model in mice, lipidomics, and human SCC immunohistochemistry. The claims are novel and are of great interest to the cancer field and genomics scientific community as the findings for fatty acid/lipid and mTOR/ERBB pathways and SREBF1/TP63/KLF5 TF feedback loop and demonstrated to be functionally relevant have not been reported for SCC and further illustrates how the multi-omics approach have facilitated these important discoveries. The work is for the most part convincing as mentioned previously given the multiple approaches to link the genomics findings to pathways that functionally impact SCC development. However, there are major concerns for claims for the superenhancer, 4C studies, and effects on SL synthesis. Details for superenhancer and 4C-seq methodologies and analyses are completely missing and hence their interpretations are difficult to assess and to justify their corresponding claims. The effects of SL synthesis are not completely clear as validation experiments were not performed to assert that SREBF1-regulated enzymes affect this pathway. The conclusions are original except for the assertion that inhibition of the superenhancer for SREBF1 resulted in decreased TP63 and KLF5 which is not substantiated as the annotation for superenhancer are not clear. It would be nice to include a few details about how this new knowledge could open up new therapeutic strategies to target these pathways. Overall, the manuscript is impactful and it very nicely establishes the SREBF1/TP63/KLF5 regulatory feedback loop and shows how they contribute to cancer associated signaling pathways specifically in SCC associated with known poor survival. The work is extensive and nicely combines both in vivo and in vitro data to convincingly illustrate their findings. However, major concerns are noted and additional comments are provided below.

Major concerns are the lack of details for superenhancer annotation and 4C-seq methodology, downstream bioinformatic analyses, sequences for bait. Which cells? Which antibody for superenhancer? How many reads? How many replicates? Filtering criteria? etc.

1. Page 3, lines 67-68 – reference for Warburg effect?
2. Page 6, lines 124- 125 – included references do not mention HPV+ tumors in HNSC as different. Need another reference. Also spell out HPV.
3. Page 6, lines 130-131 - is the $P < 1E-06$ cutoff used to determine significance for all DE genes? This is not listed in figure legend or methods.
4. Page 7, lines 137-139 - estrogen response late pathway is enriched in all 3 cancer types but is the only one out of the 9 that was not commented on in the manuscript and should be mentioned.
5. Figure 1c: when was TCGA accessed? Text (line 141) says all cancer types were analyzed, but Fig 1c only shows 23 types and TCGA currently lists 32 types. Kindly provide information on release of TCGA that was used.

6. Figure 1d. P=0 for TCGA ESCC GSEA?
7. Stats on Figure 1f?
8. Fig 1h – what about effect of TP63 knockdown on E2F and Sox5?
9. Major concern - It is not clear how super enhancers were found for the ESCC lines.
10. Fig 2b- It is not clear if “all enhancers” here also include super enhancers? Please clarify.
11. Chr number and genome build for Fig 2 is needed. Where is the bait label for SREBF1 promoter? 4C-seq shows enrichment to 4 regions but the seemingly negative results outside the 4C-seq region annotation are likely false negative.
12. How are the authors determining what is regulating SREBF1? Is it all (P, E1, E2, and E3) that were tested? Is there a rank order or hierarchy of the regulatory elements for activation? More importantly, the claim of TP63 and KLF5 binding to the superenhancer for SREBF1 (discussion) is not substantiated as there is no evidence of their binding in the “control” region and is located within the superenhancer. It’s okay to state binding to several enhancers instead of superenhancer.
13. Page 9, line 202- reference the paper for the H3K27ac ChIP data here (currently listed in the methods section)
14. Page 9, line 205 – I believe the author is referring to Fig 2e here and not 2d.
15. Authors should mention that sh knockdown of both p63 and KLF5 affected promoter activity as well and not just 3 enhancers. Is there not another shKLF5 construct? It looks like this is a pooled reagent. If so, please state in figure.
16. Fig 3d – should be compared to no dCas9 to rigorously determine relative repression for each regulatory element. At this point and with lack of how negative control is defined, it is not clear if there are significant effects for decreased SREBF1.
17. Fig 3e – TP63 does not look to be significantly decreased with siSREBF1 knockdown in TE5 cells
18. Fig 3f – 4C contact of SREBF1 promoter with SE of TP63 that is also bound by SREBF1 as determined by ChIP is not clear and well-established. The 4C is likely an artifact in the absence of showing peak height for the 4C experiment but has been determined to be enriched (how?) and yet has also not been validated via reciprocal 4C from TP63 to SREBF1 or by FISH. chr numbers are needed in figure and/or legend.
19. Fig 3h- log is inconsistently capitalized
20. Figure 4b- text says shRNA, figure marked siRNA
21. Figure 4f is a schematic of the pathways their findings seem to suggest, but they do not present the RNA-seq data that they used to create this schematic. Present in supplement
22. Page 12, line 262 – should reference figures 4d and 4e here.
23. It is unclear why only SPTLC1 is presented in figure 4g as the authors conclude multiple genes are direct downstream targets. Either don’t present this data or present this figure for all the implicated downstream targets in the supplement.
24. Does knockdown of SREBF1 affect cell viability and hence to explain decreased luciferase activity (Fig 3g) or even for any gene expression? Another marker for cell viability should be used to demonstrate live cell normalization. The impact to cell growth has also been demonstrated in fig 5d so how do the authors reconcile the perceived decrease in gene expression or luc activity? Indeed, Fig 5b shows co increased expression.
25. As SPTLC1, SPTLC2, ELOV4,-6,-7, and CERS6 are target genes and downregulated in Fatostatin-treated cells, is the downstream effects evident with decreased levels of 3kSN, SA+FA, and DHCer lipid intermediates?
26. Figure 5i-j – Representative images for the tumor sizes upon SHSREBF1 and Fatostatin treatments should be included in supplement.
27. Page 14-15, lines 312-324 – not sure what this is adding for cell type specific TFs
28. Page 15, line 336 – are the lipid enzyme genes in the 410 and 473 gene set? If not, how does the authors reconcile?

29. Page lines 361-365 – the findings for the number of decreased expressions for the genes appear to be found in one of the two biological replicates for SREBF1, TP63, and KLF5. Authors need to revisit.

30. For the mRNA heatmap shown in 7c, are these values significant as only overall FC is shown.

31. It would be helpful to arrange the methods in order of the experiments that are presented in the paper. Also, concentration of antibodies? How much protein? How were they imaged? Temperature and CO₂ levels for cell growth? Speed of centrifugation to remove debris in CHIP? What was used to reverse the crosslinks for CHIP?

The relevance of SREBPF1 deregulation *in vivo* is marginal (only tumor weight is assessed) and further evidence should be obtained. Is fatty acid metabolism really involved? The relative contribution of the canonical fatty acid versus AKT-TOR pathways, both regulated by the p63/SREBPF1/KLF5 axis, is not clear.

The manuscript would benefit substantially from a better focus on selected aspects, to be more clearly and definitely investigated. Several data obtained from public sets should be relegated to the supplementary section. As it is, the multiple observations provided by the manuscript are limited general significance.

Authors' reply:

We are abundantly appreciative of the Reviewer's careful evaluation of the work and, we thank you for these valuable comments and suggestions!

In addition to the specific comments below, the Reviewer queried the *in vivo* experiments as one of the general comments. We have performed the following additional experiments to address these concerns on the *in vivo* results:

1) In addition to the tumor weight, we have now also provided the measurement of tumor size (pasted below as **Figure R1A** for review convenience), which consistently showed that inhibition of SREBF1 using either shRNA or Fatostatin suppressed the growth of SCC xenograft.

2) During the revision, we performed immunohistochemistry (IHC) staining of KI67 protein in xenograft tissues to measure the proliferation rate. Notably, targeting of SREBF1 using either shRNA or Fatostatin potently reduced the expression of KI67 (see below **Figure R1B and 1C**), supporting our original finding that SREBF1 was required for the growth of SCC xenograft *in vivo*.

3) We further validated downstream pathways of TP63/SREBF1/KLF5 in these xenograft samples. Our original manuscripts focused on both fatty-acid metabolism and ErbB/mTOR signaling pathways, and identified target genes in each pathway (**Main Fig. 4 and 7**). We thus selected several key genes (ACLY, FASN, SCD for fatty-acid metabolism; SOS1, MAP2K2 for ErbB/mTOR signaling) for quantification in these xenograft samples. Indeed, inhibition of SREBF1 using either shRNA or Fatostatin consistently down-regulated the expression of these target genes in both pathways (**Figure R1D-E**), validating the transcriptional regulation by SREBF1 *in vivo*. Moreover, the expression levels of both TP63 and KLF5 were also expectedly reduced upon inhibition of SREBF1 (**Figure R1D-E**), again confirming the feedback co-regulatory loop of TP63/SREBF1/KLF5 we established in the original manuscript.

We have incorporated these data into the revised manuscript (**Fig. 5i-j, Supplementary Fig. S4**), and pasted below as **Figure R1** for review convenience.

Figure R1. (A) Growth curves and tumor weight of ESCC xenograft models. **(B)** Paraffin-embedded xenograft samples were stained using hematoxylin-eosin (HE) and immunohistochemistry (IHC) for KI67. Original magnification, 400X; scale bar, 50 μ m. **(C)** Protein expression of KI67 was quantified in each group and plotted as fold-changes relative to the control group. **(D)** and **(E)** mRNA expression of indicated genes in the xenograft tumors quantified by qRT-PCR. *, $P < 0.05$, Mean \pm SD, $n = 3$.

Please see our responses and revision for each of the specific comments below:

Specific points:

- The list of genes in Fig1a should be provided (as supplementary table)

Authors' reply:

As suggested, we have now provided this gene list in **Supplementary Table S1**.

- The GSEA in Fig1e, does not provide sufficient information on the regulation of fatty acid genes; this association should be verified in multiple SCC cell lines after p63 silencing by multiple siRNAs with adequate controls for off target effects. Also, a GSEA with $p=0.048$ is of questionable statistical significance.

Authors' reply:

Following the Reviewer's suggestion, we similarly performed GSEA analysis of RNA-Seq data upon TP63 knockdown in another ESCC cell line, KYSE140. As shown below (**Figure R2**), genes downregulated following the silencing of TP63 were consistently enriched in fatty acid metabolism pathway.

Moreover, our original GSEA results from patient data (**Main Fig. 1c-d**, and **Supplementary Fig. S1b-c**) further supported these *in vitro* analyses. Indeed, genes positively correlated with TP63 expression were significantly enriched in fatty acid metabolism from tumor samples across different cohorts of SCC patients.

Figure R2. GSEA plots showing the enrichment of fatty-acid metabolism pathway in RNA-Seq data upon silencing of TP63 in TE5 and KYSE140 cells. NES, normalized enrichment score.

- Fig1f is too small and it is not clear how the quantification was made; the data, like in other figures, have no adequate statistical analysis and error bars.

Authors' reply:

Following the Reviewer's comment, during the revision we repeated the staining assay of lipid droplet for three additional times. Moreover, we added another independent shRNA sequence targeting TP63 in two ESCC cell lines (TE5 and KYSE150, the same as in the original experiment).

Indeed, the new results (**Figure R3**) are highly consistent with our original finding: silencing of TP63 by two independent shRNAs reduced the total level of lipid droplet staining, suggesting decreased lipid storage. Furthermore, ectopic expression of SREBF1 potentially reversed the reduction of lipid droplet content caused by TP63-depletion, suggesting that SREBF1 functionally mediates the effect of TP63 on fatty-acid metabolism.

We have incorporated these new results into the revised manuscript, adding statistics and error bars, as suggested by the Reviewer.

Figure R3. Stable TP63-silenced TE5 and KYSE150 cells were transfected with plasmids encoding either empty vector (OE EV) or SREBF1 (OE SREBF1) for 48 hours. Lipid droplet was stained and analyzed with confocal microscopy. Fluorescence intensity was quantified in each group and plotted as fold-changes relative to the control group of Scramble+OE EV.

- in Fig1h, relative RNA expression changes (measured as $2^{-\Delta\Delta Ct}$) are unclear. Values for scrambled siRNA control should be equal to 1 (not 0.10) and it is difficult to believe that the observed changes in gene expression are statistically significant (for a difference between 0.02 and 0.01). 0 should be 0 not 0.00 and 0.000. The data for several genes are not there. Importantly, the experiment should be repeated with several SCC cell lines. This is also a major concern for the rest of the paper, which is based on analysis of very few SCC lines, often with p63 silencing by a single si/shRNA.

Authors' reply:

We agree with the Reviewer's comments and accordingly, we have now repeated this experiment in three different ESCC cell lines (TE5, KYSE150, KYSE510) in triplicates. We have also used two independent siRNAs against TP63.

In addition, we have now shown the data as the fold-change relative to the scramble control within each gene, as suggested. Validating our original results, depletion of TP63 by different siRNAs consistently decreased the expression of SREBF1 across three ESCC lines (**Figure R4**).

We have added this new result into the revised manuscript.

Figure R4. TE5, KYSE150 and KYSE510 cells were transfected with either siRNAs (30 nM) targeting TP63 or scramble siRNA for 48 hours. mRNA level of indicated genes were detected by qRT-PCR. Data are expressed as mean \pm SD; n=3; *, $P < 0.05$.

- Fig1i should be moved to supplementary. The data for the transcription factors "screening" are from TCGA. Most of these TFs were barely expressed in ESCC cells, bringing into question the biological significance. Some statements are not supported by the any evidence.

Authors' reply:

Fig.1i has now been moved to **Supplementary Fig. S1e**, as suggested.

The Reviewer also commented that most of these TFs were barely expressed in ESCC. In fact, this is one of the results supporting the significance of SREBF1 in the biology of SCC, because it is well-established that a larger number of TFs show strong cell-type-specific expression pattern^{2,3}. For example, one of our shortlisted TFs, HNF4A, is known to be expressed highly and specifically in gastrointestinal epithelial cells^{4,5}. To illustrate this point, we plotted below the expression level of HNF4A across all TCGA cancer types (**Figure R5**), which confirms the strong gastrointestinal-specific expression pattern of HNF4A.

Therefore, the observation that among all these TFs, only SREBF1 exhibited consistent upregulation across different types of SCCs suggests the biological importance of SREBF1 in SCC.

Figure R5. Box plots showing the mRNA level of HNF4A across different cancer types from TCGA tumor samples. Note that the top 4 cancer types are all gastrointestinal adenocarcinomas, namely colon adenocarcinoma (COAD), esophageal adenocarcinoma (EAC), stomach adenocarcinoma (STAD) and pancreatic ductal adenocarcinoma (PAAD).

- Fig 3a: silencing of KLF5 in KYSE510 does not change levels of SREBF1 or p63, is it not countering one of the main conclusions?

Authors' reply:

Prompted by the Reviewer's comment, during the revision we have repeated these experiments using two independent siRNAs against KLF5 in three different ESCC lines (TE5, KYSE150, KYSE510).

As shown below (**Figure R6**), these new results confirmed that silencing of KLF5 reproducibly down-regulated the expression levels of both SREBF1 and TP63 across three ESCC cell lines at both mRNA (**Figure R6, left panel**) and protein levels (**Figure R6, right panel**).

The figures have been updated accordingly in the revised manuscript.

Figure R6. TE5, KYSE150 and KYSE510 cells were transfected with either siRNAs (30 nM) targeting KLF5 or scramble siRNA for 48 hours. The expression levels of indicated mRNAs and proteins were measured by qRT-PCR (left) and Western blotting (right), respectively. Mean \pm SD are shown, $n=3$. *, $P<0.05$; **, $P<0.01$.

- Fig 3e " knockdown of SREBF1 significantly reduced the expression of TP63 and KLF5 at both mRNA and protein levels across different ESCC cell lines". This is not evident in TE5 cells at mRNA (for both KLF5 and p63) or protein (klf5) levels.

Authors' reply:

Following the Reviewer's question, we have similarly repeated the qRT-PCR experiments using two independent siRNAs targeting SREBF1 in three ESCC lines (TE5, KYSE150, KYSE510). Importantly, our new data validated that silencing of SREBF1 consistently reduced the mRNA expression of both KLF5 and TP63 across three ESCC lines (**Figure R7A**).

For the Western Botting assays, our original experiments already contained two independent siRNAs targeting SREBF1 (**Main Fig. 3e**). The less-than-obvious change of KLF5 protein was due to the over-exposure of the blot. We have now displayed the shorter-exposure version of the same blot, which showed that KLF5 protein was evidently decreased upon silencing of SREBF1 (**Figure R7B**).

Lastly, as shown above, *in vivo* xenograft tumors also consistently showed that SREBF1-inhibition down-regulated the expression of both TP63 and KLF5 (**Figure R7C-D**, which are part of Figure R1C-D shown earlier).

These new data have been incorporated into the revised manuscript.

Figure R7. (A) qRT-PCR and **(B)** Western blotting analyses for TP63 and KLF5 expression upon knockdown of SREBF1 in ESCC cell lines. Mean±SD are shown, n=3. *, $P<0.05$; **, $P<0.01$. **(C-D)** The mRNA expression of SREBF1, TP63 and KLF5 in xenograft tumors was measured by qRT-PCR. *, $P<0.05$, mean ± SD, n=3.

- The in vivo work should provide analysis of proliferation/differentiation markers and be expanded to multiple cell lines.

Authors' reply:

As replied above, during the revision we performed IHC staining of Ki67 protein in xenograft tissues from both KYSE150 and KYSE510 cell lines (**Figure R1B**), which validated our original finding that SREBF1 was required for the growth of SCC xenograft *in vivo*. We have incorporated these data into the revised manuscript (**Supplementary Fig. 4**).

- Fig 3f is too small and cannot be deciphered. What are 1.0 and 1.5 in the panel?

Authors' reply:

We have enlarged this plot in the revised manuscript. "1.0" and "1.5" are the RPM (Reads per million mapped reads) values of the peak signals as measured by ChIP-Seq, which have now been described in the revised figure legend.

- In the text is indicated that, for figure 4b, a shSREBF1 was used while in the figure it is indicated siRNA.

Authors' reply:

Thank you for spotting this typo. It should have been "siRNA" as shown in **Main Fig. 4b**, which has been corrected now.

- Some more information should be provided regarding the doses of Fatostatin used for these experiments (dose/response experiment – literature reference).

Authors' reply:

As suggested, we have provided the detailed information (Page 37, line 937-938) and cited related literatures on Fatostatin in the revised manuscript (Page 11, line 247).

- Typing error in lines 252-253 and in line 208 probably wrong reference to fig2.

Authors' reply:

Thank you again for identifying these errors, which have been corrected the revised manuscript.

- For in vitro proliferation studies (one week assays), how persistent are the gene silencing effects by siRNAs?

Authors' reply:

To address this question, we performed qRT-PCR during the revision to measure the knockdown efficiency on both day-4 and day-6 after siRNA transfection. As shown below (Figure R8), at both time points, both siRNAs were able to maintain the reduction of SREBF1 expression in all three cell lines tested, albeit the efficiency was lower on day-6. Since *in vitro* proliferation assays were performed during a 7-day window, these data demonstrated that the knockdown remained effective for these experiments.

We thank the Reviewer again for your valuable comments and suggestions!

Figure R8. TE5, KYSE150 and KYSE510 cells were transfected with 30 nM siRNAs, and the knockdown efficiency of SREBF1 was detected by qRT-PCR on both day-4 and day-6. *, $P < 0.05$, mean \pm SD, $n = 3$.

Reviewer #2 (Remarks to the Author); expert on lipid metabolism:

The manuscript by Li et al. reports a study investigating the role of the lipogenic transcription factor SREBP1 in regulating gene expression in squamous cancers (SCC). This is based on their initial finding that TP63, a transcription factor (TF) highly expressed in SCC, regulates many genes

involved in fatty acid and cholesterol metabolism. The authors then perform a series of experiments ranging from RNAseq, ChIPs and targeted analyses to show cooperation between TP63, KLF5 and SREBP1 in regulating genes involved in fatty acid metabolism in ESCC. Most experiments are performed in at least two cell lines derived from esophageal squamous cell carcinoma (ESCC). The authors use 4C ChIP to demonstrate that TP63 and KLF5 cooperate to regulate the expression of the SREBF1 gene. They also provide evidence that SREBP1, TP63 and KLF5 occupy the same promoters in ESCC cells. Moreover, gene silencing experiments show that these factors regulate each other by binding to their respective promoter elements.

Using lipidomics and RNAseq data, the authors also show that SREBP1 controls the expression of genes involved in fatty acid, glycerophospholipid and sphingolipid metabolism. Moreover, they show that SREBP1 expression is enhanced in human ESCC compared to normal tissue and that silencing of SREBP1 using shRNA or inhibition of SREBP activation using the SCAP inhibitor fatostatin blocks xenograft tumour growth of KYSE150 cells. Moreover, they use ChIPseq analysis to identify genes that are occupied by SREBP1 selectively in ESCC cells and those that are occupied by all three TFs (SREBP1, TP63 and KLF5). They then show that these genes map to ErbB2 and mTOR signalling and determine the regulation of mTORC1 and MEK1/2 signalling by SREBP1 in ESCC cells. From these experiments, the authors conclude that TP63, KLF5 and SREBP1 cooperate to regulate gene expression to drive lipid metabolism and signalling in ESCC.

Overall, the manuscript provides a substantial amount of data from well-executed experiments. These data mostly support the conclusions. However, there are some limitations in the stringency applied in data interpretation, specifically those based on data derived from public sources. This is particularly problematic in Figure 1i and 3h, where only a weak regulation and correlation can be derived. Those weaker data distract from the substantial findings of the manuscript and should be revised. Moreover, the text and figure legends contain a number of inaccuracies that make it difficult to follow the line of argument and assess the strength of the data. In addition, some experiments are not very well described and some conclusions not fully supported by the data.

The weakest part of the manuscript is the analysis of mTORC1 and MEK regulation downstream of SREBP1 in ESCC cells. Regulation of mTORC1 signalling by fatty acid metabolism has already been demonstrated (for example see Menon et al. JBC 2017 or Madak-Erdogan et al. Cancer Research 2019) and the authors do not provide substantial insight beyond the known findings. The manuscript falls short of identifying the exact mechanism by which SREBP1, TP63 and KLF5 control mTORC1 and MEK1/2. The manuscript also does not formally prove that the regulation of lipid metabolism and/or mTORC1/ErbB signalling downstream of SREBP1 is important to maintain ESCC survival and proliferation, as stated in the diagram shown in Figure 7f.

Nevertheless, the manuscript provides important insight into the cooperation of TP63 and KLF5 in ESCC and will be of substantial interests to the field. Below is a list of points that need to be carefully addressed before the manuscript can be considered for publication.

Authors' reply:

We are abundantly grateful for the Reviewer's meticulous evaluation of our manuscript, and are very delighted that the Reviewer remarks that our work provides important insight and it will be of substantial interests to the field!

We also thank the Reviewer for the valuable comments on the weaknesses, which we have addressed point-by-point below.

General comments:

Size markers should be provided for all western blots.

Authors' reply:

As suggested, size markers have been provided for all western blots in the revised manuscript.

Figure numbers should be matched to the order in which they are mentioned in the text.

Authors' reply:

We have carefully checked and matched all the orderings and citations of the figures in the revised manuscript.

Care should be taken to ensure that the figure legends correctly describe the experiments shown in the figures.

Authors' reply:

We have carefully checked and confirmed figure legends in the revised manuscript.

Methods are quite superficial and should be checked for accuracy. All siRNA and shRNA sequences need to be provided.

Authors' reply:

Following your suggestion, we have added the below information to the Methods section:

- 1) All siRNA, sgRNA, shRNA sequences have been provided in **Supplementary Table S8**;
- 2) Detailed information of the use of chemicals, antibodies and cell culture condition;
- 3) Detailed information of the definition and annotation of super-enhancer;
- 4) Detailed information of the 4C-Seq and ChIP-Seq experiments;
- 5) The method for scoring IHC staining intensity.

Specific comments:

Fig 1f: Efficiency of TP63 k/d and overexpression of SREBP1 (full length or mature?) should be demonstrated and indicators of variance across independent biological replicates and significance of the results should be displayed. If full length SREBP1 is shown, it is unclear how this relates to its transcriptional activity.

Authors' reply:

Following to the Reviewer's suggestion, we first examined the efficiency of TP63-knockdown and SREBF1-overexpression by measuring the mRNA expression of TP63 and SREBF1 using Real-Time Quantitative Reverse Transcription PCR (qRT-PCR). The result showed expected down-regulation of TP63 (by two independent shRNAs) and markedly increase of SREBF1 levels in respective groups (pasted below as **Figure R9** for review convenience) in both cell lines.

The Reviewer also asked about the transcript form of SREBF1 and its transcriptional activity. We ectopically over-expressed the full-length form of SREBF1. To determine the transcriptional activity

of the full-length form, we measured the well-established canonical downstream targets of SREBF1, including FASN and ACSS2^{6,7} (which we also validated in the original manuscript, see **Main Fig. 4a**). Notably, the full-length SREBF1 completely rescued the down-regulation of both FASN and ACSS2 caused by TP63-knockdown (**Figure R9**), validating that it was transcriptionally active.

These data have been added to the revised manuscript (**Supplementary Fig.S1d**).

Figure R9. Stable TP63-silenced TE5 and KYSE150 cells were transfected with plasmids encoding either empty vector (OE EV) or SREBF1 (OE SREBF1) for 48 hours. The mRNA expression levels of indicated genes were measured by qRT-PCR. *, $P < 0.05$, Mean \pm SD, $n = 3$.

Fig. 1h: This display is somewhat confusing. It would be better to display levels relative to control. According to methods, the results are based on semi-quantitative PCR and an absolute comparison of mRNA levels between different genes is not possible using this method. A second RNAi sequence should be shown.

Authors' reply:

We agree with the Reviewer's comment and, as suggested, we have now shown the data as the fold-change relative to the scramble control within each gene (pasted below as **Figure R10**).

Regarding the method of semi-quantitative PCR, we in fact performed Quantitative Real-time Reverse-Transcription PCR (qRT-PCR) in the original experiments; we apologize for the confusion.

In addition, as suggested, during the revision we have used two independent siRNAs against TP63. Moreover, we have now repeated this experiment in three different ESCC cell lines in triplicates (TE5, KYSE150, KYSE510).

Importantly, validating our original results, depletion of TP63 by different siRNAs consistently decreased the expression of SREBF1 across three ESCC lines (**Figure R10**).

We have incorporated this new result into the revised manuscript.

Figure R10. TE5, KYSE150 and KYSE510 cells were transfected with either siRNAs (30 nM) targeting TP63 or scramble siRNA for 48 hours. mRNA level of indicated genes were measured by qRT-PCR. Data are expressed as mean \pm SD; n=3; *, $P < 0.05$.

Fig. 1i: This figure does not show any indicators of significance. This is essential to support the conclusions stated in the text. The upregulation of SREBP1 expression in tumour tissue is rather mild (log₂FC between 0.5 and 1.5). Is this sufficient to conclude that SREBF1 is overexpressed in squamous tumours?

Authors' reply:

Following the Reviewer's comment, we have now provided the statistical test of the changes of these genes. As shown below, SREBF1 was the only gene exhibiting significant up-regulation in all

three types of SCCs (**Figure R11A-B and Table R1**). Moreover, some the *P* values of SREBF1 were highly significant, including the ones in ESCC ($P=8.22E-06$) and LUSC ($P<2.2E-16$, **Table R1**).

The Reviewer also commented that the up-regulation of SREBF1 in SCC tumors was modest, which we agree. However, albeit the mRNA up-regulation was moderate, this change was reproducible in another independent dataset (SRP064894) containing 15 ESCC paired samples⁸ (**Figure R11C**). Moreover, as demonstrated in our original manuscript, the up-regulation of SREBF1 protein was validated by IHC in ESCC tumors compared with nonmalignant samples (**Main Fig. 5a**). Nevertheless, in light of the Reviewer's comment, we have now toned down the conclusion by describing this as a *moderate* upregulation (**Page 8, line 174**).

Table R1. *P* value of the expression changes of each gene between TCGA tumor and nonmalignant samples.

Genes	ESCC	HNSC	LUSC
SREBF1	8.22E-06	0.0028	< 2.2E-16
SREBF2	0.8314	0.6426	9.58E-08
LXRB	0.3764	2.09E-06	6.85E-06
PPARA	0.0187	0.2008	< 2.2E-16
LXRA	0.1545	0.0378	0.0006
PPARG	0.0196	2.38E-06	2.68E-16
PGC1	0.0546	0.0002	1.52E-13
HNF4A	0.0332	0.0004	0.2136

Figure R11. (A) Fold changes of mRNA levels of indicated genes in SCC tumors compared with nonmalignant samples from the TCGA cohorts. *, $P<0.01$. **(B)** and **(C)** Detailed FPKM values of SREBF1 mRNA in ESCC tumors compared with nonmalignant esophageal epithelium from the TCGA and SRP064894 dataset.

Fig. 2d: A positive control for SOX2 ChIP needs to be shown in order to confirm specificity of the antibody.

Authors' reply:

As suggested, we have provided a positive control of SOX2-binding peak at the putative enhancer (as indicated by the co-presence of H3K27Ac and ATAC-Seq peaks) of SOX6 locus (**Figure R12**).

Figure R12. IGV tracks of ChIP-Seq of SOX2 and H3K27Ac as well as ATAC-Seq at SOX6 locus in TE5 cells.

Fig. 2f and 3a: A second sequence targeting KLF5 is needed to substantiate the conclusions drawn from this result.

Authors' reply:

As suggested, during the revision we have utilized two independent siRNAs against KLF5 to repeat these experiments. Indeed, both siRNAs validated our original results that: i) silencing of KLF5 reduced the luciferase activities of SREBF1 promoter and enhancers (**Figure R13A**); and ii) silencing of KLF5 down-regulated the expression of SREBF1 and TP63 at both mRNA (**Figure R13B**) and protein levels (**Figure R13C**).

These data have been updated into the revised manuscript.

Figure R13. (A) Transcriptional activity of SREBF1 promoter and enhancers measured by luciferase reporter assays following KLF5 knockdown in TE5 cells. Mean±SD are shown, n=3; *, P<0.05; **, P<0.01. **(B)** qRT-PCR and **(C)** Western blotting assays for KLF5, SREBF1 and TP63 upon knockdown of KLF5 in three different ESCC cell lines. Mean±SD are shown, n=3; **, P<0.01.

Fig. 3a, b and e (and others): It should be stated which form of SREBP1 (full length or mature) is displayed in all western blot experiments.

Authors' reply:

In **Main Figures** containing SREBF1 Western Blots, we showed its full-length form. However, following the Reviewer's comment, we have now provided both full-length and mature forms of SREBF1 in the **Supplementary Fig. S3d**.

Fig. 3d: Statistical indicators are needed. What is the negative control used here?

Authors' reply:

As suggested, we have now shown the statistical analyses for these results. The negative control was transfection of dCas9/Krab vector alone without sgRNA, which has been indicated in the revised figure legend.

Fig. 3e: The evidence for reduced expression of TP63 and KLF5 following SREBP1 k/d is not particularly strong in TE5 cells. A second siRNA sequence needs to be shown to validate this result, as it forms the basis of the overall conclusions of the work as shown in Figure 3i.

Authors' reply:

Following the Reviewer's comment, we have similarly repeated the qRT-PCR experiments using two independent siRNAs targeting SREBF1 in the same three ESCC lines (TE5, KYSE150, KYSE510). Importantly, our new data validated that silencing of SREBF1 consistently reduced the mRNA expression of both KLF5 and TP63 across three ESCC lines (**Figure R14A**).

For the Western Blotting assays, our original experiments already contained two independent siRNAs targeting SREBF1 (**Main Fig. 3e**). The less-than-obvious change of KLF5 protein in TE5 cells was due to the over-exposure of the film. We have now displayed the shorter-exposure version of the same blot, which confirmed that KLF5 protein was evidently decreased upon silencing of SREBF1 (**Figure R14B**).

Lastly, during the revision we performed qRT-PCR experiments using *in vivo* xenograft tumors, which also consistently showed that inhibition of SREBF1 down-regulated the expression of both TP63 and KLF5 (**Figure R14C-D**).

These new data have been incorporated into the revised manuscript (**Supplementary Fig. S4**).

Figure R14. (A) qRT-PCR and (B) Western blotting analyses for TP63 and KLF5 expression upon knockdown of SREBF1 in ESCC cell lines. Mean±SD are shown, n=3. *, $P<0.05$; **, $P<0.01$. (C-D) The mRNA expression of SREBF1, TP63 and KLF5 in the xenograft tumors were measured by qRT-PCR. *, $P<0.05$, mean ± SD, n=3.

Fig. 3g: The control is strongly affected by SREBP1 k/d making the results inconclusive. This needs to be addressed. The figure legend also contains errors.

Authors' reply:

We agree with the Reviewer's comment. During the revision and we have repeated this experiment for additional three times, which validated our original results (Figure R15). In addition, the errors in the figure legend have been corrected. Thank you for identifying these.

Figure R15. Luciferase reporter assays were performed after SREBF1 knockdown in TE5 cells to measure the transcriptional activities of indicated regulatory elements. Mean±SD are shown, n=3. *, $P<0.05$; **, $P<0.01$.

Fig. 3h: The correlations between SREBF1 and TP63 and KLF5 shown here are rather weak (R values around 0.3). This can only be considered as a “weak correlation” and it is not clear how this finding supports the overall conclusions.

Authors’ reply:

We agree with the Reviewer that the Pearson correlations between the expression of SREBF1 and that of TP63 and KLF5 were modest, albeit statistically significant. We reasoned that the moderate correlation might be due to that these bulk RNA-Seq data contained a variety of stromal cells (e.g., immune cells and fibroblasts), which have low expression of the three SCC TFs.

As mentioned above, during the revision we identified another public transcriptome dataset (SRP064894) containing 15 ESCC samples⁸. Importantly, we observed the positive and significant correlations between SREBF1 and TP63 as well as KLF5 (**Figure R16**), highlighting the reproducibility and robustness of the correlation between these TFs.

Regarding the Reviewer’s query that how this finding supports the overall conclusions, we suggest that these significant correlations further support our *in vitro* findings that SREBF1/TP63/KLF5 co-regulate the transcription of each other and form an inter-connected feedback loop (**Main Fig. 3i**). Nevertheless, in light of the Reviewer’s comment, we have now toned down the conclusion by describing this as a modest correlation (**Page 11, line 235**).

Figure R16. Pearson correlation coefficient between the mRNA levels of SREBF1, TP63 and KLF5 in the SRP064894 dataset.

Fig. 4b: Was siRNA or shRNA used in this experiment? There is a discrepancy between text and figure.

Authors’ reply:

Thank you for spotting this typo. It should have been “siRNA” as shown in **Main Fig. 4b**, which has been corrected now.

Fig 4e: Could a different colour code be used rather than shades of grey?

Authors’ reply:

As suggested, we have changed the colour theme to violet.

Fig. 4f: The RNAseq data displayed in this figure require further explanation in the text. Which comparison was made here (control vs fatostatin or SREBP1 silencing vs control)? This is important as fatostatin also blocks SREBP2. Abbreviations should be explained in the legend.

Authors' reply:

The comparison was made between SREBF1-knockdown and scramble-control cells. We have now clearly stated this in the revised figure legend.

The details of the RNA-Seq data have now been provided in the **Supplementary Table S3**. As suggested, we also explained the abbreviations in the revised figure legend.

Fig. 5b: The increased expression of SREBF1 in ESCC looks interesting and convincing. However, in contrast to the staining for TP63 and KLF5, SREBF1 seems to be present in both cytoplasm and nucleus. It could be interesting to score nuclear and cytoplasmic staining separately, which could make the difference between normal and tumour tissue more stringent. Scale bars need to be labelled.

Author's reply:

We agree with the Reviewer's suggestion. Accordingly, we scored the IHC staining separately in the cytoplasm and nucleus in each sample. Indeed, the IHC score of cytoplasm SREBF1 protein was slightly higher than that of TP63 and KLF5 (**Figure R17A**). However, nuclear staining of SREBF1 was significantly stronger than its cytoplasmic staining (**Figure R17A**).

Importantly, the nuclear staining of SREBF1 protein was markedly more intense in ESCC tumors than in normal samples (**Figure R17B**). Moreover, only nuclear (but not cytoplasmic) SREBF1 protein level was significantly associated with worse survival of ESCC patients (**Figure R17C**), strongly suggesting that the nuclear expression of SREBF1 is functionally relevant for ESCC biology.

We have updated this result in the revised manuscript. In addition, scale bars have now been added in **Main Fig. 5b**. We thank for the Reviewer's suggestion, which has improved our data.

Figure R17. (A) Box plots of IHC scores for nucleus and cytoplasmic staining of SREBF1, TP63 and KLF5 proteins. **(B)** Box plots of IHC scores for nuclear staining of SREBF1, TP63 and KLF5 proteins in ESCC tumor and nonmalignant esophagus samples. **, $P < 0.01$. **(C)** Kaplan-Meier analyses of ESCC patient survival stratified by the expression of either nucleus or cytoplasm SREBF1 protein.

Fig. 6a: Given the substantial cell line specificity of the SREBP1 binding peaks identified in HepG2 and MCF7 cells, peaks that are specific to KYSE150 and TE5 cells should also be displayed. This would allow an estimation whether ESCC cell lines show a higher overlap in occupancy compared to cell lines derived from other tissues. Ideally, statistical methods should be employed to calculate the similarity between the two ESCC cell lines.

Authors' reply:

As suggested, we have now provided SREBP1-binding peaks which were specific to KYSE150 and TE5 cells (**Figure R18A**).

Moreover, during the revision we formally tested the SREBP1-occupancy overlap between any two of the four cell lines. Importantly, the overlap between the two ESCC cell lines (~25%) was markedly higher than any other cell types (**Figure R18B-C**), supporting our finding of the cell-type-specificity of SREBP1 occupancy.

These data have been incorporated into the revised manuscript (**Supplementary Fig. S5**).

Figure R18. (A) Heatmaps of ChIP-Seq signals at SREBP1-peak regions (± 3 Kb of peak center), rank ordered by the intensity of peaks based on reads per million mapped reads (RPM). Lines, peaks; color scale of peak intensity is shown at the bottom. **(B)** Overlap of SREBP1-binding peaks between any two of the 4 cell lines. **(C)** The ratio of overlapped SREBP1-binding peaks overall all peaks between any two cell lines.

Fig. 6f: It is surprising that the GSEA shown in here only shows moderate correlation between the 473 ESCC specific genes and genes regulated by silencing of SREBP1 in TE5 cells (NES -1.24, $p=0.04$). The same applies to the silencing of TP63 (NES -1.14, $p=0.06$). Would it make more sense to conduct this analysis on the 274 triple-binding genes? Care should be taken to not overstate the conclusions from this result.

Authors' reply:

The Reviewer's suggestion is more thoughtful and reasonable than our original approach. Accordingly, we re-performed this GSEA analysis using the genes associated with the 274 trio-binding sites as suggested.

Notably, both the P values and normalized enrichment scores (NES) were substantially improved in the new results compared with our original data (**Figure R19**).

We thank again for the Reviewer's great suggestion, which has further strengthened our results!

Figure R19. GSEA plots showing the enrichment of the genes assigned to the 274 trio-binding peaks in RNA-Seq data upon silencing of either SREBF1, KLF5 or TP63 in TE5 cells. NES, normalized enrichment score.

Fig 7a and b: It would be helpful to indicate the size of the pathway (i.e. number of genes in the pathway) rather than just the number of occupied genes.

Authors' reply:

As suggested, we have now provided the size of the pathways in the revised **Main Fig. 7a-b**.

Fig. 7b: It is unclear whether the analysis was performed using the 473 ESCC specific genes, as indicated in the text and legend, or the 274 genes co-occupied by SREBP1, TP63 and KLF5, as indicated in the figure. The second analysis would be far more meaningful.

Authors' reply:

Again, we agree with the Reviewer's insightful suggestion on the selection of the gene set. Accordingly, we repeated the pathway enrichment analysis using the genes associated with 274 peaks trio-occupied by SREBP1, TP63 and KLF5. Importantly, the result was largely consistent with our prior data, with both ErbB and mTOR pathways highly enriched.

We have updated this result in the revised **Main Fig. 7b**.

Fig. 7c: It needs to be explained what exactly is displayed in the heatmap. Fold change over control siRNA? How much siRNA was used? How long was the silencing? There is very little information provided in the methods. Moreover, based on the results of this figure, the authors conclude that genes of the mTOR and ErbB signalling pathways are downregulated by silencing of SREBP1, TP63 or KLF5. What is the cut-off used to identify those downregulated genes? This is difficult to discern from the heatmap. It would be better to apply statistical methods to analyse if triple occupied genes are more likely to be downregulated by silencing of one or all of these TFs. This is crucial as some genes are only moderately or not at all downregulated upon silencing. However, the final conclusion of the manuscript, also outlined in the diagram in Figure 7f, strongly depends on the regulation of these genes by all three TFs. In its current form, this only represents an incidental observation.

Authors' reply:

The Reviewer first commented that more details are needed for the interpretation of **Main Fig. 7c**. We thus added the following information in the revised figure legend:

“**Main Fig. 7c** shows the fold changes (knockdown v.s. scramble control) of mRNA levels of indicated genes upon the silencing of either SREBF1, TP63 or KLF5 in three cell lines. In each group, 30 nM siRNA was transfected; after 48 hours, total RNA was extracted for the measurement the mRNA expression by qRT-PCR.”

Moreover, following the Reviewer’s comment, during the revision we have repeated this experiment for 3 additional times in three ESCC cell lines. As shown in **Figure R20**, among the 13 trio-occupied genes enriched in these two pathways, a total of 10 genes were significantly downregulated by all three TFs in at least two cell lines (defined by fold change >25%, *P* value <0.05; highlighted by red font).

We have now provided both the heatmap and bar plots in the revised manuscript.

Figure R20. TE5, KYSE150 and KYSE510 cells were transfected with 30 nM siRNA targeting either SREBF1, TP63 or KLF5 for 48 hours, and were then subjected to qRT-PCR experiments. Barplots displaying the fold changes relative to scramble control. Mean±SD are shown, n=3. *, *P*<0.05. N.D., not detectable.

Fig. 7e: The results on MEK are difficult to interpret as there are effects on total MEK1/2 levels. The WB results should be quantified and normalised to GAPDH or total proteins for phospho-analysis.

Authors' reply:

Indeed, as shown in the bar-plots above (**Figure R20**), the mRNA expression of MEK2 (encoded by gene *MAP2K2*) is under regulation of the three TFs (SREBF1, TP63 and KLF5). In fact, we initially identified *MAP2K2* because it was one of the 274 genes trio-occupied by three TFs, and it was enriched in the ErbB and mTOR pathways. As shown below using our ChIP-Seq data, *MAP2K2* was evidently co-bound by the three TFs (**Figure R21**). Therefore, data from the ChIP-Seq, qRT-PCR and Western blotting all demonstrate that the expression of *MAP2K2* is co-regulated by SREBF1, TP63 and KLF5.

As suggested, the WB results have been quantified and normalized to GAPDH.

Figure R21. IGV tracks of ChIP-Seq signals of indicated TFs and histone modifications at *MAP2K2* gene locus in ESCC cell lines.

Fig. S2: Statistical indicators should be shown.

Authors' reply:

We have added statistical indicators accordingly.

Comments to text:

Line 76: Please provide reference for this statement.

Authors' reply:

We have provided references for this statement in the revised manuscript (**Page 4, line 76**).

Line 196: Where are the data on SOX2?

Authors' reply:

In our original manuscript, we did not show the result upon SOX2 knockdown since it caused no change in the expression of SREBF1. We have now pasted this result below (**Figure R22**).

Figure R22. qRT-PCR results of the mRNA expression of SOX2 and SREBF1 upon the silencing of SOX2 in TE5 and TE7 cells. Mean±SD are shown, n=3. *, $P < 0.05$.

Line 250: Fatostatin is not a specific SREBP1 inhibitor. It prevents processing of both SREBP1 and SREBP2.

Authors' reply:

Indeed, Fatostatin prevents SCAP-mediated escort of either SREBP1 or SREBP2 to the Golgi. We have now clearly stated this in the revised manuscript (**Page 11, line 247**).

Line 257: Typo in cell line name.

Authors' reply:

We have confirmed the correct cell line name.

We thank the Reviewer again for your insightful comments and suggestions!

Reviewer #3 (Remarks to the Author); expert on transcriptional regulation:

Lin and colleagues provide strong data to support a regulatory and potential function relationship between SREBF1, TP63 and KLF5 in ESCC. They carry out a thorough characterization of the transcriptional landscape of these factors and propose how they may cooperate to regulate both lipid metabolism selective pathways including ErbB/mTOR signaling in this context. Some clarifications and a more direct demonstration of the role of p63 in lipid metabolism would further strengthen the manuscript.

Authors' reply:

We are truly delighted that the Reviewer remarks that our data is strong and our characterization is thorough!

Specific Comments:

Fig. 1f. It would be helpful to have a direct comparison of p63 knockdown to SREBF knockdown.

Authors' reply:

We agree with the Reviewer's suggestion. Accordingly, during the revision we silenced either TP63 or SREBF1 using two independent siRNAs for each gene in two ESCC cell lines (TE5 and KYSE150, the same as in our original manuscript).

As shown below (pasted below as **Figure R23** for review convenience), we first validated our original result that depletion of TP63 decreased the total level of lipid droplets. Moreover, this effect was comparable with that caused by SREBF1-knockdown, supporting our conclusion that TP63 controls lipid metabolism through the regulation of SREBF1.

We have added this new piece of data into the revised manuscript (**Supplementary Fig. S3a**)

Figure R23. TP63 and SREBF1 were silenced individually in TE5 and KYSE150 cells and lipid droplet staining was analyzed with confocal microscopy. Fluorescence intensity was quantified and shown in the bar plots. Mean±SD are shown, n=5. *, $P<0.05$.

1h/i It seems surprising that the other factors shown are suppressed/not expressed in tumors. Can the authors comment?

Authors' reply:

It is an interesting note but not entirely unexpected. This is because the 8 proteins in **Main Fig. 1h** are all transcription factors (TFs). As a gene class, TFs have been shown to exhibit strong cell-type-specific expression pattern^{2, 3}. For example, HNF4A is known to be expressed highly and specifically in gastrointestinal epithelial cells^{4, 5}. Indeed, when we plotted the expression level of HNF4A across all TCGA cancer types, the strong gastrointestinal-specific expression pattern of this TF was evident (**Figure R24**).

Therefore, the observation that, among these 8 TFs, only SREBF1 exhibited consistent upregulation across SCCs suggests the biological significance of SREBF1 in SCC.

Figure R24. Box plots showing the mRNA level of HNF4A across different cancer types from TCGA tumor samples. Note that the top 4 cancer types were all gastrointestinal adenocarcinomas, namely colon adenocarcinoma (COAD), esophageal adenocarcinoma (EAC), stomach adenocarcinoma (STAD) and pancreatic ductal adenocarcinoma (PAAD).

Fig. 2e. These data are interesting but do not prove a direct effect of p63. Can the authors demonstrate effects on an enhancer in which the canonical p63 binding site is mutated?

Authors' reply:

Following the Reviewer's suggestion, during the revision we selected the enhancer E3 as an example and performed site-directed mutagenesis to measure the direct regulatory effect of TP63 on its targeting enhancer.

Specifically, we first located the motif-sequence of TP63 on E3 enhancer by motif analysis (**Figure R25A-B**). We then mutated the bases of "CTTG" to "TGCA", and validated by Sanger sequencing (**Figure R25A**). Both wild-type and mutant E3 enhancers were next cloned into the pGL3-promoter vector. Importantly, luciferase reporter assay showed significantly decreased reporter activity of mutant E3 compared with its wild-type counterpart (**Figure R25C**), demonstrating the direct regulation of TP63 on this enhancer element.

Figure R25. (A) Sanger sequencing of both wild-type and mutant E3 enhancers. The TP63 motif was highlighted by box and the mutated bases were highlighted by red font. (B) The logo plot of TP63 canonical motif sequence. (C) TE5 cells were co-transfected with both the renilla plasmid and pGL3-promoter plasmid containing either wild-type or mutant E3 enhancers. The relative transcriptional reporter activity was analyzed by dual luciferase reporter gene assay. Mean±SD are shown, n=4. *, $P < 0.05$.

Fig. 3g. Similar to 2e, can the authors demonstrate effects on an enhancer in which the SREBF binding site is mutated? That would provide evidence of a direct effect.

Authors' reply:

Here we similarly selected T2 and T3 enhancers as examples and performed site-directed mutagenesis to measure the direct regulatory effect of SREBF1 on its targeting enhancers.

Specifically, we first located the motif-sequence of SREBF1 on T2 and T3 enhancers by motif analysis (**Figure R26A-B**). We then mutated the bases of "GTGA" to "TGTC" in T2 and T3 enhancers, which were validated by Sanger sequencing (**Figure R26A**). Both wild-type and mutant T2 and T3 were then cloned into the pGL3-promoter vector. Importantly, luciferase reporter assay showed significantly decreased reporter activity of mutant compared with wild-type enhancers (**Figure R26C**), demonstrating the direct regulation of SREBF1 on these enhancers.

Figure R26. (A) Sanger sequencing of both wild-type and mutant T2/T3 enhancers. The SREBF1 motif was highlighted by box and the mutated bases were highlighted by red font. (B) The logo plot of SREBF1 canonical motif sequence. (C) TE5 cells were co-transfected with both the renilla plasmid and pGL3-promoter plasmid containing either wild-type or mutant T2/T3 enhancers. The relative reporter activity was analyzed by dual luciferase reporter gene assay. Mean±SD are shown, n=4. *, $P < 0.05$.

Fig. 4d/e These findings are interesting and consistent with the established function of SREBF. However, a major claim of the paper is that p63 itself through its regulation of SREBF controls lipid metabolism. It would add substantially to the paper if lipidomics could be performed in the setting of p63 knockdown and compared to the results shown.

Authors' reply:

As suggested, during the revision we performed liquid chromatography tandem mass spectrometry (LC-MS/MS)-based lipidomics in the presence and absence of TP63-knockdown in KYSE510 cells in triplicates.

We then compared the new lipidomic results with the original data generated after inhibition of SREBF1 (by Fatostatin) in the same KYSE510 cells. Firstly, we identified a total of 1,734 lipid ions, which is similar with the number we originally identified (n=1,561), again confirming high coverage of lipidome by this approach and the stability of our system.

Globally, compared with SREBF1-inhibition, shTP63 also caused comparable degree of alterations in the lipidome. Specifically, SREBF1-inhibition led to a decrease of 4.4% and increase of 4.2% lipid ions, while silencing of TP63 reduced 5.7% and increased 10.4% (**Figure R27A**).

At the lipid class level (**Figure R27B**), TP63-knockdown also resembled SREBF1-inhibition to certain extent. For example, both shTP63 and SREBF1-inhibition altered a large number of lipid ions in classes such as PE, PC, TG and DG. Also, both shTP63 and SREBF1-inhibition decreased multiple lipid species in CerG1, CerG2 and PS (**Figure R27B**; for the purpose of direct comparison, we ordered the lipid classes along the X axis in an identical way between up and lower panels).

However, there were also differences in these two experiments. In particular, SREBF1-inhibition reduced multiple lipid ions in Cer, So, PG classes; whereas shTP63 either did not change (Cer, So) or triggered a more balanced alterations (PG).

We reasoned that a few confounding factors could cause some of the discordances between these two lipidomic datasets:

i) As a master regulator of SCC biology, TP63 controls thousands of enhancers and super-enhancers in SCC cells^{9, 10}. However, only a small subset of these regions were co-occupied by SREBF1 (n=274, **Main Fig. 6**). Therefore, TP63-knockdown may cause a large number of unique changes in the transcriptome, which can indirectly alter the lipidome.

ii) Unlike SREBF1-inhibition, silencing of TP63 by shRNAs causes significant apoptosis of SCC cells, which likely leads to nonspecific changes in the lipidome.

iii) We used Fatostatin to inhibit SREBF1 and performed lipidomics. However, like almost all small-molecule inhibitors, Fatostatin also has non-specific targets such as SREBF2¹¹. These could also cause changes in the lipidome.

Nevertheless, despite these strong confounding factors, we still obtained a partially comparable lipidomic results following either SREBF1-inhibition or TP63-knockdown, suggesting that TP63 regulates the lipid metabolism partially through SREBF1.

Figure R27. (A) Pie charts showing the fractions of altered and unaltered lipid ions in KYSE510 cells following either SREBF1-inhibition or TP63-knockdown. **(B)** Scatter plots of significantly changed lipid ions grouped by lipid classes upon either SREBF1-inhibition or TP63-knockdown. The order of lipid classes along the X axis is identical between up and lower panels for the purpose of direct comparison. Each dot is one lipid ion. SL, sphingolipids; GPL, glycerophospholipids, GL, glycerolipids.

Fig. 5a It is not too surprising that the levels of these proteins are higher in tumor than in normal. What we would actually like to know is whether the levels are correlated in individual tumors. Specifically, the correlation coefficients for IHC scores should be depicted in the same way that the RNA expression correlations are depicted in 3h.

Author's reply:

We agree with the Reviewer's constructive advice. In the meanwhile, *Reviewer #2* also suggested that we better separately score the staining intensity in the cytoplasm and nucleus (See **Figure. R17** on **Page 20** in this Response Letter). Accordingly, we first scored the staining intensity separately in the cytoplasm and nucleus in each sample. As expected, all three TFs showed much higher nuclear score than cytoplasmic score (**Figure R28A**).

Importantly, the nuclear staining of SREBF1 protein was more intense in ESCC tumors than in normal samples (**Figure R28B**). Moreover, only nuclear (but not cytoplasmic) SREBF1 protein level was significantly associated with worse survival of ESCC patients (**Figure R28C**), strongly suggesting that the nuclear expression of SREBF1 is functionally relevant for ESCC biology.

As suggested by the Reviewer, we next used either cytoplasmic or nuclear scores to perform Pearson correlation analysis. We observed positive, albeit weak, correlations between the nuclear staining of the three TFs (**upper panel of Figure R28D**), supporting our original results obtained from mRNA analysis (**Main Fig. 3h**). These correlations were statistically significant for both SREBF1 vs. KLF5 ($P=0.04$) and TP63 vs. KLF5 ($P<2.2E-16$), and borderline for SREBF1 vs. TP63 ($P=0.06$).

In contrast, no correlation was found in the cytoplasmic levels between SREBF1 vs. KLF5 and even a negative correlation was observed between SREBF1 vs. TP63 (**lower panel of Figure R28D**), confirming the specificity of the correlation in the nucleus. In addition, although TP63 and KLF5 were correlated in the cytoplasmic levels, the correlation coefficient ($R=0.36$) was much lower than that in the nucleus ($R=0.64$).

Figure R28. (A) Box plots of IHC scores for nucleus and cytoplasmic staining of SREBF1, TP63 and KLF5 proteins. **(B)** Box plots of IHC scores for nuclear staining of SREBF1, TP63 and KLF5 proteins in ESCC tumor and nonmalignant esophagus samples. **, $P < 0.01$. **(C)** Kaplan-Meier analyses of ESCC patient survival stratified by the expression of either nucleus or cytoplasm SREBF1 protein. **(D)** Pearson correlation coefficient between the nucleus (upper panel) or cytoplasm scores (lower panel) of SREBF1, TP63 and KLF5 in 177 ESCC tumors.

Fig. 7b Are the results the same if one looks at genes that are both associated with ESCC-specific peaks and also significantly regulated by these factors (i.e. as defined by p63 and KLF5 knockdown studies)?

Authors' reply:

Following the query, we performed similar pathway enrichment analysis using the gene set as suggested. Specifically, we first determined that among 376 genes associated with ESCC-specific peaks, 189 genes were downregulated (defined as fold change >2, $P < 0.05$) upon knockdown of either SREBF1, TP63 or KLF5, which was highly significant ($P = 5.3E-20$, hypergeometric test).

Importantly, pathway enrichment analysis of these 189 genes showed highly similar results with our initial finding. Overall, six out of nine pathways were reproducibly identified (**Figure R29A**). Moreover, both ErbB and mTOR signaling pathways, which were highlighted and focused in the original manuscript, were consistently enriched (**Red font, Figure R29B**).

Figure R29. (A) Venn diagram showing the overlap between the KEGG pathway enrichment analysis using either genes with ESCC-specific peaks (**Original**) or genes both with ESCC-specific peaks and regulated by either TFs (**New**). **(B)** Significantly enriched pathways of the 189 genes. Overlapped pathways are highlighted by either red or blue font.

We thank the Reviewer again for these insightful comments and suggestions!

Reviewer #4 (Remarks to the Author); expert on superenhancers and 4Cseq:

NCOMMS-20-05458 from Li et al. is a well-written manuscript that identifies an important finding for an integrated set of key transcription factors that affect 2 functional pathways in squamous cell carcinoma (SCC). The major claims of the paper for SCC relevance are the 1) functional significance of the fatty acid metabolic pathway that is regulated by the well-established SCC driver TP63 and mediated by SREBF1 TF-driven epigenetic regulation and 2) complex co-regulatory feedback loop for SREBF1, TP63, and KLF5 to further drive SCC formation and for additional identification of the mTOR/ERBB pathway.

The discoveries were rigorously determined using GSEA analyses of comprehensive TCGA in vivo datasets, bioinformatics analyses to identify enrichment of SREBF1 motifs in fatty acid pathway genes, multiple functional experiments using gene knockdown (siRNA), epigenetic ChIP, 4C-seq, gene expression, epigenetic repressed (CRISPRi), cellular assays for lipid formation and migration and evident in replicates, knockdown xenograft model in mice, lipidomics, and human SCC immunohistochemistry. The claims are novel and are of great interest to the cancer field and genomics scientific community as the findings for fatty acid/lipid and mTOR/ERBB pathways and SREBF1/TP63/KLF5 TF feedback loop and demonstrated to be functionally relevant have not been reported for SCC and further illustrates how the multi-omics approach have facilitated these important discoveries.

The work is for the most part convincing as mentioned previously given the multiple approaches to link the genomics findings to pathways that functionally impact SCC development. However, there are major concerns for claims for the superenhancer, 4C studies, and effects on SL synthesis. Details for superenhancer and 4C-seq methodologies and analyses are completely missing and hence their interpretations are difficult to assess and to justify their corresponding claims. The effects of SL synthesis are not completely clear as validation experiments were not performed to assert that SREBF1-regulated enzymes affect this pathway. The conclusions are original except for the assertion that inhibition of the superenhancer for SREBF1 resulted in decreased TP63 and KLF5 which is not substantiated as the annotation for superenhancer are not clear. It would be nice to include a few details about how this new knowledge could open up new therapeutic strategies to target these pathways.

Overall, the manuscript is impactful and it very nicely establishes the SREBF1/TP63/KLF5 regulatory feedback loop and shows how they contribute to cancer associated signaling pathways specifically in SCC associated with known poor survival. The work is extensive and nicely combines both in vivo and in vitro data to convincingly illustrate their findings. However, major concerns are noted and additional comments are provided below.

Authors' reply:

We are truly pleased that Reviewer considers our study rigorous, important and impactful, and further remarks that our work is extensive and nicely combines both *in vivo* and *in vitro* data to convincingly illustrate the findings!

More importantly, we are extremely appreciative for the Reviewer's suggestions and comments, which we have addressed point-by-point as detailed below:

Major concerns are the lack of details for superenhancer annotation and 4C-seq methodology, downstream bioinformatic analyses, sequences for bait. Which cells? Which antibody for superenhancer? How many reads? How many replicates? Filtering criteria? etc.

Authors' reply:

We sincerely apologize for omitting these sections of methods, which have now been added to the revised manuscript (**Page 24, Line 534 to Page 25, Line 565**). We have pasted below for review convenience:

Circular Chromosome Conformation Capture Sequencing (4C-seq) Assay

4C-seq experiments were performed as previously described by us^{1, 12, 13} and others¹⁴ with slight modifications. Firstly, 40 million TE5 cells were cross-linked with 1% formaldehyde for 10 min and quenched with 2.66 M glycine for 5 minutes. Nuclei pellets were isolated with lysis buffer on ice

containing protease inhibitors. Nuclei were re-suspended in 1x CutSmart buffer (NEB) containing 0.3% SDS and incubated at 37°C for 1hour. Next, 10% Triton X-100 was added to a final concentration of 2% and samples were incubated at 37°C for 1 hour. Nuclei were digested by 4,000 U HindIII (NEB) at 37°C overnight. Enzymatic digestion efficiency was assessed by gel electrophoresis. HindIII digestion was quenched by 10% SDS (to a final concentration of 1.6%) and incubated at 65°C for 25 minutes. 100 µg digested DNA was used for ligation with 1% Triton X-100 and 990U T4 DNA ligase (Thermo-Fisher Scientific) at 16°C overnight. Ligation reaction was then reverse cross-linked by proteinase K (Thermo-Fisher Scientific) digestion at 65°C for 4 hours and incubated at 37°C overnight. Ligation products were purified by phenol-chloroform and ethanol precipitation to produce the “3C libraries”. The 3C libraries were then digested by a 500 µl DpnII reaction: 50 µg 3C library, DpnII buffer (NEB) and 50 U DpnII (NEB) overnight at 37°C, which was followed by phenol-chloroform purification and ethanol precipitation and DNA concentration evaluation. 4C-seq libraries were then prepared by Illumina Nextera kits with two rounds of PCR reactions. Libraries were pooled followed by Agilent high sensitivity DNA kit. DNA Sequencing was performed in MiSeq platform using Illumina 150 bp pair-end kit. Two replicates were generated and sequencing results were analyzed by r3Cseq package¹⁵.

The bait of SREBF1 promoter region was chr17:17738740 - 17740085 with 4C primers: NEST-Forward-GCAACCAGCTGGGCTCAT, OUTER-Forward-GAAGCAACGGGCCTCCTAAT, NEST-Reverse-CTGCTGACCGACATCGAAGG and OUTER-Reverse-TTGCGAGGTTACTCACGGTC.

Computational annotation of typical-enhancer and super-enhancer

The method of ROSE (Rank Order of Super Enhancers)^{13, 16} was first used to identify enhancers defined as H3K27Ac peaks 2 kb away from any transcription start site (TSS). Following stitching enhancer elements clustered within a distance of 12.5 Kb, typical-enhancers and super-enhancers were then classified using a cutoff at the inflection point (tangent slope =1) based on the ranking order. The scatterplots (**Main Fig. 2b**) contained all typical-enhancers and super-enhancers from each sample.

For the detailed methods of ChIP-Seq and bioinformatics analyses, please refer to our original manuscript (**Page 23, line 503 to Page 24, line 532**).

1. Page 3, lines 67-68 – reference for Warburg effect?

Authors' reply:

As suggested, we have added the reference¹⁷ for Warburg effect in the revised manuscript.

2. Page 6, lines 124- 125 – included references do not mention HPV+ tumors in HNSC as different. Need another reference. Also spell out HPV.

Authors' reply:

Accordingly, we have now cited two papers^{18, 19} which compared HPV⁺ vs HPV⁻ HNSC and showed the differences in gene expression programs.

We have also spelled out HPV in the text.

3. Page 6, lines 130-131 - is the $P < 1E-06$ cutoff used to determine significance for all DE genes? This is not listed in figure legend or methods

Authors' reply:

Here " $P < 1E-06$ " indicates the highly significant overlap of the differentially-expressed genes (DEGs) between different SCC types. For example, 72.8% (182/250) of genes in TP63-high samples identified in ESCC were shared with either LUSC ($P < 1E-06$) or HNSC ($P < 1E-06$) cohorts. This high degree of overlapping is supportive of the notion that TP63 has shared biological functions in different types of SCCs.

On the other hand, the statistical cutoff we used to identify differentially-expressed genes was $|\text{Log}_2\text{FC}| > 2$ and q -value < 0.05 (after multiple-testing correction), which have now been added to the revised figure legend.

4. Page 7, lines 137-139 - estrogen response late pathway is enriched in all 3 cancer types but is the only one out of the 9 that was not commented on in the manuscript and should be mentioned.

Authors' reply:

We added estrogen response late in the revised manuscript in **Page 6, line 138**.

5. Figure 1c: when was TCGA accessed? Text (line 141) says all cancer types were analyzed, but Fig 1c only shows 23 types and TCGA currently lists 32 types. Kindly provide information on release of TCGA that was used.

Authors' reply:

We utilized the TCGA data released on March 26, 2019 (GDC V16.0). Indeed, as the Reviewer pointed out, TCGA had 32 cancer types at the time. However, to ensure the statistical power for GSEA analyses, we required that the numbers of tumor and normal samples were over 70 and 5, respectively. This selection criterion resulted in 23 cancer types as shown in **Main Fig. 1c**. We have updated this in the revised manuscript (**Page 20, Line 444**).

6. Figure 1d. $P=0$ for TCGA ESCC GSEA?

Authors' reply:

Yes, we confirmed that this value is correct.

7. Stats on Figure 1f?

Authors' reply:

As suggested, we have added statistics, including significance and error bars, in the revised figure.

8. Fig 1h – what about effect of TP63 knockdown on E2F and Sox5?

Authors' reply:

To address this question, we measured the mRNA expression of E2F and SOX5, along with other candidates, in TP63-knockdown cell lines (**Main Fig. 1h**), using two different siRNAs against TP63.

As shown below (**Figure R30**), the expression of E2F changed inconsistently between three cell lines upon knockdown of TP63. On the other hand, SOX5 expression was undetectable in any of the three cell lines.

This result has been updated in **Main Fig. 1h** (pasted below as **Figure R30** for review convenience).

Figure R30. TE5, KYSE150 and KYSE510 cells were transfected with either siRNAs (30 nM) targeting TP63 or scramble siRNA for 48 hours. mRNA level of indicated genes were measured by qRT-PCR. Data are expressed as mean \pm SD; n=3; *, P<0.05.

9. Major concern - It is not clear how super enhancers were found for the ESCC lines.

Authors' reply:

As replied above, we have now provided detailed methods for the annotation of super-enhancers as well as ChIP-Seq experiments in the revised manuscript.

Briefly, we performed ChIP-Seq using the H3K27Ac antibody in ESCC cell lines and annotated both typical-enhancers and super-enhancers based the well-established ROSE computational pipeline. In fact, our lab has been utilizing this established methodology to identify super-enhancers in over 100 cell lines as well as fresh tissue samples from difference cancer types^{1, 10, 12, 13, 20-23}.

Moreover, we have also developed computational programs and analyzed super-enhancers from over 500 public samples profiled by H3K27Ac ChIP-Seq²⁴. Together, these works demonstrate our experience and expertise in analyzing and studying super-enhancers.

However, we do apologize again for omitting the detailed methods in the original manuscript.

10. Fig 2b- It is not clear if “all enhancers” here also include super enhancers? Please clarify.

Authors’ reply:

Yes, **Main Fig. 2b** shows both of the classes of typical-enhancers and super-enhancers. We have now clarified this in the revised figure legend.

11. Chr number and genome build for Fig 2 is needed. Where is the bait label for SREBF1 promoter? 4C-seq shows enrichment to 4 regions but the seemingly negative results outside the 4C-seq region annotation are likely false negative.

Authors’ reply:

As suggested, we have now added Chr number and genome build. We have also added the bait label for the *SREBF1* promoter.

Indeed, there were sequencing reads outside the enriched regions, which had lower *q* values (**Supplementary Table S2**), and we have now provided the entire DNA-DNA contact reads detected by 4C-seq as **Supplementary Table S2**.

12. How are the authors determining what is regulating SREBF1? Is it all (P, E1, E2, and E3) that were tested? Is there a rank order or hierarchy of the regulatory elements for activation? More importantly, the claim of TP63 and KLF5 binding to the superenhancer for SREBF1 (discussion) is not substantiated as there is no evidence of their binding in the “control” region and is located within the superenhancer. It’s okay to state binding to several enhancers instead of superenhancer.

Authors’ Reply:

The results from the Luciferase reporter assays (**Main Fig. 3a-b**) and CRISPR-mediated interference (**Main Fig. 3c-d**) suggested that all of the 4 regulatory elements (Promoter, E1, E2, E3) confer activity for the transcription of SREBF1.

The Reviewer next queried whether there exists a rank order or hierarchy of the regulatory elements, which is an interesting question. However, we consider that answering this question will entail future *in vivo* experiments including CRISPR-based activation (by VP64 vector)^{25, 26} of individual element in the presence of the inhibition of other elements (i.e., *in vivo* rescue assays of individual enhancers/promoter). For example, if activation of E1 does not rescue the effect caused by E2-inhibition, then E1 region may function as the primary force for SREBF1 transcription and E2 may be additive.

The Reviewer also commented that it is more appropriate to conclude that TP63 and KLF5 bind to several constituent enhancers rather than super-enhancer, which we agree. We have revised the text accordingly.

13. Page 9, line 202- reference the paper for the H3K27ac ChIP data here (currently listed in the methods section)

Authors' reply:

Accordingly, we have added this reference for the H3K27Ac CHIP data here.

14. Page 9, line 205 – I believe the author is referring to Fig 2e here and not 2d.

Authors' reply:

Yes, it should have been **Main Fig. 3a**. Thank you for finding this typo.

15. Authors should mention that sh knockdown of both p63 and KLF5 affected promoter activity as well and not just 3 enhancers. Is there not another shKLF5 construct? It looks like this is a pooled reagent. If so, please state in figure.

Authors' reply:

Yes, the Reviewer is right that knockdown of TP63 and KLF5 decreased the promoter activity of SREBF1 as well as the three enhancers. We have revised the text accordingly (**Page 9, line 205-207**).

Yes, we used a pooled siRNAs targeting KLF5 in the original manuscript. To confirm this result, during the revision we used two different siRNAs targeting KLF5, and repeated the luciferase reporter assay. As shown below (**Figure R31**), the reporter activities of promoter and three enhancers of SREBF1 significantly decreased after knockdown of KLF5 by either siRNA, validating our original data.

This new piece of data has been updated in the revised manuscript and pasted below as **Figure R31** for review convenience.

Figure R31. TE5 cells were transfected with two different siRNAs targeting KLF5 for 12 hours, followed by the co-transfection of the Renilla vector and the pGL3-based plasmids containing either the promoter or E1/E2/E3 enhancers. The luciferase activity was analyzed by dual luciferase reporter gene assay. Mean \pm SD are shown, n=4. *, $P<0.05$.

16. Fig 3d – should be compared to no dCas9 to rigorously determine relative repression for each regulatory element. At this point and with lack of how negative control is defined, it is not clear if there are significant effects for decreased SREBF1.

Authors' reply:

In our original manuscript, we used the dCas9/Krab vector only (without sgRNA) as the negative control.

Following the comment of the Reviewer, we performed additional experiments wherein two different negative controls were included: i) the dCas9/Krab vector only (without sgRNA) and ii) sgRNA only (without the dCas9/Krab vector, as suggested by the Reviewer).

As shown below (**Figure R32**), validating our original data, transfection of dCas9/Krab with sgRNAs targeting the SREBF1 promoter markedly down-regulated the expression of three TFs. In contrast, neither negative control produced any effect on the expression of any TFs, confirming these results.

Figure R32. qRT-PCR measuring mRNA levels of SREBF1, TP63 and KLF5 following the transfection of either dCas9/Krab alone (without sgRNA), sgRNA alone (without dCas9/Krab), or dCas9/Krab together with sgRNAs targeting the SREBF1 promoter. Mean±SD are shown, n=3. *, $P<0.05$.

17. Fig 3e – TP63 does not look to be significantly decreased with siSREBF1 knockdown in TE5 cells

Authors' reply:

Following the Reviewer's question, we have repeated the qRT-PCR experiments using two independent siRNAs targeting SREBF1 in three ESCC lines (TE5, KYSE150, KYSE510). Importantly, our new data validated that silencing of SREBF1 consistently reduced the mRNA expression of both TP63 and KLF5 across three ESCC lines (**Figure R33A**).

Moreover, during the revision we performed qRT-PCR experiments using *in vivo* xenograft tumors, which also consistently showed that inhibition of SREBF1 down-regulated the expression of both TP63 and KLF5 (**Figure R33C and 3D**).

Figure R33. (A) qRT-PCR for TP63 and KLF5 expression upon knockdown of SREBF1 in ESCC cell lines. Mean \pm SD are shown, n=3. *, $P<0.05$; **, $P<0.01$. **(B-C)** The mRNA expression of SREBF1, TP63 and KLF5 in the xenograft tumors was measured by qRT-PCR. *, $P<0.05$, mean \pm SD, n=3.

18. Fig 3f – 4C contact of SREBF1 promoter with SE of TP63 that is also bound by SREBF1 as determined by CHIP is not clear and well-established. The 4C is likely an artifact in the absence of showing peak height for the 4C experiment but has been determined to be enriched (how?) and yet has also not been validated via reciprocal 4C from TP63 to SREBF1 or by FISH. chr numbers are needed in figure and/or legend.

Authors' reply:

Firstly, there exists a misunderstanding in this query: in our original manuscript, we described that this **Main Fig. 3f** showed the 4C-Seq results using *TP63*-promoter (**not SREBF1-promoter**) as the bait region. Therefore, the enriched regions showed the interaction between *TP63*-enhancers with *TP63*-promoters. We apologize if our original description was not sufficiently clear.

Secondly, as the Reviewer pointed out, we did not show the detailed 4C-seq peaks in **Main Fig. 3f**. This is because we recently published this piece of data, which was utilized to study the functions of enhancers of *TP63*¹. Indeed, in that paper, we performed 4C-Seq to identify all interacting genomic regions with *TP63*-promoter (**Figure R34**).

Importantly, in the present work, we observed that SREBF1 bound to three of the interacting enhancers with *TP63*-promoter as identified by 4C-Seq: T1, T2, T3, which were highlighted in the panel B in the below **Figure R34**.

In short, the 4C-Seq experiment baiting *TP63*-promoter was published and here we only showed the annotated interacting regions for confirmation. Again, we sincerely apologize for omitting this information.

As suggested, we have added chr numbers in the revised figure.

Figure R34. 4C-Seq identifies long-range interactions between TP63-enhancers and TP63-promoter in TE5 cells. (A) ChIP-seq tracks for indicated antibodies surrounding *TP63* gene locus in ESCC, esophageal adenocarcinoma (EAC) and normal esophagus samples (NEM). Super-enhancer regions are depicted as red bars. **(B)** Top 20 genomic loci exhibiting long-range interactions with TP63-promoter in TE5 cells, as identified by 4C-Seq. Vertical and horizontal purple columns indicate interaction density and the distance between interaction loci and TP63-promoter, respectively. **Note that this Figure was published by our group¹ and was pasted here for review purpose.** In the present work, we identified that T1, T2 and T3 regions were bound by SREBF1, and these regions are highlighted here. Pro, promoter.

19. Fig 3h- log is inconsistently capitalized

Authors' reply:

Thank you for spotting this inconsistency. We have corrected it in the revised figure.

20. Figure 4b- text says shRNA, figure marked siRNA

Authors' reply:

Thank you for identifying the typo. It should have been siRNA, and we have revised the text.

21. Figure 4f is a schematic of the pathways their findings seem to suggest, but they do not present the RNA-seq data that they used to create this schematic. Present in supplement

Authors' reply:

As suggested, we have presented the RNA-Seq data as **Supplementary Table S3**.

22. Page 12, line 262 – should reference figures 4d and 4e here.

Authors' reply:

We have added it accordingly.

23. It is unclear why only SPTLC1 is presented in figure 4g as the authors conclude multiple genes are direct downstream targets. Either don't present this data or present this figure for all the implicated downstream targets in the supplement.

Authors' reply:

We agree with the Reviewer, and we have removed this panel as suggested.

24. Does knockdown of SREBF1 affect cell viability and hence to explain decreased luciferase activity (Fig 3g) or even for any gene expression? Another marker for cell viability should be used to demonstrate live cell normalization. The impact to cell growth has also been demonstrated in fig 5d so how do the authors reconcile the perceived decrease in gene expression or luc activity? Indeed, Fig 5b shows co increased expression.

Authors' reply:

We understand the Reviewer's point. However, these changes in cell proliferation did not explain the differences in gene expression and luciferase reporter activity for the following reasons:

1) Silencing of SREBF1 inhibited colony growth (a 7-day assay) but did not affect short-term cell proliferation (a 4-day assay, **Figure R35A**). However, all of the gene expression assays by qRT-PCR and luciferase reporter assays were performed 48 hours after the transfection of siRNAs against SREBF1, when there was no difference in cell proliferation.

2) The luciferase assays were conducted using the Dual-Luciferase® Reporter (DLR™) System, wherein the “dual reporter” refers to the simultaneous expression and measurement of two individual reporter enzymes: “experimental” and “control” reporters. Specifically, the “experimental” Firefly reporter (measuring our E1-E3, T1-T3 or promoter regions) and “control” Renilla reporter were measured simultaneously from the same samples. Thus, the activity of the co-transfected “control” reporter Renilla provides an internal control for cell viability, cell number, transfection efficiency, etc.

Figure R35. Cell proliferation was determined by MTT assays after knockdown of SREBF1 in ESCC cells. Mean±SD are shown, n=3.

25. As SPTLC1, SPTLC2, ELOV4, -6, -7, and CERS6 are target genes and downregulated in Fatostatin-treated cells, are the downstream effects evident with decreased levels of 3kSN, SA+FA, and DHCer lipid intermediates?

Authors' reply:

Our LC-MS/MS-based lipidomics could not detect small lipid intermediates such as 3kSN, SA+FA and DHCer. This is because our LC-MS/MS was conducted by extracting ion signals for lipids with chain lengths ranging from 18 to 91 carbon atoms. Because 3kSN, SA+FA, and DHCer (the precursors of ceramides) belong to small ion class (C6-C20), they are at the borderline of detection threshold in our methodology.

Nevertheless, the downstream lipid classes of 3kSN, SA+FA and DHCer, including Ceramide (Cer), Sphingosine (So), Glucosylceramide (CerG1), Diglucosylceramide (CerG2) and Triglucosylceramide (CerG3) were indeed decreased after inhibition of SREBF1. These changes were highly consistent with the down-regulation of SPTLC1, SPTLC2, ELOV4, -6, -7, and CERS6 (**Main Fig. 4e and Supplementary Table S3**).

26. Figure 5i-j – Representative images for the tumor sizes upon SHSREBF1 and Fatostatin treatments should be included in supplement.

Authors' reply:

As suggested, we have now shown photos of the xenograft tumors as **Supplementary Fig. S4**.

27. Page 14-15, lines 312-324 – not sure what this is adding for cell type specific TFs

Authors' reply:

This section described the result of sequence motif enrichment analysis for identifying putative upstream TFs. Briefly, we first classified SREBF1-binding peaks into either ubiquitous peak-set or cell-type-specific peak-set. We then performed sequence motif enrichment analysis on these peak-sets separately (**Main Fig. 6a**).

Our main observation is that cell-type-specific TF motifs were enriched in cell-type-specific peak sets. For example, TP63 motif was significantly enriched in ESCC-specific peaks, and HNF4A motif in HepG2-specific peaks. In contrast, these cell-type-specific TFs were not observed in the ubiquitous peak set.

Since these peaks were all identified from SREBF1 ChIP-Seq experiment, these results suggest that SREBF1 cooperates with cell-type-specific TFs to regulate cell-type-specific peaks.

28. Page 15, line 336 – are the lipid enzyme genes in the 410 and 473 gene set? If not, how does the authors reconcile?

Authors' reply:

The Reviewer referred to the “410 genes associated with 473 peaks”, which were **ESCC-specific peaks**.

As demonstrated in our manuscript, genes belong to the fatty-acid metabolism and lipid metabolism process pathways were enriched in the **ubiquitous peak-set (Main Fig. 7a)**. Indeed, these data confirm the common function of SREBF1 in regulating lipid and fatty-acid synthesis, regardless of cell types.

Therefore, the lipid enzyme genes (e.g., FASN, ACLY, ACSS2, ACACA) were associated with ubiquitous peaks but not the ESCC-specific peaks. We also provided exemplary genes in the original **Main Fig. 6c** and **Supplementary Fig. S2c**.

29. Page lines 361-365 – the findings for the number of decreased expressions for the genes appear to be found in one of the two biological replicates for SREBF1, TP63, and KLF5. Authors need to revisit.

Authors' reply:

Following the Reviewer's comment, during the revision we have repeated this experiment for in three ESCC cell lines. As shown in **Figure R36**, among the 13 trio-occupied genes enriched in these two pathways, a total of 10 genes were significantly downregulated by all three TFs in at least two cell lines (highlighted by **red font**).

We have now provided both the heatmap and bar plots in the revised manuscript.

Figure R36. TE5, KYSE150 and KYSE510 cells were transfected with 30 nM siRNA targeting either SREBF1, TP63 or KLF5 for 48 hours, and were then subjected to qRT-PCR experiments. Barplots displaying the fold changes relative to scramble control. Mean±SD are shown, n=3. *, $P < 0.05$. N.D., not detectable.

30. For the mRNA heatmap shown in 7c, are these values significant as only overall FC is shown.

Authors' reply:

As replied above, we have now provided bar plots showing the statistics in the revised manuscript (**Supplementary Fig. S6**).

31. It would be helpful to arrange the methods in order of the experiments that are presented in the paper. Also, concentration of antibodies? How much protein? How were they imaged? Temperature and CO₂ levels for cell growth? Speed of centrifugation to remove debris in ChIP? What was used to reverse the crosslinks for ChIP?

Authors' reply:

As suggested, we re-ordered the Method section in the revised manuscript. The following details have also been added accordingly:

- 1) We have added the concentrations of all antibodies (**Page 21, line 477-489**).
- 2) We have added the loading amount of proteins (**Page 22, line 499**).
- 3) The methods to acquire each group of images have been described in the revised manuscript (**Page 28, line 618-619**).
- 4) Temperature and CO₂ levels for cell culture have been added (**Page 21, line 466**).
- 5) More details for ChIP experiments were added (**Page 23, line 514-515**).

We thank the Reviewer again for your valuable comments and suggestions!

References

1. Jiang, Y.Y. *et al.* TP63, SOX2, and KLF5 Establish a Core Regulatory Circuitry That Controls Epigenetic and Transcription Patterns in Esophageal Squamous Cell Carcinoma Cell Lines. *Gastroenterology* **159**, 1311-1327 (2020).
2. Whyte, W.A. *et al.* Master transcription factors and mediator establish super-enhancers at key cell identity genes. *Cell* **153**, 307-319 (2013).
3. Lambert, S.A. *et al.* The Human Transcription Factors. *Cell* **172**, 650-665 (2018).
4. Xu, C. *et al.* HNF4alpha pathway mapping identifies wild-type IDH1 as a targetable metabolic node in gastric cancer. *Gut* **69**, 231-242 (2020).
5. Chang, H.R. *et al.* HNF4alpha is a therapeutic target that links AMPK to WNT signalling in early-stage gastric cancer. *Gut* **65**, 19-32 (2016).
6. Xu, H. *et al.* Acyl-CoA synthetase short-chain family member 2 (ACSS2) is regulated by SREBP-1 and plays a role in fatty acid synthesis in caprine mammary epithelial cells. *Journal of cellular physiology* **233**, 1005-1016 (2018).
7. Sun, Y. *et al.* SREBP1 regulates tumorigenesis and prognosis of pancreatic cancer through targeting lipid metabolism. *Tumour biology : the journal of the International Society for Oncodevelopmental Biology and Medicine* **36**, 4133-4141 (2015).

8. Li, C.Q. *et al.* Integrative analyses of transcriptome sequencing identify novel functional lncRNAs in esophageal squamous cell carcinoma. *Oncogenesis* **6**, e297 (2017).
9. Saladi, S.V. *et al.* ACTL6A Is Co-Amplified with p63 in Squamous Cell Carcinoma to Drive YAP Activation, Regenerative Proliferation, and Poor Prognosis. *Cancer cell* **31**, 35-49 (2017).
10. Jiang, Y. *et al.* Co-activation of super-enhancer-driven CCAT1 by TP63 and SOX2 promotes squamous cancer progression. *Nat Commun* **9**, 3619 (2018).
11. Kamisuki, S. *et al.* A small molecule that blocks fat synthesis by inhibiting the activation of SREBP. *Chemistry & biology* **16**, 882-892 (2009).
12. Pan, J. *et al.* Lineage-Specific Epigenomic and Genomic Activation of Oncogene HNF4A Promotes Gastrointestinal Adenocarcinomas. *Cancer Res* **80**, 2722-2736 (2020).
13. Chen, L. *et al.* Master transcription factors form interconnected circuitry and orchestrate transcriptional networks in oesophageal adenocarcinoma. *Gut* **69**, 630-640 (2020).
14. Splinter, E., de Wit, E., van de Werken, H.J., Klous, P. & de Laat, W. Determining long-range chromatin interactions for selected genomic sites using 4C-seq technology: from fixation to computation. *Methods* **58**, 221-230 (2012).
15. Thongjuea, S., Stadhouders, R., Grosveld, F.G., Soler, E. & Lenhard, B. r3Cseq: an R/Bioconductor package for the discovery of long-range genomic interactions from chromosome conformation capture and next-generation sequencing data. *Nucleic Acids Res* **41**, e132 (2013).
16. Loven, J. *et al.* Selective inhibition of tumor oncogenes by disruption of super-enhancers. *Cell* **153**, 320-334 (2013).
17. Liberti, M.V. & Locasale, J.W. The Warburg Effect: How Does it Benefit Cancer Cells? *Trends Biochem Sci* **41**, 211-218 (2016).
18. Slebos, R.J. *et al.* Gene expression differences associated with human papillomavirus status in head and neck squamous cell carcinoma. *Clin Cancer Res* **12**, 701-709 (2006).
19. Cancer Genome Atlas, N. Comprehensive genomic characterization of head and neck squamous cell carcinomas. *Nature* **517**, 576-582 (2015).
20. Huang, M. *et al.* dbInDel: a database of enhancer-associated insertion and deletion variants by analysis of H3K27ac ChIP-Seq. *Bioinformatics* **36**, 1649-1651 (2020).
21. Peng, L. *et al.* Super-Enhancer-Associated Long Noncoding RNA HCCL5 Is Activated by ZEB1 and Promotes the Malignancy of Hepatocellular Carcinoma. *Cancer Res* **79**, 572-584 (2019).
22. Lin, L. *et al.* Super-enhancer-associated MEIS1 promotes transcriptional dysregulation in Ewing sarcoma in co-operation with EWS-FLI1. *Nucleic Acids Res* **47**, 1255-1267 (2019).
23. Jiang, Y.Y. *et al.* Targeting super-enhancer-associated oncogenes in oesophageal squamous cell carcinoma. *Gut* **66**, 1358-1368 (2017).
24. Qian, F.C. *et al.* SEanalysis: a web tool for super-enhancer associated regulatory analysis. *Nucleic Acids Res* **47**, W248-W255 (2019).
25. Maeder, M.L. *et al.* CRISPR RNA-guided activation of endogenous human genes. *Nat Methods* **10**, 977-979 (2013).
26. Baumann, V. *et al.* Targeted removal of epigenetic barriers during transcriptional reprogramming. *Nat Commun* **10**, 2119 (2019).

Reviewers' comments:

Reviewer #1 (Remarks to the Author):

The Authors employed a substantial effort in addressing most of the reviewers' concerns, the revised version of the manuscript is clearly improved and the evidence much better supported. The insights into the cooperation of TP63 and KLF5 in ESCC are of substantial interest for the field. However, the data fall short of identifying how SREBP1 regulates mTORC1/MEK and how signalling downstream of SREBP1 sustains ESCC proliferation or survival. A more focused approach definitively addressing these specific points would increase the enthusiasm for the manuscript.

Reviewer #2 (Remarks to the Author):

The authors have substantially revised the manuscripts and have addressed all major points raised. Importantly, they now provide additional clarification of experimental procedures and statistical analysis and revised the stringency of their conclusions. There is a small number of corrections that can still be included.

1) The reference for the Warburg effect should be: Warburg, O. Über den Stoffwechsel der Carcinomzelle. Die Naturwissenschaften 12, 1131-1137 (1924). Please also provide a reference for enhanced glutamine metabolism in cancer cells.

2) The two references for the effect of fatostatin and betulin (Refs 15 and 16) do not include cancer studies, as stated in the text. Please extend.

3) Lines 102-107: It is highly unusual to refer to specific experiments and results already in the introduction section. Can this be restructured, e.g. by not referencing the figure?

4) Line 207: insert "the" to make clearer. "three enhancers and the reporter"

5) Line 295ff: More details should be provided when explaining the results shown in figure S4. This figure does not contain any panels, which makes the specific references difficult.

6) Please include figures R12 and R22 also in the manuscript as supplementary information.

Reviewer #3 (Remarks to the Author):

The authors have provided an extensive amount of data and additional controls to address points raised by all the reviewers regarding rigor, methodology, and specific claims of the manuscript. Regarding the specific points raised by this reviewer, the authors have addressed them partially. As noted by multiple reviewers, some correlations and inferred mechanistic relationships are relatively weak and/or only partially supported.

Specific Points:

Fig. 2h (Fig. R25)– The mutant reporter is shown to have reduced basal activity. This finding is supportive but not an ideal demonstration of a direct regulatory effect of p63. Specifically, it would have been more convincing to show either a) effect of p63 overexpression and lack

of an effect of on the mutant enhancer, or b) effect of p63 knockdown that failed to occur with the mutant enhancer.

Fig. 3g (Fig. R26)- As for 2h, this experiment is not highly convincing because there is no direct manipulation of SREBF1 in the context of the reporter. Additionally, knockdown of SREBF1 gives nearly 90% reduction in activity for T3 enhancer, while mutagenesis gives only 50% -so this finding is only modestly supportive of the hypothesis of a direct regulatory relationship by SREBP.

Fig. 4. The data shown (Fig. R27) do not seem to provide strong evidence to support the central claim in the abstract of “SREBF1 as a central mediator linking TP63 with fatty-acid metabolism”. Specifically, R27 provides no quantitative analysis supporting specific co-regulation of lipid species by p63 and SREBF1 that support the statement.

Fig. 5. Regarding correlations among these transcription factors by IHC – here one can see the very weak correlation between p63 and SREBF1 compared to the strong correlation between p63 and KLF5, which already has an established regulatory relationship with p63 (e.g. Latil, Blanpain et al., 2017, Cell Stem Cell). So again these new data are not strongly supportive of some central claims.

Reviewer #4 (Remarks to the Author):

The authors have adequately addressed all concerns and comments. The manuscript is greatly improved and recommended for publication in *Nature Communications*.

REVIEWER COMMENTS

Reviewer #1 (Remarks to the Author):

The Authors employed a substantial effort in addressing most of the reviewers' concerns, the revised version of the manuscript is clearly improved and the evidence much better supported. The insights into the cooperation of TP63 and KLF5 in ESCC are of substantial interest for the field. However, the data fall short of identifying how SREBP1 regulates mTORC1/MEK and how signalling downstream of SREBP1 sustains ESCC proliferation or survival. A more focused approach definitively addressing these specific points would increase the enthusiasm for the manuscript.

Authors' reply:

We are truly delighted that the Reviewer is overall satisfied with our last revision work. And we thank you for the additional comment.

Regarding the comment on how SREBP1 regulates the mTORC1/MEK pathway, our following results together have revealed a clear mechanism at the molecular level:

- 1) From unbiased pathway enrichment analyses, the epigenomic data identified a total of eight genes in mTOR signaling pathway which were trio-occupied by SREBF1/TP63/KLF5 (**Main Fig.7b**, which is pasted here as **Figure R1A** for review convenience). Representative trio-binding peaks were also provided (*MAP2K2* and *SLC7A5* were shown in **Figure R1B-C**, and *WNT9A* was shown in **Main Fig.7d**).
- 2) During this 2nd round of revision, we selected the promoter element of *SLC7A5* for luciferase reporter assays. Importantly, our new data validated that silencing of SREBF1 reduced the luciferase activities of the *SLC7A5* promoter (**Figure R1C and R1D**) in two ESCC cell lines, confirming the direct regulatory effect of SREBF1 on the promoter activity.
- 3) We measured the mRNA expression of these eight genes in mTOR pathway in SREBF1-knockdown cell lines. The result showed that silencing of SREBF1 significantly decreased 7 out of the 8 factors across three ESCC cell lines (**Figure R1E**).
- 4) Lastly, Western Blotting assays validated that at the protein level, the mTOR signaling pathway was regulated by SREBF1 in ESCC cell lines (**Figure R1F**).

Together, these data have established a direct molecular mechanism underlying the regulation of SREBP1 on the mTOR pathway in SCC cells.

The Reviewer also asked how signaling downstream of SREBF1 (that is, Fatty-acid synthesis pathway, mTOR, and ERBB signaling pathways, **Main Fig.7f**) sustains ESCC proliferation or survival. Indeed, we did not highlight these data in our manuscript primarily because there are a number of published data supporting that these signaling pathways strongly promote ESCC proliferation and survival¹⁻¹². For example, Orita *et al.* reported that chemical inhibition of fatty-acid synthesis pathway reduced the growth of xenograft tumors of ESCC². Inhibition of either mTOR³⁻⁶ or ERBB pathways⁷⁻¹² also has well-documented anti-tumor properties against ESCC both *in vitro* and *in vivo*.

Figure R1. (A) Significantly enriched KEGG pathways of the corresponding genes assigned to the 274 ESCC-specific peaks trio-occupied by SREBF1, TP63 and KLF5. The number of genes enriched v.s. the total number of genes in each pathway are shown in the brackets. (B)-(C) ChIP-Seq profiles of indicated factors at *MAP2K2* or *SLC7A5* gene loci in SCC cell lines. (D) Promoter activity of *SLC7A5* was measured by luciferase reporter assays upon SREBF1 knockdown in TE5 or KYSE150 cells. Mean±SD are shown, n=5. **, $P < 0.01$. (E) TE5, KYSE150 and KYSE510 cells were transfected with 30 nM siRNA targeting either SREBF1 or scramble control, and were then subjected to qRT-PCR experiments. Barplots displaying the fold changes relative to the scramble control. Mean±SD are shown, n=3. *, $P < 0.05$. (F) Western blotting analyses following siRNA knockdown of SREBF1 in ESCC cells. Quantification of the blots is shown below.

Reviewer #2 (Remarks to the Author):

The authors have substantially revised the manuscripts and have addressed all major points raised. Importantly, they now provide additional clarification of experimental procedures and statistical analysis and revised the stringency of their conclusions. There is a small number of corrections that can still be included.

Authors' reply:

We greatly appreciate the Reviewer's positive comment on our manuscript!

1) The reference for the Warburg effect should be: Warburg, O. Über den Stoffwechsel der Carcinomzelle. Die Naturwissenschaften 12, 1131-1137 (1924). Please also provide a reference for enhanced glutamine metabolism in cancer cells.

Authors' reply:

As suggested, we have updated the reference for the Warburg effect and added a reference for enhanced glutamine metabolism in cancer cells¹³ in the revised manuscript.

2) The two references for the effect of fatostatin and betulin (Refs 15 and 16) do not include cancer studies, as stated in the text. Please extend.

Authors' reply:

As suggested, we added another two references for fatostatin¹⁴ and betulin¹⁵ in cancer studies in the revised manuscript.

3) Lines 102-107: It is highly unusual to refer to specific experiments and results already in the introduction section. Can this be restructured, e.g. by not referencing the figure?

Authors' reply:

Thank you for your suggestion. We have removed the reference for the figure in the revised Introduction section.

4) Line 207: insert "the" to make clearer. "three enhancers and the reporter"

Authors' reply:

Thank you. We have added it accordingly.

5) Line 295ff: More details should be provided when explaining the results shown in figure S4. This figure does not contain any panels, which makes the specific references difficult.

Authors' reply:

As suggested, we have now added more details and panels for this figure (**line 312-312 and line 316**; because we added new data in this round of 2nd revision, the original Supplementary Fig. 4 has now become Supplementary Fig. 7).

6) Please include figures R12 and R22 also in the manuscript as supplementary information.

Authors' reply:

These data of Figure R12 and R22 have been incorporated into the revised manuscript as **Supplementary Fig. S3**. We have also added description for this figure on **line 202-203** and **line 205**.

Reviewer #3 (Remarks to the Author):

The authors have provided an extensive amount of data and additional controls to address points raised by all the reviewers regarding rigor, methodology, and specific claims of the manuscript. Regarding the specific points raised by this reviewer, the authors have addressed them partially. As noted by multiple reviewers, some correlations and inferred mechanistic relationships are relatively weak and/or only partially supported.

Specific Points:

Fig. 2h (Fig. R25)– The mutant reporter is shown to have reduced basal activity. This finding is supportive but not an ideal demonstration of a direct regulatory effect of p63. Specifically, it would have been more convincing to show either a) effect of p63 overexpression and lack of an effect of on the mutant enhancer, or b) effect of p63 knockdown that failed to occur with the mutant enhancer.

Authors' reply:

We are abundantly appreciative of the Reviewer's careful evaluation of the work and, we thank you for your additional comments.

Regarding the Fig. R25, to further confirm a direct regulatory effect of TP63 on the enhancer element E3, we followed the Reviewer's suggestion and performed ectopic expression of TP63. We first verified the over-expression of TP63 at both the mRNA (**Figure R2A**) and protein (**Figure R2B**) levels in two ESCC cell lines.

We then measured the luciferase reporter activity of the wildtype and mutant enhancer. As expected, over-expression of TP63 increased significantly the reporter activity of the wildtype enhancer (**Figure R2C**). In contrast, over-expression of TP63 produced no detectable effect on the mutant enhancer (**Figure R2C**). These results validate a direct regulatory effect of TP63 on the enhancer activity of E3.

We have incorporated these new results into the revised manuscript (**Supplementary Figure 4a-c; line 215-219**).

Figure R2. (A) qRT-PCR and **(B)** Western blotting analyses of the overexpression efficiency 48 hours after the transfection with either empty vector (OE EV) or TP63 (OE TP63) in ESCC cell lines. Mean±SD are shown, n=3. **(C)** TE5 and KYSE150 cells were transfected with either empty vector (OE EV) or TP63 (OE TP63) for 24 hours, and then co-transfected with both the renilla plasmid and pGL3-promoter plasmid containing either wild-type or mutant E3 enhancer for another 48 hours. The relative transcriptional reporter activity was analyzed by dual luciferase reporter gene assay. Mean±SD are shown, n=5. **, $P < 0.01$.

Fig. 3g (Fig. R26)- As for 2h, this experiment is not highly convincing because there is no direct manipulation of SREBF1 in the context of the reporter. Additionally, knockdown of SREBF1 gives nearly 90% reduction in activity for T3 enhancer, while mutagenesis gives only 50% -so this finding is only modestly supportive of the hypothesis of a direct regulatory relationship by SREBP. Authors' reply:

We similarly conducted over-expression assay of SREBF1 to confirm the regulatory effect on T2/T3 elements. Upon validating the successful over-expression at mRNA (**Figure R3A**) and protein levels (**Figure R3B**), we performed the luciferase reporter assays of the wildtype and mutant elements in two ESCC cell lines.

Indeed, over-expression of SREBF1 consistently increased the reporter activity of both the wildtype T2/T3 elements, but failed to affect the mutant T2/T3. Therefore, these results supported a direct regulatory activity by SREBF1 on these elements.

We have incorporated these new results into the revised manuscript (**Supplementary Fig. S5a-d; Line 246-249**).

The Reviewer also noted that knockdown of SREBF1 caused larger size-effect than the mutagenesis approach. We consider that this may be explained by the cooperation between SREBF1 and other co-binding TFs, which can create additional non-canonical motifs for SREBF1 to occupy. Indeed, it has been shown that cooperating TFs can together compete with nucleosomes with higher potency and occupy DNA elements with more flexible motif sequence¹⁶.

Figure R3. (A) qRT-PCR and **(B)** Western blotting analyses of the overexpression efficiency 48 hours after the transfection with either empty vector (OE EV) or SREBF1 (OE SREBF1) in ESCC cell lines. Mean±SD are shown, n=3. **(C)** TE5 and KYSE150 cells were transfected with either empty vector (OE EV) or SREBF1 (OE SREBF1) for 24 hours, and then co-transfected with both the renilla plasmid and pGL3-promoter plasmid containing either wild-type or mutant T2/T3 enhancers for another 48 hours. The relative transcriptional reporter activity was analyzed by dual luciferase reporter gene assay. Mean±SD are shown, n=5. **, $P < 0.01$.

Fig. 4. The data shown (Fig. R27) do not seem to provide strong evidence to support the central claim in the abstract of “SREBF1 as a central mediator linking TP63 with fatty-acid metabolism”. Specifically, R27 provides no quantitative analysis supporting specific co-regulation of lipid species by p63 and SREBF1 that support the statement.

Authors' reply:

Prompted by the Reviewer's comment, we performed additional quantitative analysis for the altered lipid classes caused by either SREBF1-inhibition or TP63-knockdown. Specifically, we asked whether lipid classes that were regulated by SREBF1 were also controlled by TP63.

Importantly, Fisher's exact test showed that lipid classes under control of SREBF1 were significantly more likely to be also regulated by TP63 (**Table R1, $P = 0.02$**), suggesting a functional connection between TP63 and SREBF1 in the regulation of lipid classes.

Moreover, to further strengthen the conclusion that SREBF1 acts as a key mediator linking TP63 with fatty-acid metabolism, we performed a number of new experiments during this 2nd round of revision, as summarized below:

- 1) We performed rescue assays to interrogate the regulation of TP63/SREBF1 on the key rate-limiting enzymes for fatty-acid synthesis. Specifically, knockdown of TP63 by two independent shRNAs strongly inhibited the mRNA levels of these central enzymes. Importantly, over-expression of SREBF1 consistently rescued the decreased expression of all of these factors in two ESCC cell lines (**Figure R4**). In some genes, such as SCD and CERS1, the rescue effect was almost complete.
- 2) On the other hand, we found that over-expression of TP63 enhanced the mRNA levels of these five enzymes. Notably, knockdown of SREBF1 largely and consistently reversed this effect in both TE5 and KYSE150 cells (**Figure R5**).
- 3) We also performed rescue assays to measure the changes in lipid droplet content. Indeed, over-expression of TP63 increased the lipid droplet staining, which was abolished by the knockdown of SREBF1 in both TE5 and KYSE150 cells (**Figure R6**).
- 4) Conversely, knockdown of TP63 reduced the lipid droplet staining, and over-expression of SREBF1 potently rescued the level of lipid droplet (**Figure R7**).

Taken together, we believe that these new results, plus multiple lines of evidence from our original findings (Main Fig.1 and 4), strongly support that SREBF1 functions as a key mediator linking TP63 with fatty-acid metabolism.

We have also added these new data into the revised manuscript (**Supplementary Fig. S2a-c; Line 168-175**).

Table R1. Altered lipid classes between SREBF1-inhibition group v.s. TP63-knockdown group.

TP63-knockdown group	SREBF1-inhibition group		P value (Fisher's exact test)
	Significantly changed classes	Unchanged classes	
Significantly changed classes	13	0	0.02
Unchanged classes	3	3	

Figure R4. TE5 and KYSE150 cells were transfected with shRNA plasmids targeting either TP63 or scramble for 24 hours, followed by the transfection of plasmids encoding either empty vector (OE EV) or SREBF1 (OE SREBF1) for another 48 hours. The samples were then subjected to qRT-PCR experiments. Barplots displaying the fold changes relative to scramble control. Mean±SD are shown, n=3. *, $P<0.05$; **, $P<0.01$.

Figure R5. TE5 and KYSE150 cells were transfected with plasmids encoding either empty vector (OE EV) or TP63 (OE TP63) for 24 hours, followed by the transfection of shRNA plasmids targeting either SREBF1 or scramble for another 48 hours. The samples were then subjected to qRT-PCR experiments. Barplots displaying the fold changes relative to control. Mean±SD are shown, n=3. *, $P<0.05$; **, $P<0.01$.

Figure R6. TE5 and KYSE150 cells were transfected with plasmids encoding either empty vector (OE EV) or TP63 (OE TP63) for 24 hours, followed by the transfection of shRNA plasmids targeting either SREBF1 or scramble for another 48 hours. Lipid droplet was stained and analyzed with confocal microscopy. Fluorescence intensity was quantified in each group and plotted as fold-changes relative to the control group of OE EV+scramble. Mean±SD are shown, n=4. **, $P<0.01$.

Figure R7. Stable TP63-silenced TE5 and KYSE150 cells were transfected with plasmids encoding either empty vector (OE EV) or SREBF1 (OE SREBF1) for 48 hours. Lipid droplet was stained and analyzed with confocal microscopy. Fluorescence intensity was quantified in each group and plotted as fold-changes relative to the control group of Scramble+OE EV. Mean±SD are shown, n=5. **, $P < 0.01$.

Fig. 5. Regarding correlations among these transcription factors by IHC – here one can see the very weak correlation between p63 and SREBF1 compared to the strong correlation between p63 and KLF5, which already has an established regulatory relationship with p63 (e.g. Latil, Blanpain et al., 2017, Cell Stem Cell). So again these new data are not strongly supportive of some central claims.

Authors' reply:

Indeed, we agree with the Reviewer that the correlation between nuclear TP63 and nuclear SREBF1 was much weaker than the well-established correlation between TP63/KLF5. We reason that this weak correlation at the subcellular protein level is partially attributed to the complex activation and regulation of SREBF1 protein which involves three different subcellular compartments: endoplasmic reticulum, Golgi and nucleus¹⁷.

However, our interpretation and analyses of the correlation between TP63/SREBF1 is in order to investigate the transcriptional co-regulation between these factors as identified in **Main Fig.2-3**. Therefore, in this context, the more appropriate data to consider is the mRNA correlation (**Main Fig.3h** and **Supplementary Fig. S5e**) since this supports the transcriptional regulation.

We have added the discussion on the weak correlation between TP63/SREBF1 in the revised manuscript (**Line 427-431**).

Reviewer #4 (Remarks to the Author):

The authors have adequately addressed all concerns and comments. The manuscript is greatly improved and recommended for publication in *Nature Communications*.

Authors' reply:

We are very pleased that the Reviewer recommends our manuscript for publication in *Nature Communications*!

References:

1. Chen, Y. *et al.* Simvastatin, but not pravastatin, inhibits the proliferation of esophageal adenocarcinoma and squamous cell carcinoma cells: a cell-molecular study. *Lipids in health and disease* **17**, 290 (2018).
2. Orita, H. *et al.* High levels of fatty acid synthase expression in esophageal cancers represent a potential target for therapy. *Cancer biology & therapy* **10**, 549-554 (2010).
3. Hou, G. *et al.* mTOR inhibitor rapamycin alone or combined with cisplatin inhibits growth of esophageal squamous cell carcinoma in nude mice. *Cancer letters* **290**, 248-254 (2010).
4. Nishikawa, T. *et al.* Antiproliferative effect of a novel mTOR inhibitor temsirolimus contributes to the prolonged survival of orthotopic esophageal cancer-bearing mice. *Cancer biology & therapy* **14**, 230-236 (2013).
5. Huang, Y. *et al.* A dual mTORC1 and mTORC2 inhibitor shows antitumor activity in esophageal squamous cell carcinoma cells and sensitizes them to cisplatin. *Anti-cancer drugs* **24**, 889-898 (2013).
6. Kim, S.H. *et al.* Clinicopathologic significance and function of mammalian target of rapamycin activation in esophageal squamous cell carcinoma. *Human pathology* **44**, 226-236 (2013).
7. Hara, F. *et al.* Antitumor effect of gefitinib ('Iressa') on esophageal squamous cell carcinoma cell lines in vitro and in vivo. *Cancer letters* **226**, 37-47 (2005).
8. Ako, E. *et al.* The pan-erbB tyrosine kinase inhibitor CI-1033 inhibits human esophageal cancer cells in vitro and in vivo. *Oncology reports* **17**, 887-893 (2007).
9. Mimura, K. *et al.* Lapatinib inhibits receptor phosphorylation and cell growth and enhances antibody-dependent cellular cytotoxicity of EGFR- and HER2-overexpressing esophageal cancer cell lines. *International journal of cancer. Journal international du cancer* **129**, 2408-2416 (2011).
10. Fichter, C.D. *et al.* ErbB targeting inhibitors repress cell migration of esophageal squamous cell carcinoma and adenocarcinoma cells by distinct signaling pathways. *Journal of molecular medicine* **92**, 1209-1223 (2014).

11. Wong, C.H., Ma, B.B., Hui, C.W., Tao, Q. & Chan, A.T. Preclinical evaluation of afatinib (BIBW2992) in esophageal squamous cell carcinoma (ESCC). *American journal of cancer research* **5**, 3588-3599 (2015).
12. Kim, H.S. *et al.* Phase II clinical and exploratory biomarker study of dacomitinib in recurrent and/or metastatic esophageal squamous cell carcinoma. *Oncotarget* **6**, 44971-44984 (2015).
13. Hensley, C.T., Wasti, A.T. & DeBerardinis, R.J. Glutamine and cancer: cell biology, physiology, and clinical opportunities. *The Journal of clinical investigation* **123**, 3678-3684 (2013).
14. Brovkovich, V. *et al.* Fatostatin induces pro- and anti-apoptotic lipid accumulation in breast cancer. *Oncogenesis* **7**, 66 (2018).
15. Han, Y.H., Mun, J.G., Jeon, H.D., Kee, J.Y. & Hong, S.H. Betulin Inhibits Lung Metastasis by Inducing Cell Cycle Arrest, Autophagy, and Apoptosis of Metastatic Colorectal Cancer Cells. *Nutrients* **12** (2019).
16. Long, H.K., Prescott, S.L. & Wysocka, J. Ever-Changing Landscapes: Transcriptional Enhancers in Development and Evolution. *Cell* **167**, 1170-1187 (2016).
17. Shimano, H. & Sato, R. SREBP-regulated lipid metabolism: convergent physiology - divergent pathophysiology. *Nature reviews. Endocrinology* **13**, 710-730 (2017).

Reviewers' comments:

Reviewer #3 (Remarks to the Author):

The authors have performed the specific experiments requested which generally strengthen support for the proposed model. Importantly, the data are also appropriately qualified in the descriptions. For example, Figure 3h demonstrates the “three TFs were modestly correlated with each other”. Indeed, some of the experiments show evidence that is “modest” in its strength, but overall the proposed model appears adequately supported.

REVIEWER COMMENTS

Reviewer #3 (Remarks to the Author):

The authors have performed the specific experiments requested which generally strengthen support for the proposed model. Importantly, the data are also appropriately qualified in the descriptions. For example, Figure 3h demonstrates the “three TFs were modestly correlated with each other”. Indeed, some of the experiments show evidence that is “modest” in its strength, but overall the proposed model appears adequately supported.

Authors' reply:

Thank you for the positive comments on our work.